

# Drawing lessons for multi-model ensemble design from emulator experiments: application to future sea level contribution of the Greenland ice sheet

Jeremy Rohmer[1], Heiko Goelzer[2], Tamsin Edwards[3], Goneri Le Cozannet[1], Gael Durand[4]

[1]BRGM, 3 av. C. Guillemin - 45060 Orléans Cedex 2 - France
[2]NORCE Norwegian Research Centre, Bjerknes Centre for Climate Research, Bergen, Norway
[3]Department of Geography, King's College London, Bush House, North East Wing, 40 Aldwych, London, WC2B 4BG, London, UK
[4]Univ. Grenoble Alpes, CNRS, IRD, Grenoble INP, IGE, 38000 Grenoble, France

*Correspondence to*: Jeremy Rohmer (j.rohmer@brgm.fr)

**Abstract.** Multi-model ensembles (MME) are key ingredients for future climate projection and the quantification of its uncertainty. Developing robust protocols to design balanced and complete computer experiments for MME is a matter of active research. In this study, we take advantage of a large-size MME produced for Greenland ice sheet contributions to future sea level by 2100 to define a series of computer experiments that are closely related to practical MME design decisions: what is the added value of including specific set of experiments in the projections, i.e. either adding new models (Regional Climate Model RCM, or Ice Sheet Model ISM) or extending the range of some parameter values. By using these experiments to build a random-forest-based emulator, changes in the emulator's predictive performance and the emulator-based probabilistic projections provided information on several aspects: (1) the utmost importance of including the SSP5-8.5 scenario, due to the large number of simulations available and the range of global warming they cover; (2) the importance of having diverse ISM and RCM models; (3) the lesser importance of the choice in the range of the Greenland tidewater glacier retreat parameter. We expect these recommendations to be informative for the design of next generations of MME, in particular for the next Ice Sheet Model Intercomparison Project ISMIP7 in preparation.

## 1 Introduction

Multi-model ensembles (MME) are key ingredients for future climate projection and the quantification of its uncertainty. They consist of co-ordinated sets of numerical experiments performed under common forcing conditions with different model designs (i.e. different model formulations, input parameter values, initial conditions, etc.) to generate multiple realisations known as ensemble members. This is the approach of Model Intercomparison Projects, MIPs, which are key for the understanding of past, present, and future climates and contribute to assessments from the Intergovernmental Panel on Climate Change (IPCC: e.g. Lee et al., 2021). In this study, we are interested in projected Greenland ice sheet contributions to sea level



change this century, which are the subject of recent MME studies (Goelzer et al., 2018; 2020) within the Ice Sheet Model Intercomparison Project for CMIP6 (ISMIP6: Nowicki et al., 2016; 2020).

However, interpreting MME results is complicated by the choices made in their construction (e.g. Knutti et al., 2010). Ideally, each member of a MME should evenly span a representative and exhaustive set of plausible realisations of the combined sources of uncertainty, e.g. distinct climate models with different but plausible strategies for simulating the global climate (GCMs), equally represented by a single model run. However, members of a MME are often structurally similar, and the degree of their dependence is difficult to quantify (e.g. Merrifield et al., 2020). This difficulty is particularly emblematic of the Coupled Model Intercomparison Project (CMIP), coined an "ensemble of opportunity" (Tebaldi and Knutti, 2007) because it collects "best guesses" (Merrifield et al., 2020) from modelling groups with the capacity to participate. This capacity may range from substantial resources to develop climate models and perform relatively large ensembles through to the ability to perform only a small number of simulations with an existing version of a climate model. These disparities, combined with the high computational expense of climate models and the partial dependence of MME members, results in limited and unbalanced multi-model ensemble designs, in which various combinations of modelling choices and forcing conditions are either over-represented or missing in the MME, and a full sampling of modelling uncertainties is impossible to perform or even to define. Section 2.1 provides in the following an illustration for the MME considered in this study.

Emulators (also named surrogate models) have been proposed to address these limitations. An emulator is a fast statistical approximation of a computationally expensive numerical model, often building on machine learning techniques. Their key advantage is that they can be used to predict the numerical model's response at untried input values, to explore the uncertain input space far more thoroughly: potentially overcoming the incompleteness of ensemble designs and being used to produce probabilistic projections.

Some emulation studies have broadened this approach to represent entire MME at once, rather than individual models. One example in this field is provided by Edwards et al. (2021), who emulate ISMIP6 simulations for the Greenland and Antarctic ice sheets and multi-model glacier ensembles, driven by multi-model climate model ensemble simulations, to estimate land ice contributions to twenty-first-century sea level rise. Emulating an MME requires an assumption (and check) that the simulations are quasi-independent: i.e. that the differences induced by different model setups (in particular, initialisation) outweigh any similarities induced by common model structures. This was found by Edwards et al. (2021) to be the case for ice sheet and glacier MMEs. Another type of application is provided by Van Breedam et al. (2021) who used emulators to perform a large number of sensitivity tests with numerical simulations of ice sheet–climate interactions on a multi-million-year timescale.

In this study, we aim to explore how the results provided by an emulator can be informative for the design of an MME. Key design questions relate to the added value of including specific sets of experiments in the projections, i.e. either adding new models (e.g. new Regional Climate Model, RCM, new GCM, etc.) or extending the range of some parameter values (e.g., the Antarctic basal melt parameter or Greenland tidewater glacier retreat parameter described by Edwards et al. (2021)). To address these questions, we take advantage of a large MME of Greenland ice sheet contributions to sea level this century, based on



which we define a series of validation tests (referred to as emulator's experiments) that are closely related to practical MME design decisions. The evaluation of the emulator prediction capability with each of these experiments is used to provide information on the added value of including specific set of experiments.

The paper is organized as follows. We first describe the sea level numerical simulations as well as details of the statistical methods used to build the emulator and assess the different design questions (Section 2). In Section 3, we apply the experiments

and assess the influence of each design question. We discuss results in Section 4, and we draw lessons and guidance related to the MME design, and discuss the implications from a stakeholder's point of view. Finally, we conclude in Section 5.

## 2. Data and methods

### 2.1 Multi-model ensemble case study

We focus on the sea level contribution from the Greenland ice sheet (GrIS) in 2100 based on a new MME study performed for

the European Union's Horizon 2020 project PROTECT (http://protect-slr.eu). Some modelling choices are taken from the protocols of the ISMIP6 initiative (Goelzer et al. (2020): in particular, the two main emissions scenarios, and the main model parameter explored. In the following, we provide a brief summary of the GrIS MME dataset and refer the interested reader to Goelzer et al. (2020) and references therein for further details, where appropriate.

The full modelling chain for these projections combines: (1) a number of CMIP5 and CMIP6 GCMs that produce climate

projections according to different emissions scenarios; (2) two Regional Climate Models (RCMs), and their variants, that locally downscale the GCM forcing to the GrIS surface; (3) a range of ISM models that produce projections of ice mass changes and sea level contributions (initialised to reproduce the present-day state of the GrIS as best as possible, at a given initial year sometime before the start of emissions scenarios in 2015). The ISMs are forced by surface mass balance (SMB) changes from the RCMs, added to their own reference SMB assumed during initialisation. Marine-terminating outlet glaciers

are in turn forced by an empirically-derived parameterisation that relates changes in meltwater runoff from the RCM and ocean temperature changes from the GCMs to the retreat of calving front positions (Slater et al., 2020). The parameter that controls retreat is denoted $\kappa$ and is used to sample uncertainty in the parameterisation (Slater et al., 2019).

In what follows, we use the generic term 'inputs' to designate all the choices made throughout the modelling chain, i.e. the choices in the models used, the choices in the scenarios and the parameter values. The inputs are described in detail in Table

1. It should be noted that the two first inputs, i.e. the choice in the SSP-RCP scenario and in the GCM model, are not considered for the emulator construction described in Sect. 2.2. They are combined with a similar approach as Edwards et al. (2021), by relating each 'SSP-RCP, GCM' combination to the corresponding value of global annual mean surface air temperature change since 2015, denoted GSAT.




**Table 1: Inputs considered in the GrIS MME. The inputs listed below the double line are those used for the building of the RF emulator described in Sect. 2.2.**

| Type | Symbol | Type of variable | Value range / Categories |
|---|---|---|---|
| Future climate and societal conditions | SSP-RCP | Categorical | 5 scenarios: three Shared Socio-economic Pathways (SSP1-2.6, SSP2-4.5, SSP5-8.5) and two Representative Concentration Pathways (RCP2.6, RCP8.5). The latter, older, scenarios are grouped with the nearest equivalent SSPs (RCP2.6 with SSP1-2.6; RCP8.5 and SSP5-8.5). |
| General Circulation Model | *GCM* | Categorical | 15 global climate models: ACCESS1.3, CESM2, CESM2-Leo*, CESM2-WACCM, CNRM-CM6-1, CNRM-ESM2-1, CSIRO-Mk3.6.0, HadGEM2-ES, IPSL-CM5A-MR, IPSL-CM6A-LR, MIROC5, MPI-ESM1-2-HR, NorESM1-M, NorESM2-MM, UKESM1-0-LL-r1 |
| Global mean temperature change 2015−2100 | GSAT change | Continuous | The joint influence of SSP-RCP and GCM is treated with a similar approach as Edwards et al. (2021), by relating each 'SSP-RCP and GCM' combination to the corresponding value of global annual mean surface air temperature change since 2015. |
| Ice Sheet Model | *ISM* | Categorical | 4 models: CISM, Elmer/Ice, GISM, IMAUICE |
| Regional Climate Model | *RCM* | Categorical | 6 model approaches: four versions of the RCM MAR (v3.9, v3.12, v3.13-e05, and v3.13-e55), one version of the RCM RACMO (v2.3p2), and statistical downscaling (SDBN1). |
| Retreat parameter | $\kappa$ | Continuous | From -0.9705 to +0.0070 km.$(m^3.s^{-1})^{-0.4}$ °C |
| Minimal spatial resolution | *res_min* | Continuous | From 1 to 40 km |
| Sliding basal Law | *Sliding* | Categorical | 5 laws: Coulomb, Linear, Schoof, Weertman, Zoet-Iverson |
| Account for thermodynamics | *thermodin.* | Categorical | TRUE or FALSE |
| RCM used for initialisation | *RCM_init* | Categorical | 4 model variants: IMAU-ITM, and MAR (v3.9, v3.11.5, and v3.12). |
| Type of initialisation method | *Init* | Categorical | Data assimilation based on velocities (DAv); nudging to ice mask (NDm); or nudging to surface elevation (NDs). |
| Number of years of the initialisation period | *init_yrs* | Continuous | From 20 to 240,000 years |
| Location of the surface elevation feedback | *elev_feedback* | Categorical | In the ice sheet model (with two formulations of the SMB-elevation gradient, X or B), or in the regional climate model RCM. |

*\*CESM2-Leo is a variant pre-dating the official CESM2 release for CMIP6. It can be considered as another ensemble member of CESM2.*




One input setting defines a member of the MME. Formally, the inputs are either treated as continuous variables (e.g., for $\kappa$, minimum resolution), or as categorical variables (e.g., RCM or ISM choice). The considered MME comprises $n=1{,}303$ members, which are used to estimate the sea level contribution in 2100 (denoted *slc* expressed in meters sea level equivalent SLE) with respect to 2015. Figure 1 shows a probability density distribution of *slc* constructed directly using the members of

the MME, which has a median value of 8.7 cm SLE and 5% and 95% quantiles of 3.1 and 19.9 cm; the latter being used to define the 90% credibility interval.

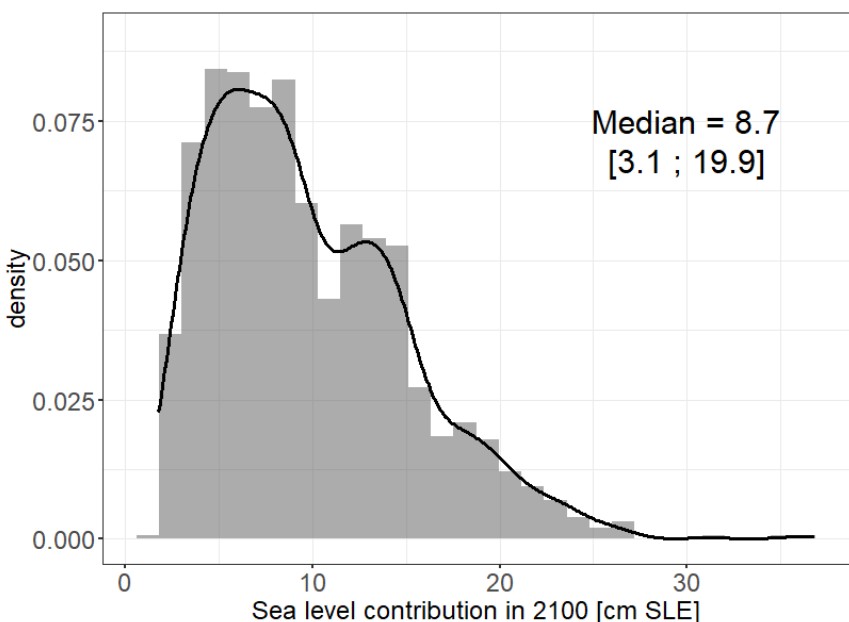

**Figure 1: (a) Probability density function of the sea level contribution in 2100 (with respect to 2015) from the Greenland ice-sheet (in cm seal level equivalent, SLE) based on the raw MME ensemble data considered in this study. The black straight line is calculated**
**from a Gaussian kernel density estimation with a bandwidth chosen by following Silverman (1986)'s 'rule of thumb'. The median value and the 90% credibility interval are also indicated.**

Figures 2 and 3 show the histograms for a selection of the continuous and categorical variables described in Table 1. For sake of space, we focus here on the 7 of 11 variables identified to have the highest importance with respect to *slc* (see Sect. 3 and

Appendix B). Figure 2 shows that the design of experiments is clearly unbalanced for some categories, e.g. *Elmer/Ice* and *GISM* models for the choice in the ISM, S*DN1* for the choice in the RCM, and the use of a RCM for the choice in the approach to represent the feedback between the ice sheet surface elevation and climate (variable named *elev_feedback*). Unbalanced distributions are also clear for continuous variables as shown in Fig. 3. Furthermore, gaps in the distributions are outlined, e.g. $\kappa$ between -0.9705 and -0.3700 km.$(\text{m}^3.\text{s}^{-1})^{-0.4}$ °C and the minimum spatial ISM resolution between 20 and 40 km. Both Figures

2 and 3 illustrate the unbalanced and incomplete nature of the design of experiments.





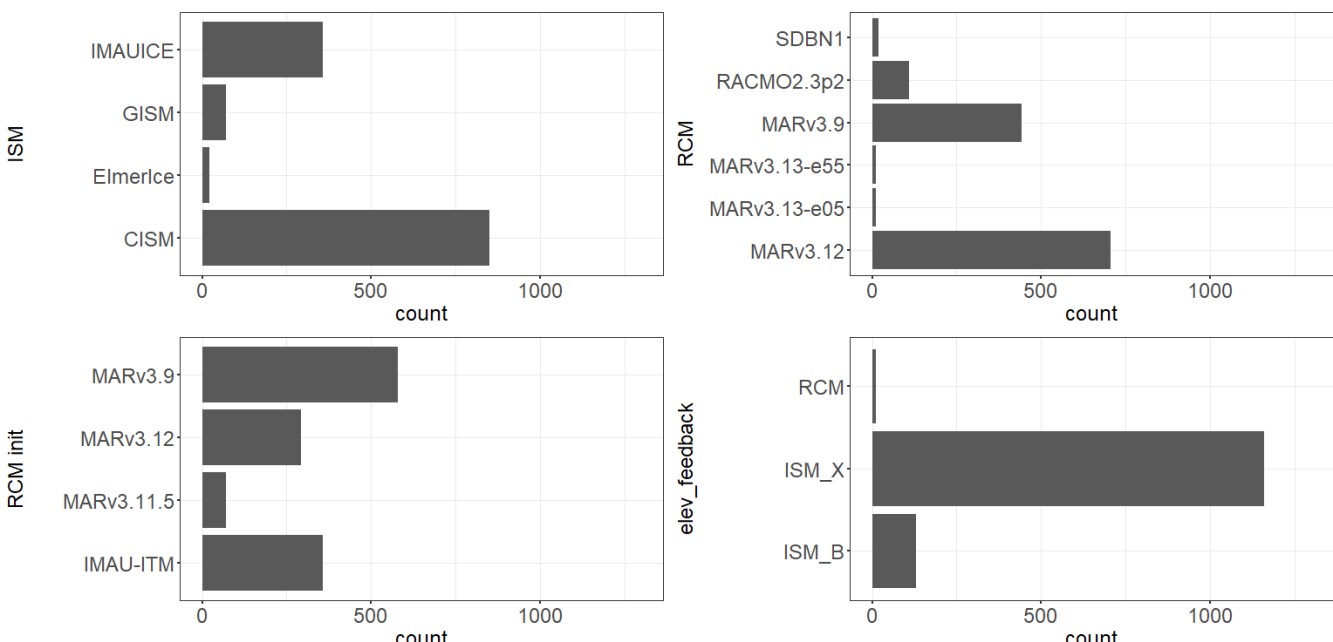

**Figure 2: Count number of the MME members with respect to the different inputs classified as "categorical" in Table 1: ISM (ice sheet model), RCM (regional climate model used for downscaling climate projections), RCM init (regional climate model used for initialisation climate), and elev_feedback (approach to representing the feedback between the ice sheet surface elevation and climate).**

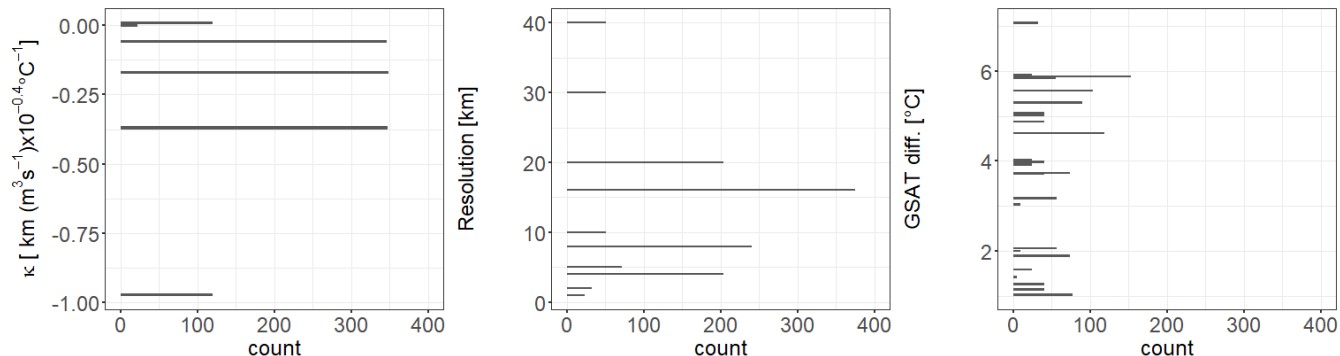

**Figure 3: Count number of the MME members with respect to the different inputs classified as "continuous" in Table 1: $\kappa$ (ice sheet tidewater glacier retreat parameter), minimum spatial resolution of the ice sheet model, and GSAT diff (global mean surface air temperature change during the driving global climate model simulation).**





## 2.2 Setting up the emulator

The objective is to evaluate the sea level contribution *slc* (with respect to a reference date) at a given time t. The chain of
models described in Sect. 2.1, denoted f, is used to numerically simulate *slc*. The different models (part of the MME) are
assumed to share the same characteristics, which correspond to p different inputs described in Table 1. Mathematically, a
random variable *x* is assigned to each of these inputs, and the vector of p input variables is denoted by $\mathbf{x} = \{x_1, x_2, ..., x_p\}$. The
MME results (of size n) at a given time t are $\{slc^{(i)}, \mathbf{x}^{(i)}\}_{i=1,...,n}$ with $slc^{(i)} = f(\mathbf{x}^{(i)})$. Since our knowledge on the mathematical
relationship f is only partial and based on the n MME results, we replace f by a machine-learning-based proxy (named emulator)
built using the MME results. The main advantage is to be able to make predictions for input configurations that are not present
in the original MME dataset at a low computation time cost.

Among the different types of emulators (see e.g., Yoo et al. (2024) for a recent overview of different options), we focus in this
study on the Random Forest (RF) regression model, as introduced by Breiman (2001). The interested reader can refer to
Appendix A for further technical details. RF has shown high efficiency in different sea level projection studies (Hough &
Wong, 2022; Rohmer et al., 2022; Turner et al., 2024). More importantly, this emulator has the advantage of dealing, by
construction, with different mixed types of input variables, categorical and continuous, which is a key aspect in our case (see
Table 1).

## 2.3 Emulator experiments related to design questions

In this study, we address a series of questions described in Table 2 that are relevant for the design of MMEs.
In general, the central concern is to investigate what is the added value of including a specific set of experiments in the
projections. This could be subsets in already defined value range / categories, or subsets not currently categorised. For four
different categories of inputs related to specific modelling choices (choice in SSP-RCP, choice in RCM, choice in ISM, and
range of $\kappa$ values), the design questions are formalised in Table 2. To assess the added value of including a specific set of
experiments in the projections, we propose to construct RF emulators by leaving out specific results from the original MME.
The last column of Table 2 translates the design questions into a specific emulator's experiment. Using a RF emulator trained
with the complete original MME as a reference solution, we assess changes in three types of criteria: changes in the MME
characteristics, performance of the RF emulator, and the probability estimates of *slc* in 2100 given future GSAT change
scenarios, here chosen at 2°C, 3°C or 4 °C. The details of this assessment are explained in Sect. 2.4.






**Table 2: Design questions and corresponding emulator's experiments. Modelling choices are evaluated based on the RF emulator**
**performance and the probability estimate of *slc* in 2100 given GSAT at 2, 3 or 4°C.**

| Input | Question | Definition of the emulator's experiment | Name of the experiment | Number of members* |
|---|---|---|---|---|
| SSP-RCP scenario | Does including a medium scenario SSP2-4.5 improve the results or is it enough to use the extreme scenarios SSP1-2.6 and SSP5-8.5? | A RF emulator is trained using only the results for SSP1-2.6 & SSP2-4.5, i.e. without SSP5-8.5 | Without SSP5-8.5: 'woSSP585' | 418 (32%); |
| | | SSP1-2.6 & SSP5-8.5, without SSP2-4.5; | 'woSSP245' | 1,114 (86%) |
| | | SSP2-4.5 & SSP5-8.5, without SSP1-2.6 | 'woSSP126' | 1,074 (83%) |
| RCM choice | What is the added value of including a new RCM, i.e. is it sufficient to focus on MAR regional climate model (Fettweis et al., 2017) only? | A RF emulator is built using only the results for MAR (regardless of the version: MARv3.12, MARv3.13-e05, MARv3.13-e55, or MARv3.9), in particular without Regional Atmospheric Climate Model RACMO (Ettema et al., 2010). | 'MAR' | 1,143 (88%) |
| ISM choice | What is the added value of accounting for all ISM except for one? | A RF emulator is trained using only the results for the most selected ISM, namely the Community Ice Sheet Model (CISM; Lipscomb et al., 2019) | 'CISM' | 851 (65%) |
| | | Built without the results of CISM (experiment 'woCISM'). | 'woCISM' | 452 (35%) |
| Range of $\kappa$ values | Should the design cover a large range of values, i.e. is it sufficient to focus on extreme values? | A RF emulator is built using the central value of -0.1700 and the endpoints, of -0.9705 and 0.007 km.(m³.s⁻¹)$^{-0.4}$ °C only, i.e. without intermediate values. | 'Med. & Extr.' | 588 (45%) |
| | | Built only with central and medium values, from -0.37 to 0 km.(m³.s⁻¹)$^{-0.4}$°C. | 'Narrow' | 588 (55%) |

*% of the total number of members

## 2.4 Criteria for measuring the impact of the design questions

The first set of criteria aims to measure the extent to which the new subset of the MME differs from the original MME, with
two indicators: (1) the percentage decrease in size ($D_S$); (2) the changes in the histograms of the new MME subsets with respect
to the original ones (as depicted in Fig. 2 and 3) for each inputs used to build the RF emulator (see Supplementary materials
S1 for an illustration). The latter is defined as the average difference in the count numbers between the two histograms
(normalised by the total number of members).



The second criterion measures the decrease in the predictive performance of the emulator, using a metric of relative error. It is
assessed through a validation test exercise that consists in randomly selecting $n_{\text{test}}$ test samples from the original MME,
conducting the experiments described in Table 2, and estimating the *slc* error, i.e. $e^{(i)} = slc^{(i)} - \widehat{slc}^{(i)}$ by comparing the true
and the RF emulator's predicted value for each test sample i=1,…,$n_{\text{test}}$. To ensure that the test samples cover a broad range of
situations, they are selected randomly as follows: (1) the GSATs are classified into a finite number of intervals, the ends of
which are defined by the GSAT percentiles, with levels ranging from 0 to 100% with a fixed increase of 10%; (2) for each
interval, five samples are randomly selected. For one iteration of the procedure, a total of $n_{\text{test}}$=55 test samples are randomly
selected, and the mean relative error is estimated, $RAE = \frac{1}{n_{\text{test}}} \sum_{i=1}^{n_{\text{test}}} \left| \frac{e^{(i)}}{slc^{(i)}} \right|$ (quoted as a percentage). This predictive
performance indicator measures whether the RF emulator is capable of predicting simulated *slc* with high accuracy given yet-
unseen instances of the inputs. A high predictive capability is achieved for a low *RAE* value.

Finally, the third set of criteria measures the changes in the emulator-based probabilistic projections, which are assessed
through a Monte-Carlo random sampling procedure. For fixed GSAT change values, the input variables are randomly sampled
by assuming a uniform discrete probability distribution for the categorical variables, and a uniform probability distribution for
the continuous variables except for $\kappa$ which is sampled as in (Edwards et al. 2021) from the smoothed version of the empirical
density function by Slater et al. (2019). The changes in the median and the endpoints of the 90% credibility interval (defined
by the percentile at 5 and 95%, denoted Q5% and Q95%) are then quantified.

## 3. Results

### 3.1 Emulator reference solution

We train a RF model to predict *slc* in 2100 using the results of the GrIS MME. A preliminary screening analysis was conducted
(detailed in Appendix B), and showed that four predictor variables have no significant influence: the choice to account for
thermodynamics, the choice in sliding law, the type of initialisation and the number of years for the initialisation phase. We
therefore build the RF emulator using only 7 out of 11 possible input variables described in Sect. 2.

To select values for the two main RF parameters, node size (*ns*) and the number of variables to randomly sample as candidates
at each split (*m*$_{\text{try}}$), we use a 10-fold cross validation exercise (Hastie et al., 2009) varying *ns* from 1 to 10, and *m*$_{\text{try}}$ from 1 to
7, and selecting the most optimal combination with respect to cross-validation predictive error. The number of random trees is
fixed at 1,000; preliminary tests having showed that this latter parameter has little influence provided that it is large enough.





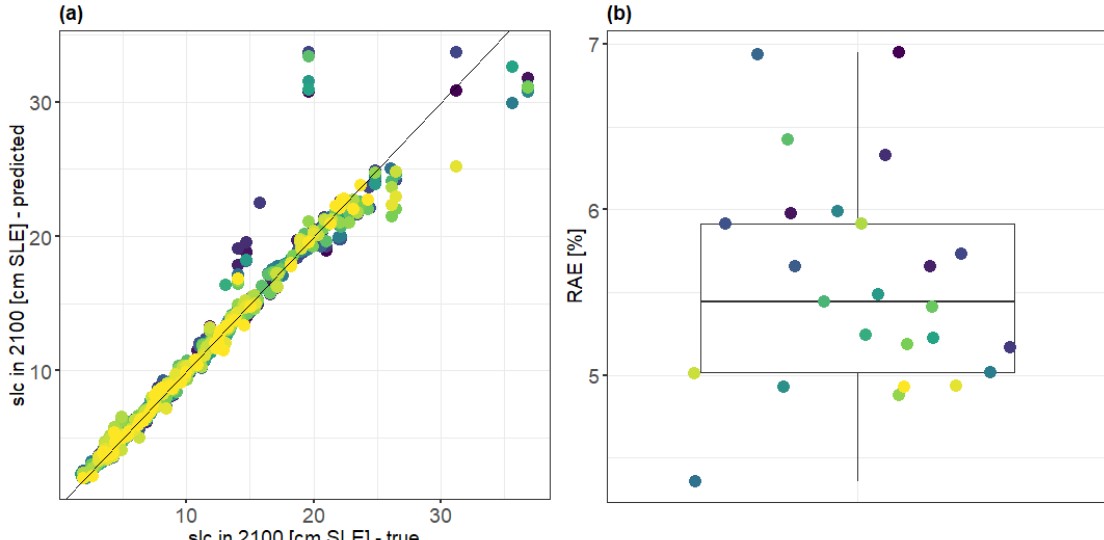

**Figure 4: (a) Comparison between the true numerically computed** *slc* **and the emulator's predicted values for the 25 validation tests (described in Sect. 2.4). Each colour indicates a different iteration of the testing; (b) Boxplot of the** *RAE* **performance indicator over the 25 validation tests. The lower** *RAE***, the higher the predictive capability.**


On this basis, we compute the reference solution for the criteria used to investigate the influence of the design questions. First, the changes in the MME size and distributions of the members resulted from random validation tests are assessed (see details in Sect. 3.2). Then, the RF model's predictive performance is tested by applying the 25 random validation tests, as described in Sect. 3.2. Figure 4a shows the comparison between the "true" numerically computed *slc* and the emulator's predicted values.

The dots align relatively well along the 1:1 line, which indicates a high predictive capability. This is confirmed by the performance indicator *RAE* which reaches very satisfactory values, with a median value well below 10% (Figure 4b). Finally, the probability distribution of *slc* (Figure 5) is constructed using the Monte-Carlo-based procedure (with 10,000 random samples) described in Sect. 2.4 given three GSAT change values fixed at 2°C, 3°C and 4°C. This results in a median value of respectively 4.7cm, 6.9cm and 10.5cm for *slc* with a 90% credibility interval of [3.7; 6.3cm], of [6.9; 8.7cm] and [8.5; 13.5cm].



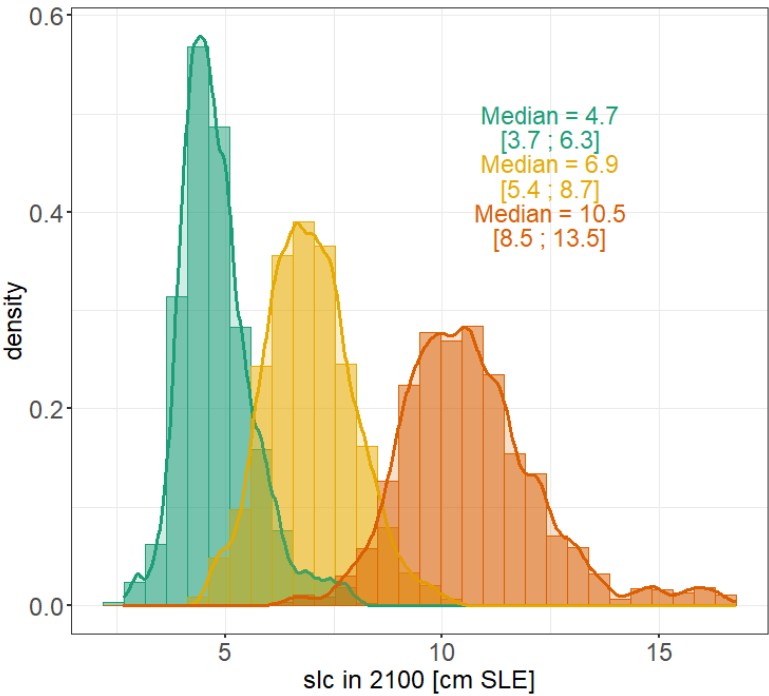


**Figure 5: Probability density function of *slc* in 2100 (with respect to 2015) constructed using a combined RF emulator - Monte-Carlo-based procedure (with 10,000 random samples, see Sect. 2.4) for three GSAT change values of 2°C (green), 3°C (yellow), and 4°C (orange). This results in a median value of respectively 4.7cm, 6.9cm and 10.5cm with a 90% credibility interval of [3.7; 6.3cm], of [6.9; 8.7cm] and [8.5; 13.5cm]. The straight line is calculated from a Gaussian kernel density estimation with a bandwidth chosen**
**by following Silverman (1986)'s 'rule of thumb'. The number and interval indicate the median value and the 90% credibility interval.**

**3.2 Impact of design decisions on the MME characteristics**

We first analyse in Figure 6 the impact of the design decisions in terms of MME size and distributions of the members as measured by the indicators defined in Sect. 2.4. As expected, the larger the MME size decrease (measured by $D_S$), the larger the perturbation of the histograms (measured by $D_h$). More interestingly, some experiments lead to different MME sizes,
namely experiments 'Med. & extr. Kappa' and 'CISM' with $D_S$ of respectively >55% and >35%, but with an approximately equivalent impact on the members' distributions, with $D_h$ on the order of 20%. For experiments 'woSSP585' and 'woCISM', this is the opposite with different $D_h$ values of respectively ~27% and ~37%, but with the same resulting MME size with $D_S > 65\%$. For comparison, we also show in blue the reference solution defined as the mean value assessed over the 25 iterations of the random validation exercise, described in Sect. 2.4, applied to the original dataset. In Sect. 3.3 and 3.4, we assess to
which extent these changes in the MME design translate in changes in the emulator's performance (prediction and probabilistic projections).



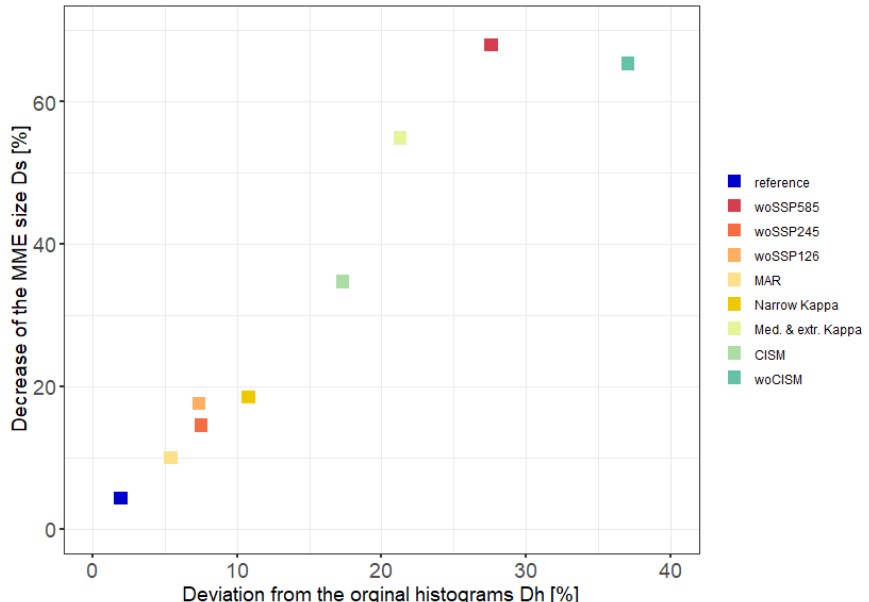

**Figure 6: Position of the emulator's experiment in a ($D_h$, $D_s$) diagram where $D_s$ measures the relative decrease in the MME size after applying the experiment, and $D_h$ measures the deviation of the histograms from the original ones (see Sect. 2.4). The blue-coloured marker refers to the reference solution defined as the mean value over the 25 iterations of the random validation exercise, described in Sect. 2.4, applied to the original dataset.**

### 3.3 Impact of design decisions on the emulator performance

We analyse in Figure 7 the impact of design decisions with respect to the RF predictive capability (measured by *RAE* defined in Sect. 2.4).

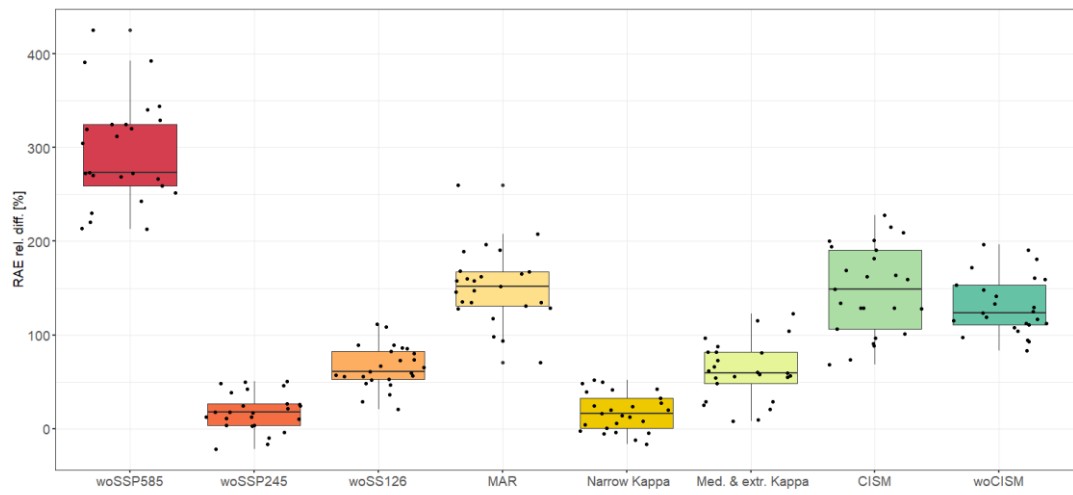

**Figure 7: Relative difference (in %) for the estimates of RF predictive capability measured by *RAE*, between the RF reference solution and the RF emulators trained by applying the experiment described in Table 2. The dots indicate the results of the 25 repetitions of random validation tests (described in Sect. 2.4).**




This shows that the design decisions regarding SSP scenarios has a strong impact depending on the particular scenarios that
are considered. Excluding the extreme SSP scenario SSP5-8.5 (experiment 'woSSP585') has the largest impact in terms of
*RAE* relative difference with respect to the original RF performance (Sect. 3.1), where *RAE* is increased of ~10% compared to
the original *RAE* value (Fig. 4). This result has a connection to the indicators described in Sect. 3.1. The 'woSSP585'
experiment, which excludes SSP5-8.5 members, removes the largest number of all members (see Fig. 6), i.e. of almost 70%,
which logically degrades the predictive capability since the RF is trained on a small dataset. Conversely, removing the
intermediate SSP scenario, experiment 'woSSP245', or restricting the analysis to intermediate $\kappa$ values (experiment 'Narrow
Kappa') have the lowest performance decrease (with a relative *RAE* difference on the order of 25%, since they result in low
$D_S$ values (see bottom left of Fig. 6).

However, this 'size effect' is not the only contributor to the performance impact, as shown by the 'woCISM' experiment,
which removes an equivalent number of members to the 'woSSP585' experiment (Fig. 6), and the resulting *RAE* increase
reaches half that of 'woSSP585' experiment. The experiment 'woCISM' has the largest impact on the member distributions
as indicated by a high $D_h$ value (see top right of Fig. 6). This shows that the second important factor here is the diversity among
the members within the MME after applying the experiment. The $D_h$ indicator remains, however, a first-order approximation
of this diversity, as underlined by the design decisions concerning the choices of models, ISM ('CISM' and 'woCISM'
experiments) and RCM ('MAR' experiment): they influence the predictive performance of the emulator in a similar way, with
a relative *RAE* difference of about half that of the exclusion of SSP5-8.5, but with different positions in the $D_S$-$D_h$ diagram.
The analysis of an alternative indicator of emulator's predictive capability in Supplementary materials S2 confirms these
results.

**3.4 Impact of design decisions on the emulator-based probabilistic projections**

We analyse here the impact on the RF-based probabilistic projections. Since the impact on the percentiles has here more
interest from the perspective of end-users, we primarily focus the analysis on the changes in the *slc* percentiles, Q5%, Q95%
and the median value in Fig. 8. The interested reader can refer to Supplementary materials S3 for an analysis of the whole *slc*
probability distributions' changes. Figure 8 shows that, depending on the GSAT change, the percentiles are perturbed in
different ways. Several observations can be made:

- Overall the design decision for $\kappa$ range has only a minor impact regardless of the GSAT change and the considered
percentile. This result is in agreement with the analysis on the RF predictive capability in Sect. 3.3;
- Restricting to a unique RCM, here MAR, has the largest impact on the median value for the lowest GSAT change
scenario, resulting in a very high over-estimation >50%. Its impact on the percentiles, Q5% and Q95%, remains
however low, which is consistent with the horizontal shift of the distribution shown in Supplementary Materials S3;
- The design decisions for ISM (experiment with CISM and without CISM) impact all percentiles regardless of the
GSAT changes;




- Excluding other ISMs than CISM (experiment 'CISM') has a high influence on the spread of the *slc* probability distribution measured by the Q95%-Q5% difference, which leads to a high overestimation up to 20% for GSAT≥3°C. This is consistent with the horizontal shift of the distribution as shown in Supplementary Materials S3;

- Excluding some particular SSP-RCP scenarios has an influence for GSAT change≥3°C. This results in a moderate under-estimation of about -15% when excluding the extreme (in terms of radiative forcing) SSP5-8.5 scenario (experiment 'woSSP585') for both Q5% and the median, and with an overestimation of >10% for the median when excluding the intermediate SSP2-4.5 scenario. Interestingly, 'woSSP585' experiment has here not the largest impact though it resulted in the largest decrease of the emulator's predictive performance (Fig. 6);

- The high percentile Q95% remains unchanged for GSAT change ≤3°C regardless of the emulator's experiment;

- The median value is impacted moderately (of the order of 10-15%) by different experiments provided that GSAT change≥3°C.

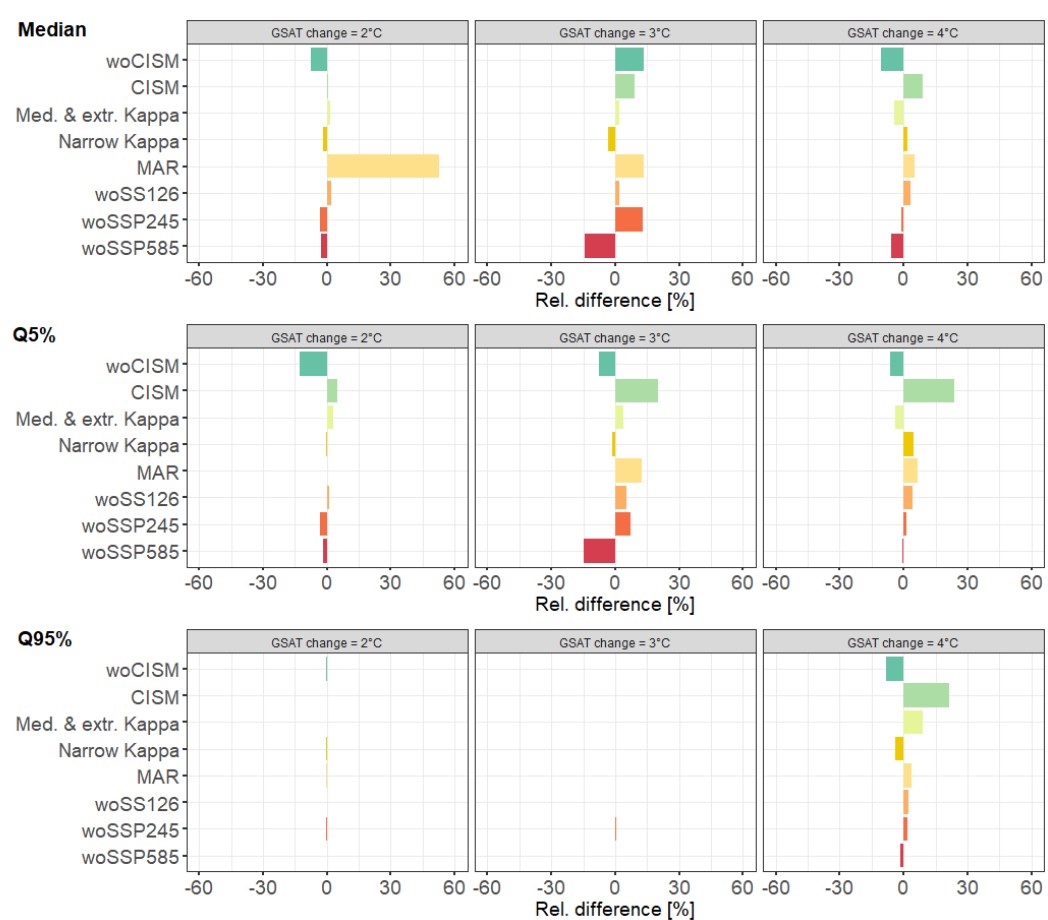

**Figure 8: Relative difference (in %) between the RF reference solution and the RF model trained when considering the experiments indicated in the y-axis (see Table 2 for full details) for the estimates of three *slc* percentiles, the median and the quantile at 5% (Q5%) and at 95% (Q95%), considering three GSAT changes, 2°C, 3°C and 4°C. The almost blank panels in the lower left and middle indicate that the relative differences are small and very low (<1%) when invisible.**





## 4. Synthesis and Discussion

### 4.1 Implications for MME design

Table 3 summarises the main results from the emulator's experiments for each design question. In the following, we take the
295 viewpoint of a MME designer, and derive the practical recommendations from these results.

**Table 3. Summary of the results from the emulator's experiments for each design question.**

| Input | Question | Results from the emulator's experiments |
|---|---|---|
| SSP-RCP | Does including a medium scenario SSP2-4.5 improve the results or is it enough to use the end members SSP1-2.6 and SSP5-8.5? | Excluding the medium scenario has a small-to-moderate impact. As expected, the inclusion of SSP5-8.5 is the necessary condition for achieving an emulator's high performance and accurate percentile estimates. |
| RCM choice | What is the added value of including new RCM, i.e. is it sufficient to focus on MAR regional climate model only? | It is the most impactful decision for the median value more particularly for low GSAT value. |
| ISM choice | What is the added value of accounting for all ISMs except for one? | Restricting the analysis to one unique ISM, here CISM, might lead to a clear over-estimation of the probabilistic projections. In addition, having different ISM is beneficial (experiment 'woCISM'), but not sufficient with possible under-estimation of the median values. This decision has the largest impact regarding the spread of the probabilistic projection (measured by the Q95-Q5% differences). |
| Range of $\kappa$ values | Should the design cover a large range of values, i.e. is it sufficient to focus on extreme values? | This decision is the least impactful relatively to the others. Results suggest that restricting to the '*Medium and Extreme*' scenario is sufficient. |

On the one hand, some conclusions were expected beforehand and more specifically for the SSP-RCP choice. The inclusion
300 of SSP5-8.5 appears to be a necessary condition for achieving a high performance emulator and accurate percentile estimates,
since this scenario contains many simulations and also covers a wide range of global warming levels. On the other hand,
conclusions on the three other design questions could not have been anticipated. They support, to some extent, a posteriori, the
choices that have been made for the construction of the MME considered here (based on that of Goelzer et al. (2020)). A very
practical implication can be derived from the $\kappa$ experiments: results indicate that restricting to the extreme and medium
305 scenario is sufficient here because of the lesser impact between the two experiments, '*Med. & Extr.*' or "*Narrow*". This result
is interpreted as being linked to a quasi-linear relationship between $\kappa$ and *slc*. This was confirmed by analysing the partial
dependence plot PDP of the RF emulator, which models the relationship between the input variable (here $\kappa$) of interest and the
response (here *slc*) while accounting for the average effect of the other input variables (see Friedman (2001) for technical



details). The high Pearson correlation derived from the PDP >90% confirms the evidence of quasi-linear behaviour. In practice,
this result implies that the number of scenarios explored in the MME can be limited to a three-scenario approach (low-medium-
high value), i.e. the number of members can be reduced, thus reducing the number of long numerical simulations required.
Results for RCM and ISM choice can be seen as an additional justification for intensifying the model intercomparison efforts
initiated in the past. Both modelling choices impact the results similarly, but on different aspects, either on the best estimate,
here measured by the median, or on the spread of the probabilistic projection, here measured by the Q95-Q5% differences. On
the one hand, restricting the analysis to one unique ISM, here CISM, might lead to a clear over-estimation of the probabilistic
projections, which means that having a diversity of ISMs is here beneficial. This is in line with the initiative originally launched
in ISMIP6 (Nowicki et al., 2016), which included coupled ISMs as well as stand-alone ISMs in CMIP for the first time. On
the other hand, restricting the MME to MAR only, has the most impactful decision for the median, more particularly for low
GSAT value. The importance of the RCM choice is in agreement with other studies (e.g., Wirths et al., 2024). We have also
completed this analysis with a 'woMAR' scenario, which showed a very large impact with absolute deviations >40% especially
for large GSAT values (not shown). Although it reinforces our conclusions, this result can only be considered qualitatively,
because MAR is so widely used in members (88% of the total number) that its removal from the original MME results in a too
small training dataset. This calls however for intensifying the cooperative research efforts, potentially within a MIP, by
extending this study to different RCM models, instead of MAR only, or investigating the relevance of using different versions
of MAR (see Table 2). This also relates to the question of initialisation (and initial mass loss estimates) where the RCM choice
is a key ingredient (e.g., Otosaka et al., 2023).

## 4.2 Implications from stakeholders' point of view

Our work can help stakeholders in several ways. First, our study contributes to a better understanding of the contribution of
Greenland ice sheet melt to sea level rise, which is estimated by the latest authoritative sea level projections developed by the
IPCC (Fox-Kemper et al., 2021) at 8 cm [4 cm; 13 cm] (median [likely range]) by 2100 for the SSP2-4.5 scenario, i.e.
Greenland has a non-negligible share in the total mean sea level rise, which was estimated at 56 cm [44 cm; 76 cm] for the
same SSP2-4.5 scenario. Second, our results support coastal adaptation practitioners in their decision-making. Our emulator's
experiments in Sect. 3.4 highlight how the different modelling choices affect differently the median or the upper tail (here
measured by the Q95% percentile). This difference is importance, because the literature on adaptation decision-making has
clearly shown that knowing the median is not sufficient for coastal adaptation practitioners managing long-living critical
infrastructures or making strategic decisions for regions or countries (Hinkel et al., 2019). These practitioners need credible
assessments of the uncertainties in ice mass losses in Greenland, including for the low probability scenarios corresponding to
the tail of probabilistic projection. Thus, our study supports the need for improved experimental designs by making some
practical recommendations, especially regarding the consideration of ISM, RCM and RCP8.5/SSP5-8.5 simulations.
Finally, the importance of SSP5-8.5, although expected, also underlines the fact that more extensive experiments on radiative
forcing values should be considered in the future. The SSP5.8-5 scenario contains many simulations and covers a wide range



of global warming levels including some simulations leading to 3°C GWL by 2100. In the current set of experiments that are available in the literature, this means that SSP5-8.5 may be extreme in terms of radiative forcing, but it is not in terms of temperature. To represent plausible outcomes of failure of states to meet their own commitments or political backlashes leading

to climate policy setbacks (see recent discussion by Meinshausen et al., 2024) assuming radiative forcing between 4.5 and 7.0W/m$^2$ in 2100 should be considered. This aspect is all the more important as another need is now emerging: projections of ice mass loss for specific levels of global warming (e.g. 3°C as recommended in the latest adaptation plan in France), as these can potentially be better understood by stakeholders than the SSP or RCP scenarios. Here again, the development of these projections requires emulators, whose accuracy and precision can be improved by better experimental design.

**5. Concluding remarks and further work**

Developing robust protocols to design balanced and complete numerical experiments for MME is a matter of active research that has called multiple studies either for sea level projections via selection criteria (Barthel et al., 2020) or from an uncertainty assessment's perspective (Aschwanden et al., 2021), and more generally for regional impact assessment (Merrifield et al., 2023; Evin et al., 2019). In this study, we take advantage of a large MME produced for Greenland ice sheet contributions to

future sea level to define a series of emulator's experiments that are closely related to practical MME design decisions. As expected, our results confirm the utmost importance of including the SSP5-8.5 scenario. Interestingly, our results also highlight the importance of having diverse ISM and RCM models. Finally, the less impactful choice is the one in the range of the Greenland tidewater glacier retreat parameter. These recommendations (detailed in Table 3) can be informative for the design of next generations of MME, in particular with the 7[th] Phase of the Coupled Model Intercomparison Project, and more

particularly ISMIP7 in preparation (Nowicki et al., 2023).

A first avenue of this study is to multiply the application of our procedure to additional MMEs of interest, in particular for Antarctica (Seroussi et al., 2020), for multi-millennial projections (e.g., Seroussi et al., 2024), and for glaciers (Marzeion et al., 2020). Though of interest to validate or highlight key design questions, our recommendations are derived, by construction, a posteriori, i.e., based on the available members of a large-size MME. Therefore, a second avenue here is to derive

recommendations a priori, i.e. during the construction of the MME design ideally in an iterative manner between the phase of MME construction and that of the emulator training. From a methodological perspective, robust tools may be found in the data valuation domain (Sim et al., 2022), which aims to study the worth of data in machine learning models based on similar methods as the ones used by Rohmer et al. (2022) in the context of sea level projections. Transposed to the MME context, these tools could be used in future studies to assess the impact of each member in the emulator's predictions, i.e. the worth of

each member. This type of result is expected to serve as guidance to the MME design in particular regarding the question of completeness and the necessity for balanced design.



**Author contributions**

JR and HG designed the concept. TE pre-processed the MME results. JR set up the methods and undertook the statistical analyses. JR and HG defined the protocol of experiments. JR, HG, TE, GLC, HG, GD analysed and interpreted the results. JR
wrote the manuscript draft. JR, HG, TE, GLC, GD reviewed and edited the manuscript.

**Competing interests**

The authors declare that they have no conflict of interest.

**Code/Data availability**

We provide the data and R scripts to run the experiments and analysing the results on the Github repository:
https://github.com/rohmerj/MMEdesign

**Acknowledgements**

We acknowledge the modelling work that constitutes the MME analysed in this study: the PROTECT Greenland ice sheet modelling and regional climate modelling groups, and the World Climate Research Programme and its Working Group on Coupled Modelling for coordinating and promoting CMIP5 and CMIP6. We thank the modelling groups for producing and
making available their model output and the Earth System Grid Federation (ESGF) for archiving the CMIP data and providing access. HG has received funding from the Research Council of Norway under project 324639 and had access to resources provided by Sigma2 - the National Infrastructure for High Performance Computing and Data Storage in Norway through projects NN8006K, NN8085K, NS8006K, NS8085K and NS5011K. This project has received funding from the European Union's Horizon 2020 Research and Innovation Programme under grant agreement No 869304, PROTECT. The authors would
like to acknowledge the assistance of DeepL (https://www.deepl.com/fr/translator) in refining the language and grammar of this manuscript.





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

**Appendix A Random Forest regression model**

Let us first denote $slc^{i=1,\dots,n}$ the $i^{th}$ value of sea level contribution calculated relative to the $i^{th}$ vector of p input parameters' values $\boldsymbol{x}^{i=1,\dots,n} = \{x_1, x_2, \dots, x_p\}^{i=1,\dots,n}$ where n is the total number of experiments. The Random Forest (RF) regression model

is a non-parametric technique based on a combination (ensemble) of tree predictors (using regression tree, Breiman et al. 1984). By nature, tree models can deal with mixed types of variables, categorical or continuous. Each tree in the ensemble (forest) is built based on the principle of recursive partitioning, which aims at finding an optimal partition of the input parameters' space by dividing it into L disjoint sets $R_1, \dots, R_L$ to have homogeneous $Y_i$ values in each set $R_{l=1,\dots,L}$ by minimizing a splitting criterion (for instance based on the sum of squared errors, see Breiman et al. 1984). The minimal number of

observations in each partition is termed nodesize (denoted $ns$).

The RF model, as introduced by Breiman (2001), aggregates the different regression trees as follows: (1) random bootstrap sample from the training data and randomly select $m_{try}$ variables at each split; (2) construct $n_{tree}$ trees T($\boldsymbol{\alpha}$), where $\boldsymbol{\alpha_t}$ denotes the parameter vector based on which the $t^{th}$ tree is built; (3) aggregate the results from the prediction of each single tree to estimate the conditional mean of $sl$ as:

$E(sl|\boldsymbol{X} = \mathbf{x}) = \sum_{j=1}^{n} w_j(\mathbf{x})sl^j,$                     (A1)

where E is the mathematical expectation, and the weights $w_j$ are defined as

$w_j(\mathbf{x}) = \frac{\sum_{t=1}^{n_{tree}} w_t(\mathbf{x}, \boldsymbol{\alpha_t})}{n_{tree}}$ with $w_j(\mathbf{x}, \boldsymbol{\alpha}) = \frac{I_{\{X_i \in R_{l(\mathbf{x},\alpha)}\}}}{\#\{j: X_i \in R_{l(\mathbf{x},\alpha)}\}},$          (A2)

where I($A$) is the indicator operator which equals 1 if $A$ is true, 0 otherwise; $R_{l(x,\alpha)}$ is the partition of the tree model with parameter $\boldsymbol{\alpha}$ which contains $\boldsymbol{x}$.





The RF hyperparameters considered in the study are *ns* and $m_{try}$ which have shown to have a large impact on the RF performance (Probst et al., 2019). The number of $n_{tree}$ was set up to a large value of 1,000 because of its smaller influence on the RF model performance (relative to *ns* and $m_{try}$).

**Appendix B Screening analysis**

We rely on the hypothesis testing of Altmann et al. (2010). To identify the significant predictor variables, the null hypothesis
"no association between *slc* and the corresponding predictor variable" is tested. The corresponding p-value is evaluated by (1) computing the probability distribution of the importance measure of each predictor variable through multiple replications (here 1,000) of permuting *slc*; (2) training a RF model; and (3) computing the permutation-based variable importance. When the p-value is below a given significance threshold (typically of 5%), it indicates that the null hypothesis should be rejected, i.e., the considered predictor variable has a significant influence on *slc*. Figure B1 shows that four predictor variables have non-
significant influence with p-values well above 5%, namely the choice in the account for thermodynamics, the choice in the sliding law, the type of initialisation and the number of years for initialisation phase.

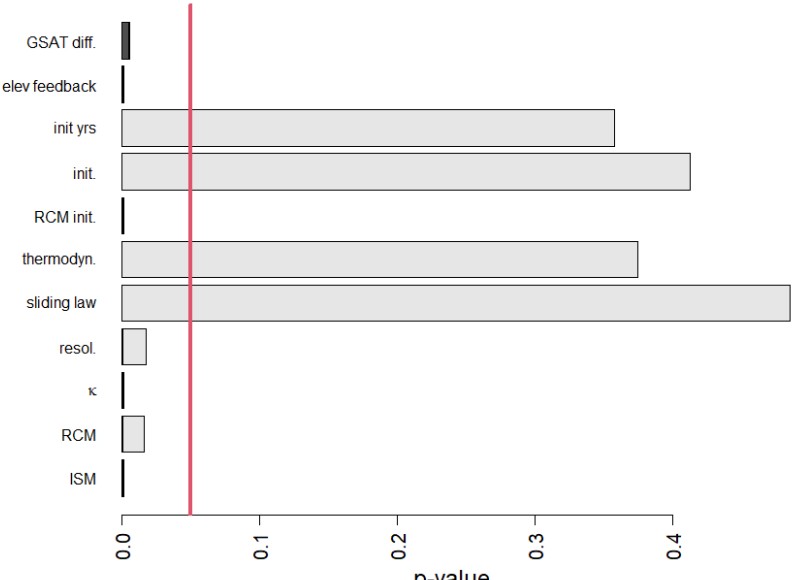

**Figure B1: Screening analysis showing the p-values of the RF variable importance-based test of independence of Altmann et al. (2010). The vertical red line indicates the significance threshold at 5%. When the p-value is below 5%, it indicates that the null hypothesis should be rejected, i.e., the considered variable has a significant influence, and should retained in the RF construction.**





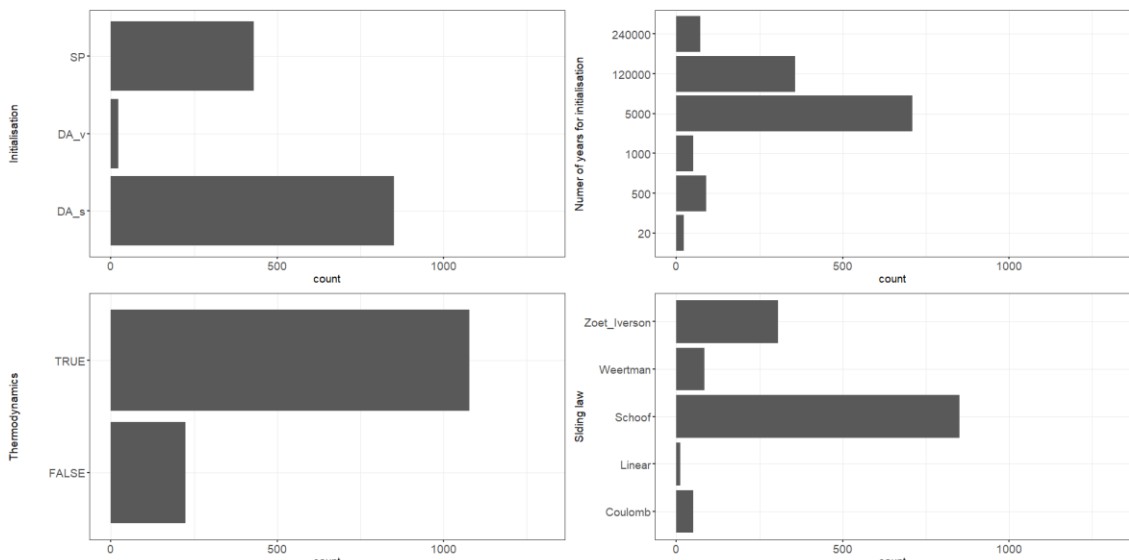

**Figure B2: Count number of the MME members with respect to the variables identified as non-influential.**