# Peer review of "Lessons for multi-model ensemble design drawn from emulator experiments: application to a large ensemble for 2100 sea level contributions of the Greenland ice sheet"

_EGUsphere, 2025_

## Referee Comment (RC1)

**Review of "Drawing lessons for multi-model ensemble design from emulator experiments: application to future sea level contribution of the Greenland ice sheet" by Rohmer et al.**

The paper *"Drawing lessons for multi-model ensemble design from emulator experiments: application to future sea level contribution of the Greenland ice sheet"* investigates the impact of different choices in constructing an emulator capable of predicting the future contribution of the Greenland Ice Sheet to sea-level rise. Specifically, the authors examine how the selection of models and parameters associated with different physical processes and scales (climate scenarios, regional climate models, ice-flow models, and ice-flow parameters) influences the emulator in terms of its fidelity, the estimated contribution to sea-level rise, and the associated uncertainty.

This study represents a valuable contribution to our understanding of multi-model ensemble approaches and emulator design. The numerical experiments are clearly described, and the results are noteworthy. Overall, the paper provides an interesting addition to the scientific literature. Nonetheless, I have a series of comments and questions that I would like the authors to address. On the one hand, in terms of the way the manuscript is written, some sections could be streamlined for conciseness (see my first general comment below, as well as specific comments). On the other hand, I have a series of more fundamental general comments on key aspects of the manuscript. There are three main comments, described herefater, which are followed by a series of specific comments addressing more detailed points. Once these concerns have been satisfactorily addressed, I will be happy to recommend the paper for publication in the Special Issue: *Improving the contribution of the land cryosphere to sea level rise projections*.

**General comment 1**

This paper lies at the intersection of two fields: glaciological modeling and statistical methods. Such interdisciplinary studies are particularly valuable, as the glaciology community may not be fully familiar with statistical techniques, while statisticians may not always be aware of the challenges involved in estimating future sea-level rise. Furthermore, comparative studies have gained increasing importance in glaciology, as they help assess the robustness of different modeling approaches. Given this, it is crucial to also investigate how these comparisons are constructed in the first place. In this context, the present study is highly relevant.

That being said, I believe the paper could be more explicit about its practical conclusions, specifically conclusions (1) and (2) mentioned in the abstract. I would also suggest highlighting the conclusions related to the (different) impact of the RCM/ISM choice in the abstract and in the introduction (see lines 313–315). Additionally, some technical details could be either removed or moved to an appendix or supplementary materials. The reasoning behind these suggestions is that most readers of this journal are likely geoscientists, who will primarily be interested in the study's results. By streamlining technical details, the paper's key findings –which are noteworthy– could be emphasized more effectively, improving its accessibility.

I also wonder whether subsections 2.4 and 3.2 could be simplified by focusing on just one of the two indicators ($D_S$ and $D_h$), e.g. $D_S$. While some variability has been observed in the results, these indicators are strongly correlated (see figure 6). Furthermore, from a practical point-of-view, what really matters is the number of runs to be made (i.e., of $D_S$). Similarly, figure captions could be streamlined by omitting details that may not be particularly useful (e.g., the exact method used for kernel density estimation; see specific comments below). The technical lines 196–219 could also be relegated to supplementary materials.

**General comment 2**

Regarding the paper's methodology, I have some questions about how the way ensembles are introduced. Specifically, lines 34–36 state: *"Each member of a MME should evenly span a representative and exhaustive set of plausible realizations of the combined sources of uncertainty, (...), equally represented by a single model run"*. This suggests that each member of the ensemble should have the same weight. However, I find this somewhat misleading, and I believe the authors could elaborate further on this choice. The assumption that all runs should have equal weight holds only if our current knowledge suggests they are equally probable. From a Bayesian perspective, this would correspond to assuming a uniform prior distribution. However, this assumption may not always be justified. For example:

(i) Runs from lower-resolution models might be considered less reliable than those from higher-resolution models as they might not capture relevant small-scale processes.

(ii) Some values of the uncertain parameters might be less probable if they lead to results that deviate significantly from current observations.

Formally, these concerns can be addressed by updating the weights of each run based on their likelihood given observational data (e.g., Aschwanden and Brinkerhoff, 2022; Nias et al., 2023).

I understand that the authors did not include such a step in their analysis, as their focus was on assessing the emulator's capabilities. However, it seems to me that this point should be discussed in the methodology or discussion section for two reasons. First, to clarify for the reader that the choice of equal probability stems from an assumption about our current knowledge and that alternative approaches are possible. Second, because weighting model runs based on observational constraints is an emerging direction in the field, and this should be discussed in the context of future ISMIP ensemble designs. This could also be mentioned as a perspective for future work, as it would be interesting to see whether the conclusions remain valid when runs are weighted as a result of a calibration.

**General comment 3**

My third general remark concerns the parameter $\kappa$ associated with the calving rate. It is unclear why this particular parameter was chosen over others. From reading the paper, the rationale behind this choice is not obvious—perhaps it is based on modeling considerations or supported by previous studies? If so, it seems to me that the authors should provide a stronger justification for including this specific parameter.

More fundamentally, I wonder whether comparing the effect of $\kappa$ to that of RCP scenarios or RCMs is entirely meaningful. This comparison contrasts the impact of a single parameter of an ice-flow model with that of an entire climate scenario or a regional climate model, which incorporate numerous physical parameters. Given this, it is perhaps unsurprising that the effect of $\kappa$ appears quite limited.

To illustrate this, one could consider a similar comparison in the opposite direction: assessing the impact of choosing a glaciological model of varying complexity (e.g., full Stokes, BP, or SIA) against a single parameter from a RCM. This would likely lead to the conclusion that the specific parameter from the RCM has a minimal influence. Therefore, I wonder whether including $\kappa$ as an isolated parameter in this study is fully justified. Could the authors maybe clarify its relevance within the broader context of the study's objectives?

**Specific Comments**

(1) [Line 12] 'projection and the quantification of its uncertainty' → 'projections and the quantification of their uncertainties'.

(2) [Lines 15, 17, and 65] You use 'experiments' for two distinct concepts: numerical simulations (e.g., line 16) and numerical tests (e.g., line 15). Consider using separate words to avoid any confusion.

(3) [Line 16] '(Regional Climate Model RCM, or Ice Sheet Model ISM)' → '(Regional Climate Model; RCM, or Ice Sheet Model; ISM)'.

(4) [Line 19] Consider removing 'utmost' as it might be overly formal.

(5) [Line 25] 'projection and the quantification of its uncertainty' → 'projections and the quantification of their uncertainties'.

(6) [Line 26] 'co-ordinated sets of numerical experiments' → 'sets of numerical experiments'.

(7) [Line 47] Consider adding references related to (machine-learning-based) emulators.

(8) [Line 47] I might be a bit picky here, but I would argue that the key advantage of statistical emulators is their low computational cost; being able to predict the model response at untried input values is only useful if it can be done at a reasonable cost.

(9) [Line 63] Please be consistent with your use of acronyms: either your define what you mean by RCM and GCM, or you use directly the corresponding acronyms. Also, a table of acronyms would be useful in the paper.

(10) [Line 63] Avoid using 'validation tests' as this can lead to confusion when it comes to glaciological modeling (for which 'validation' has another meaning).

(11) [Line 76] '(Goelzer et al. (2020): in particular (...))' → '(Goelzer et al., 2020; in particular (...))'.

(12) [Line 79] Consider adding a schematic displaying the modeling chain and indicating where modeling choices (MME inputs) are introduced. This could be very useful to effectively obtain an overview of the context.

(13) [Line 80] Here you define again what a RCM is. If you have already defined it before that is not necessary.

(14) [Table 1] Ensure consistent formatting of symbols (italics vs. non-italics).

(15) [Table 1] Consider renaming 'Symbol' to 'Symbol/Acronym' or simply 'Name' to clarify that most entries are acronyms.

(16) [Table 1] 'Sliding basal law' → 'Sliding law' or 'Basal friction law'.

(17) [Line 101] Consider defining 'input setting' explicitly, e.g. as a particular combination of inputs.

(18) [Line 102] 'Minimal resolution' is ambiguous. If referring to spatial resolution, clarify: 'e.g., for $\kappa$, or for the minimum spatial resolution'.

(19) [Figure 1] Consider changing the $y$-label to 'Probability distribution', 'Probability distribution function', or 'PDF'.

(20) [Figure 1] 'Greenland ice-sheet' $\rightarrow$ 'Greenland ice sheet'.

(21) [Figure 1] 'raw MME' $\rightarrow$ 'original MME'.

(22) [Figure 1] Consider simplifying the caption by just stating that the black line is based on kernel density estimation, removing extraneous details.

(23) [Line 114] 'have the highest importance' is not clear; consider replacing it with something more precise, e.g. 'contributes to most of the uncertainty of the slc'.

(24) [Line 118] Replace 'distributions' to avoid confusion with PDFs.

(25) [Figure 2] The orientation of the plots is a bit unusual. Consider switching the $x$ and $y$ axes, with the count number in the $y$ direction.

(26) [Figure 2] 'to the different inputs' $\rightarrow$ 'to different inputs'.

(27) [Figure 3] The orientation of the plots is a bit unusual. Consider switching the $x$ and $y$ axes, with the count number in the $y$ direction.

(28) [Figure 3] 'to the different inputs' $\rightarrow$ 'to different inputs'.

(29) [Line 134] It seems that this study only concerns sea-level contribution at 2100. If so, avoid referring to 'a given time $t$'.

(30) [Table 2] Note that your question regarding the SSP-RCP scenario only corresponds to the 'woSSP245' case. Maybe make it a bit more general by asking whether removing one of the scenarios (SSP1-2.6, SSP2-4.5, SSP5-8.5) actually leads to an improvement of the results.

(31) [Table 2] Is there a reason why you did not consider a 'woMAR' case (by contrast with the 'CISM'/'woCISM' cases for the ISM)?

(32) [Line 175] Avoid using 'validation test' as this can lead to confusion when it comes to glaciological modeling (for which 'validation' has another meaning).

(33) [Line 180] I wonder if it would not be simpler to represent the probability distribution of the normalized error $|e/\text{slc}|$ by treating the input setting as an uncertain parameter (i.e. by considering the full sample space of the original MME). By doing so, you would avoid depending on additional numerical parameters ($n_{\text{test}} = 55$ and 25 test samples) for your statistics.

(34) [Figure 5] Consider changing the $y$-label to 'Probability distribution', 'Probability distribution function', or 'PDF'.

(35) [Figure 5] Consider simplifying the caption by removing details about density estimation. Stating that the PDF was estimated using a Monte-Carlo procedure does not seem to be a relevant information here.

(36) [Line 249] Avoid using 'logically'. If the rejected members are not representative of the input space, their removal may not necessarily improve the predictability of the random-forest surrogate.

(37) [Line 262] I wonder whether a table that contains for each test both the median RAE and the value of $D_S$ could be an efficient way to summarize these results. You could then obtain a measure associated with their ratios, i.e., that corresponds to the decrease in the number of ensemble members ($D_S$) and the 'cost' to pay (the RAE).

(38) [Table 3] Note that your question regarding the SSP-RCP scenario only corresponds to the 'woSSP245' case. Maybe make it a bit more general by asking whether removing one of the scenarios (SSP1-2.6, SSP2-4.5, SSP5-8.5) actually leads to an improvement of the results.

(39) [Line 311] 'long numerical simulations' → ' numerical simulations' as you never discussed the length of numerical simulations previously.

(40) [Lines 328–332] Consider removing this sentence as it is not a novelty of this study.

(41) [Line 342] Did you previously introduce the acronym 'GWL'?

(42) [Line 356] Consider removing 'utmost' as it might be overly formal.

(43) [Figure B1] The orientation of the plot is a bit unusual. Consider switching the $x$ and $y$ axes, with the $p$-value in the $y$ direction.

**References**

Aschwanden, A. and Brinkerhoff, D. J. (2022). Calibrated Mass Loss Predictions for the Greenland Ice Sheet. *Geophysical Research Letters*, 49(19).

Nias, I. J., Nowicki, S., Felikson, D., and Loomis, B. (2023). Modeling the Greenland Ice Sheet's Committed Contribution to Sea Level During the 21st Century. *Journal of Geophysical Research: Earth Surface*, 128(2).

---

## Referee Comment (RC2)

Review of "Drawing lessons for multi-model ensemble design from emulator experiments: application to future sea level contribution of the Greenland ice sheet" by Rohmer et al.
Reviewer: Vincent Verjans

This study develops a Random Forest (RF) emulator to emulate Greenland 2100 sea level contribution (slc) output from a Multi-model ensemble (MME). In particular, the RF is trained using a set of 7 inputs, associated with the climate scenario, the ice sheet model (ISM) used, the regional climate model (RCM) used, and different settings of the ISM run. The authors investigate how changing the MME design leads to changes in the emulator performance and in its range of emulated slc. Based on these metrics, they provide guidelines for future MME designs that aim at estimating future ranges of slc from the Greenland ice sheet.

This study addresses an important and difficult question: how can we improve the design of MMEs to provide the best information about the probability density function (PDF) of future slc? The concept underlying this study is that the MME itself does not need to characterize this PDF, but that it should be designed optimally such that an emulator can do this characterization a posteriori. This is a valid and efficient approach to uncertainty quantification. It is also a challenging topic, and work on this topic is important. However, at this stage, I believe that several points need to be improved to make this study a valuable contribution in addressing this question. The authors make recommendations and they "expect these recommendations to be informative for the design of next generations of MME" (L22). But I believe that their recommendations are dependent on many assumptions or choices that they made, without always justifying them and making them clear to the reader. Furthermore, more methodological details about the RF emulator are needed because all the results presented depend on the emulation and, therefore, the RF design influences strongly any interpretation, and thorough RF evaluation is critical as well. I detail my concerns in this review, which consists of Major and Minor comments. I do not provide technical comments at this stage of the reviewing process because I believe that the more substantial aspects should be addressed first. Line numbers in this review correspond to the preprint manuscript.

**Major comment 1: Inherent assumptions associated with the MME**
Many of the conclusions are strongly dependent on the particular MME used in this study. I have several reservations about this.

First, it is unclear to me how the MME used in this study was acquired and designed. The only details provided about the MME are (L74):
"We focus on the sea level contribution from the Greenland ice sheet (GrIS) in 2100 based on a new MME study performed for the European Union's Horizon 2020 project PROTECT (http://protect-slr.eu). Some modelling choices are taken from the protocols of the ISMIP6 initiative (Goelzer et al. (2020): in particular, the two main emissions scenarios, and the main model parameter explored." Has this MME been peer-reviewed? Why are the authors not using the well-established ISMIP6 MME? The latter MME also has the advantage of providing a larger set of experiments, notably including many more ISMs than the MME of this study. At least, why has the MME not been combined with the ISMIP6 MME? Also, given that no publication describing the MME is referenced, I believe that it is important to give many more details about the MME configuration: Did all ISMs run under high- and low-warming forcing? Are the 15 global climate models used in the MME well-balanced across the runs? Etc.

Second, the conclusions of this study are strongly dependent on the initial MME used in the emulation process. For example, the authors argue that there is "a quasi-linear relationship between $\kappa$ and slc" (L306). But this conclusion is based only on the set of 4 ISMs used in this study: CISM, IMAUICE, GISM, and ElmerIce. Furthermore, given that very little experiments were performed by ElmerIce and GISM, I assume (I need to assume here because no information is provided on the design of the MME) that these two models may well have only been run with a single $\kappa$ value. In this case, the "quasi-linear relationship" would be derived only from two ISMs. Given that different ISMs can show very different sensitivities to movement of the tidewater glacier front and grounding line positions, this conclusion could well be very different if other ISMs are included. So, if one was to perform a similar study with the ISMIP6 MME, would the "recommendations" for future MME design be different? As another example, I mention above that only 4 ISMs are included in the MME, two of which account for $> 90\%$ of the simulations. By excluding CISM from the training experiments, the authors then make "recommendations" about the ability of the emulator to estimate the slc simulated by ISMs not included in the design. Here also, this evaluation depends critically on how similar simulated slc from CISM is to the simulated slc from IMAUICE. This similarity depends on numerous aspects that are specific to these two paticular models. I would expect that the "recommendations" would be very different if other ISMs (ISSM, PISM ...) show more or less similarity with CISM.

Such assumptions are not made explicit by the authors. This could very well be misleading to the readership targeted by the authors, especially those less familiar with ice sheet modeling (e.g., "stakeholders" (L328) and "coastal adaptation practitioners" (L332)).

**Major comment 2: Characterization of uncertainty**
The authors use their random forest (RF) emulator such that "changes in the emulator's predictive performance and the emulator-based probabilistic projections provided information on several aspects" (L18). After reviewing the manuscript, I identify remaining limitations about the RF emulator regarding uncertainty characterization.

The authors use changes in the predictive performance of the RF as a proxy for uncertainty remaining about a hypothetical MME (here, a MME excluding some of the experiments). But this metric is sensitive to the particular machine learning model used for the emulation. Here, the emulation output is thus conditioned on the RF architecture, with a single fixed combination of hyperparameters. Is any decrease in predictive performance of this specific RF therefore a meaningful assessment of uncertainty imputable to the MME design? This question is critical, because the conclusions of this study use this as a fundamental assumption.

This issue is further exacerbated by the fact that the RF does not provide probabilistic output. By this, I mean that the RF only provides a point estimate. There is no uncertainty quantification. Ideally, the design of a MME should target the strongest reduction in posterior covariance (i.e., the uncertainty remaining given the current MME). But this particular RF emulator does not provide such metric. This could be addressed by choosing another architecture (e.g., Gaussian processes, Williams and Rasmussen (2006)), by subsampling techniques for RF models (Mentch and Hooker, 2016), or by adapting the RF to output conditional quantiles (Meinshausen and Ridgeway, 2006).

**Minor comment 1: Lack of technical information**
All the results and conclusions from the study are dependent on the RF emulator. As such, I find that more information on the RF development and evaluation are needed. I highlight some aspects

to prioritize here below.

(a) The evaluation of the RF (L174-183) is assessed through a random sampling evaluation, but I find the details about the evaluation method somewhat unclear. First, the authors mention the "iteration of the procedure" (L180). However, it is not explained what is iterative in this procedure. Later in the manuscript, the authors often refer to "25 validation tests" (e.g., caption of Figure 4). But this number of 25 is not explained in the description of the evaluation method. Thus, I can only assume that the random validation is iterated 25 times. Second, it is unclear what the validation performance measure shown in Figure 4 represents. In Figure 4a, there are clearly much more than 25 points, but clearly less than $25 \times 55 = 1375$ points (where 55 is the number of test samples mentioned on L180). Thus, what does each point represent? In addition, why are there much less points shown in Figure 4b than Figure 4a? Finally, the authors explain that there are 55 test samples, but they draw 5 samples for 10% ranges between 0 and 100% (L179-180). As such, there should be $5 \times 10 = 50$ test samples I believe, not 55.

(b) I wonder why it was decided to use this random evaluation procedure. In particular, the commonly-used 10-fold cross validation procedure would have been a more natural choice. This would also avoid the influence of sampling biases related with the random sampling of relatively few experiments (55 from the 1303 experiments per iteration). Since 10-fold cross validation was used for parameter fitting (L197), I suppose that there is no computational issue for this. Also, it would be straightforward to exclude the members from the 9 training folds as required by the specific experiments (e.g., exclude all SSP5-8.5 when training for woSSP585). Thus, is there any reason to prefer the random evaluation over the 10-fold cross validation?

(c) More technical details about the RF emulator construction would be beneficial. In particular, mixing categorical and continuous inputs is not straightforward, and may incur performance sensitivity to the RF design. For example, what is the splitting criterion used: mean absolute error, mean squared error, other? And how did the authors alleviate the potential issue of selection bias towards the inputs that have more possible splits? This could partly influence the different sensitivities to, for example, SSP5-8.5 scenario (global annual mean surface air temperature change, GSAT, is a continuous input with many different values), ISM (categorical input), $\kappa$ (continuous input with few different values). As such, some information on these technical aspects would help the reader understand how modeling challenges may affect the results or not.

**Minor comment 2: Use of global mean temperature change**
The authors aggregate all the combinations of emission scenario (SSP) and global climate model (GCM) as a value of GSAT. I wonder if this does not risk misrepresenting the climate forcing affecting the Greenland ice sheet (GrIS). In particular, a given GSAT could very well lead to different magnitudes of:
(1) GrIS surface air temperature change
(2) GrIS precipitation
(3) GrIS ocean forcing
I expect that there may well be some substantial differences in these 3 components between different GCMs. It would be interesting to explore whether separating the single GSAT variable into these 3 separate components refines the emulator predictions.

**Minor comment 3: Interpretation of some results**
I find that the interpretation of results are not always well supported quantitatively. I note that,

in some cases, this may simply be due to a lack of clarity in the interpretation. I provide here a few examples.

2.1 The $D_h$, $D_S$ definition

In Figure 6, the authors show the different combinations of decrease in MME size ($D_S$) and deviations from original histograms ($D_h$) resulting from their model experiments. Firstly, I think that the manuscript would benefit from a clearer definition of $D_h$. It is defined as "the average difference in the count numbers between the two histograms (normalised by the total number of members)" (L172-173). I believe that the normalization is by the histogram counts, not the total number of members, because otherwise $D_h$ would be proportional to $D_S$. For example, assume that for a given variable, we have a hypothetical 3-category histogram with counts 5, 10, 85 (i.e., n=100). In hypothetical experiment 1, the counts are 0, 10, 85 (i.e., n=95). In this case, $D_S = \frac{100-5}{100} = 0.95$ and, following the definition, $D_h = \frac{5+0+0}{3} \times \frac{1}{100} = \frac{1}{60}$. In hypothetical experiment 2, the counts are 5, 10, 80 (i.e., n=95). In this case, $D_S = \frac{100-5}{100} = 0.95$ and $D_h = \frac{0+0+5}{3} \times \frac{1}{100} = \frac{1}{60}$. This shows that taking "the average difference in the count numbers between the two histograms (normalised by the total number of members)" results in an identical pair ($D_S, D_h$) for these two hypothetical experiments. I am probably misunderstanding here, but I think that a more precise definition would help.

2.2 The $D_h$, $D_S$ results

I do not understand the interpretation of the impact from $D_h$, $D_S$ on the emulator performance (Sect. 3.3). First, the authors write "Excluding the extreme SSP scenario SSP5-8.5 (experiment 'woSSP585') has the largest impact in terms of $RAE$ relative difference with respect to the original RF performance (Sect. 3.1), where $RAE$ is increased of $\sim 10\%$ compared to the original $RAE$ value (Fig. 4)" (L245). However, Figure 7 shows a $\sim 275\%$ relative difference in $RAE$, so it is not clear to me where the value "$\sim 10\%$" comes from. Second, I do not follow the logic of the arguments. The authors write that (i) the high $D_S$ of woSSP585 causes large errors. But then, (ii) they argue that "this 'size effect' is not the only contributor to the performance impact, as shown by the 'woCISM' experiment, which removes an equivalent number of members to the 'woSSP585' experiment (Fig. 6), and the resulting $RAE$ increase reaches half that of 'woSSP585' experiment" (L253). And (iii) that the woCISM experiment has the largest $D_h$ value. However, when I interpret Figures 6 and 7, I find that (a) woSSP585 and woCISM have similar $D_S$ values (i.e., (ii)), (b) woCISM has higher $D_h$ than woSSP585 (i.e., (iii)), but (c) that the errors from woSSP585 ar much higher than those of woCISM (Figure 7). So, it seems that the lower $D_h$ of woSSP585 is accompanied by larger errors. This is the opposite message to that conveyed in the text: "This shows that the second important factor here is the diversity among the members within the MME after applying the experiment. The $D_h$ indicator remains, however, a first-order approximation of this diversity (...)" (L256). The statement of greater diversity leading to lower errors, is not supported by the larger errors of woSSP585 compared to woCISM. To summarize: $D_S(\text{woSSP585}) \approx D_S(\text{woCISM})$, $D_h(\text{woSSP585}) < D_h(\text{woCISM})$ where low $D_h$ implies greater "diversity", but $RAE(\text{woSSP585}) >> RAE(\text{woCISM})$.

2.3 Figure S3 (in Section S2)

The authors write "The analysis of an alternative indicator of emulator's predictive capability in Supplementary materials S2 confirms these results" (L261). However, in my view, Figure 6 ($RAE$ results) and Figure S3 ($Q^2$, coefficient of determination results) show contrasting conclusions. For example, $RAE$ of woCISM, CISM, and MAR are comparable (Figure 6). However, $Q^2$ is clearly lower for woCISM than for MAR and CISM (Figure S3). This indicates differences when evaluating relative errors versus explained variance. Thus, these differences are potentially interesting

to analyze, instead of being discarded as is done in the main text. In particular, they could relate to the emulator performance sensitivity to high versus low slc (the latter being more influential on relative metrics), or its sensitivity in the ability to predict values away from the mean value, or other aspects that would require investigation. Note that this links back to my general comment about the importance of understanding the RF emulator, because the interpretation of the results depends strongly on this understanding.

2.4 Figure 8

There are many aspects that I find puzzling or questionable in Figure 8. Firstly, the results do not correspond to what is shown in Figure S4, where the Q5% and Q95% are shown with the black error bars. For example, in the column $\Delta$GSAT=+3°, Q95% of woCISM, woSSP245, and woSSP585 are clearly strongly different from the Q95% labeled "original" (Figure S4). But Figure 8 shows that these differences are $\leq 1\%$. I believe that there is an inconsistency here, or something that I misunderstand about Figure 8.

Secondly, I do no understand how it is possible that the changes in median and quantiles at $\Delta$GSAT=+4° are so small for woSSP585. In this design experiment, the RF model has presumably not even seen such levels of warming during training because the SSP 5-8.5 scenario has been excluded. But, by definition, tree models (including RF) predict slc based on decision rules seen during training. Thus, it is not clear how the RF can predict relatively similar slc values under $\Delta$GSAT=+4° when excluding SSP 5-8.5 as when it is not excluded. I am probably misunderstanding something here, but I believe that the authors should explain this counter-intuitive aspect of their results.

**Minor comment 4: Some conclusions need to be put into perspective**
For different aspects, I find that better communication and/or more context about the conclusions is needed. I highlight some key examples here.

(a) Concerning $\kappa$, the authors argue for "the lesser importance of the choice in the range of the Greenland tidewater glacier retreat parameter" (L21). However, they compare it with the influence of the SSP scenario and of the ISM choice. It is expected that a single parameter should have much less influence than a global warming scenario and than a full ice sheet model.

(b) It should be better emphasized that the probabilistic ranges shown by the authors are not probabilistic projections of Greenland slc. Instead, they show a range of emulator predictions (thus conditioned on the emulator architecture) assuming a uniform distribution over the different inputs (L186). Thus, it does not represent calibrated uncertainty accounting for model-observation misfits (e.g., Aschwanden and Brinkerhoff, 2022). And neither does it represent the slc PDF from the MME, because the uniform distribution over the input space is not representative of the MME itself (e.g., the minimum spatial resolution is clearly not uniform between 1 and 40 km, see Fig. 3). As such, I believe that the true meaning of the PDFs shown in Figure 5 should be explained explicitly in order to avoid any reader misinterpreting those PDFs.

(c) The authors make a conclusion on "the utmost importance of including the SSP5-8.5 scenario, due to the large number of simulations available and the range of global warming they cover" (L19-20). However, I do not think that the authors have proven the co-existence of these two points. For example, could it be that including only a few training simulations with high global warming forcing would be sufficient to drastically decrease the errors of woSSP585 shown in Figure7? In other words, maybe the emulator needs only a few high-warming training examples to correctly

interpolate in the existing range of warming scenarios. Or maybe, as the authors write (L19-20), it is also the high number of experiments that is important. However, as far as I understand, the results presented in this study do not allow to evaluate the relative importance of these two aspects.

**References**

Andy Aschwanden and DJ Brinkerhoff. Calibrated mass loss predictions for the greenland ice sheet. *Geophysical Research Letters*, 49(19):e2022GL099058, 2022.

Nicolai Meinshausen and Greg Ridgeway. Quantile regression forests. *Journal of machine learning research*, 7(6), 2006.

Lucas Mentch and Giles Hooker. Quantifying uncertainty in random forests via confidence intervals and hypothesis tests. *Journal of Machine Learning Research*, 17(26):1–41, 2016.

Christopher KI Williams and Carl Edward Rasmussen. *Gaussian processes for machine learning*, volume 2. MIT press Cambridge, MA, 2006.

---

## Author Comment (AC2)

**Replies to Referee #2's comments on "Drawing lessons for multi-model ensemble design from emulator experiments: application to future sea level contribution of the Greenland ice sheet" (egusphere-2025-52)**

We would like to thank Referee #2 for the constructive comments. We agree with most of the suggestions and, therefore, we will modify the manuscript to take on board the comments and suggestions. We recall the reviews and we reply to each of the comments in turn (outlined in blue). Following the journal's reviewing procedure, the revised manuscript will be provided after the interactive review process, in a second phase.

**Additional changes**
Since the submission of this manuscript, we had the opportunity to include HIRHAM in the MME as well. In addition to the modifications suggested, we will thus include this third RCM model in the analysis. A major advantage will be to define an experiment where only the members of HIRHAM and RACMO are used ('woMAR' experiment) in addition to the 'MAR only' experiment. We expect that this modification will bring new insights and strengthen our conclusions which are, in the original version of the manuscript, based on two RCMs only. The results discussed in Sect. 4 will thus be modified accordingly.

**Referee #2:**

*This study develops a Random Forest (RF) emulator to emulate Greenland 2100 sea level contribution (slc) output from a Multi-model ensemble (MME). In particular, the RF is trained using a set of 7 inputs, associated with the climate scenario, the ice sheet model (ISM) used, the regional climate model (RCM) used, and different settings of the ISM run. The authors investigate how changing the MME design leads to changes in the emulator performance and in its range of emulated slc. Based on these metrics, they provide guidelines for future MME designs that aim at estimating future ranges of slc from the Greenland ice sheet.*

*This study addresses an important and difficult question: how can we improve the design of MMEs to provide the best information about the probability density function (PDF) of future slc? The concept underlying this study is that the MME itself does not need to characterize this PDF, but that it should be designed optimally such that an emulator can do this characterization a posteriori. This is a valid and efficient approach to uncertainty quantification. It is also a challenging topic, and work on this topic is important. However, at this stage, I believe that several points need to be improved to make this study a valuable contribution in addressing this question. The authors make recommendations and they "expect these recommendations to be informative for the design of next generations of MME" (L22). But I believe that their recommendations are dependent on many assumptions or choices that they made, without always justifying them and making them clear to the reader. Furthermore, more methodological details about the RF emulator are needed because all the results presented depend on the emulation and, therefore, the RF design influences strongly any interpretation, and thorough RF evaluation is critical as well. I detail my concerns in this review, which consists of Major and Minor comments. I do not provide technical comments at*

*this stage of the reviewing process because I believe that the more substantial aspects should be addressed first. Line numbers in this review correspond to the preprint manuscript.*

We thank Referee #2 for the in-depth analysis of the manuscript. In the following, we provide details on how we will account for them in the revised manuscript.

**Major comment 1: Inherent assumptions associated with the MME**

*Many of the conclusions are strongly dependent on the particular MME used in this study. I have several reservations about this.*

*First, it is unclear to me how the MME used in this study was acquired and designed. The only details provided about the MME are (L74): "We focus on the sea level contribution from the Greenland ice sheet (GrIS) in 2100 based on a new MME study performed for the European Union's Horizon 2020 project PROTECT (http://protectslr.eu). Some modelling choices are taken from the protocols of the ISMIP6 initiative (Goelzer et al. (2020): in particular, the two main emissions scenarios, and the main model parameter explored."*

*Has this MME been peer-reviewed? Why are the authors not using the well-established ISMIP6 MME? The latter MME also has the advantage of providing a larger set of experiments, notably including many more ISMs than the MME of this study. At least, why has the MME not been combined with the ISMIP6 MME? Also, given that no publication describing the MME is referenced, I believe that it is important to give many more details about the MME configuration: Did all ISMs run under high- and low-warming forcing? Are the 15 global climate models used in the MME well-balanced across the runs? Etc.*

The MME has been peer reviewed within the H2020 Protect project and is currently in preparation for submission to the Cryosphere Special Issue journal. In the revised version of the manuscript, we will refer to the EGUSPHERE preprint for further details. In addition, we will also provide a more detailed description of the Protect MME by paying particular attention to highlight the key differences with ISMIP6 MME.

To specifically reply to Referee #2 comment, we confirm that the experimental design builds on the ISMIP6 protocol with four different ice sheet models and extends it to more fully account for uncertainties in sea-level projections for the GrIS. This is underlined in Sect. 2. The Protect MME is thought as an extension of ISMIP6 MME regarding different aspects:
  - we have included a wider range of CMIP6 climate model output, more climate change scenarios (SSP126, SSP245, SSP585);
  - we have provided retreat forcing before 2015 that is calculated from reconstructions of past runoff and ocean thermal forcing. This allows for a consistent forcing of the models in past and future and to consider historical retreat of the outlet glaciers, which was an important source of mass loss after 1990;
  - we have provided surface mass balance forcing from several RCMs, i.e. MAR and RACMO for the MME used for this study.

Since the submission of this manuscript, we had the opportunity to include HIRHAM as well. We believe that adding this third RCM model will add new insights and will make our conclusions more robust (based on two RCMs in the original version of the study). The results discussed in Sect. 4 will thus be modified.

To appreciate this extension, the following table gives the repartition of the members with respect to the RCP/SSP scenarios for ISMIP6 MME and for Protect MME (before inclusion of

HIRHAM). In addition, it should be noted that the total number of members has increased by a factor of 4 compared to ISMIP6 MME.

|  | ISMIP6 MME | Protect MME |
|---|---|---|
| RCP26 | 23 | 40 |
| RCP85 | 156 | 319 |
| SSP126 | 18 | 189 |
| SSP245 | 0 | 189 |
| SSP585 | 59 | 566 |
| Total number | 256 | 1303 |

As a second illustration, the following figure gives an overview of the results of the ensemble of projections at the year 2100 for all available Earth System models (ESMs), RCMs and ISMs under high, med and low retreat sensitivity. This allows to appreciate, graphically, how well the range of sea level is covered. It should however be underlined that, although the collection of forcing data covers a wide range of variations across different ESMs and scenarios, it ultimately still represents an 'ensemble of opportunity' similarly as for ISMIP6 MME. Our study aims to address the potential implications of this characteristic.

[Figure]

*Second, the conclusions of this study are strongly dependent on the initial MME used in the emulation process. For example, the authors argue that there is "a quasi-linear relationship between κ and slc" (L306). But this conclusion is based only on the set of 4 ISMs used in this study: CISM, IMAUICE, GISM, and ElmerIce. Furthermore, given that very little experiments were performed by ElmerIce and GISM, I assume (I need to assume here because no information is provided on the design of the MME) that these two models may well have only been run with a single κ value. In this case, the "quasi-linear relationship" would be derived only from two ISMs. Given that different ISMs can show very different sensitivities to movement of the tidewater glacier front and grounding line positions, this conclusion could well be very different if other ISMs are included. So, if one was to perform a similar study with the ISMIP6 MME, would the "recommendations" for future MME design be different? As another example, I mention above that only 4 ISMs are included in the MME, two of which account for > 90% of*

*the simulations. By excluding CISM from the training experiments, the authors then make "recommendations" about the ability of the emulator to estimate the slc simulated by ISMs not included in the design. Here also, this evaluation depends critically on how similar simulated slc from CISM is to the simulated slc from IMAUICE. This similarity depends on numerous aspects that are specific to these two particular models. I would expect that the "recommendations" would be very different if other ISMs (ISSM, PISM ...) show more or less similarity with CISM.*

We thank Referee #2 for the insightful analysis. Here we feel that some clarifications about $\kappa$ should be given. As described in Sect. 2, we rely on a standard approach for integrating ocean forcing, i.e. based on an empirically derived retreat parameterization for tidewater glaciers (Slater et al., 2019, 2020). In this approach, $\kappa$ is not a parameter in the ice-flow model; it rather represents the sensitivity of the ocean forcing as a whole. It may be thought of as defining the sensitivity of the downscaling from global model to local ice sheet scale, similar to the combined parameter choices in RCMs for downscaling climate conditions. In the studied MME, we have different RCMs, which have different sensitivities and produce different melt for the same global forcing.

Regarding the specific comment on the quasi linear behaviour, we expect this relationship not to change much by adding more ISMs because $\kappa$ is "external" to the ISM as afore-described. To support this result, Referee #2 should refer to the following figure adapted from Fig. 7 of Rohmer et al. (2022) based on ISMIP6 MME: it shows the sensitivity index (denoted $\mu$) that measures the contribution, in terms of sea level equivalent SLE, depending on the value of $\kappa$. A quasi-linear trend has here been identified.

[Figure]

We however agree with Referee #2 that a more careful attention should be paid not to 'over-interpret' our results by making the recommendations too general. The conclusions will be nuanced and reformulated in this sense. In addition, we also propose to modify the title to highlight that our results are linked to the specificities of our ensemble as follows: "Lessons for multi-model ensemble design from emulator experiments: application to a large ensemble for future sea level contributions of the Greenland ice sheet".

**Reference**
Rohmer, J., Thieblemont, R., Le Cozannet, G., Goelzer, H., and Durand, G.: Improving interpretation of sea-level projections through a machine-learning-based local explanation approach, Cryosphere, 16, 4637–4657, https://doi.org/10.5194/tc-16-4637-2022, 2022.

*Such assumptions are not made explicit by the authors. This could very well be misleading to the readership targeted by the authors, especially those less familiar with ice sheet modeling (e.g., "stakeholders" (L328) and "coastal adaptation practitioners" (L332)).*

We totally agree with Referee #2. The clarification on $\kappa$ will be added to Sect. 2 to improve our message to stakeholders and coastal adaptation practitioners.

**Major comment 2: Characterization of uncertainty**

*The authors use their random forest (RF) emulator such that "changes in the emulator's predictive performance and the emulator-based probabilistic projections provided information on several aspects" (L18). After reviewing the manuscript, I identify remaining limitations about the RF emulator regarding uncertainty characterization.*

*The authors use changes in the predictive performance of the RF as a proxy for uncertainty remaining about a hypothetical MME (here, a MME excluding some of the experiments). But this metric is sensitive to the particular machine learning model used for the emulation. Here, the emulation output is thus conditioned on the RF architecture, with a single fixed combination of hyperparameters. Is any decrease in predictive performance of this specific RF therefore a meaningful assessment of uncertainty imputable to the MME design? This question is critical, because the conclusions of this study use this as a fundamental assumption.*

*This issue is further exacerbated by the fact that the RF does not provide probabilistic output. By this, I mean that the RF only provides a point estimate. There is no uncertainty quantification. Ideally, the design of a MME should target the strongest reduction in posterior covariance (i.e., the uncertainty remaining given the current MME). But this particular RF emulator does not provide such metric. This could be addressed by choosing another architecture (e.g., Gaussian processes, Williams and Rasmussen (2006)), by subsampling techniques for RF models (Mentch and Hooker, 2016), or by adapting the RF to output conditional quantiles (Meinshausen and Ridgeway, 2006).*

We thank Referee #2 for this suggestion. We agree that complementing the study with uncertainty quantification of the emulators itself would bring new insights and will allow us to better discuss the results. As suggested, we propose to implement the quantile random forest emulator, denoted qRF, for both the experiments on the emulator's performance (Sect. 3.3) and for the probabilistic projections (Sect. 3.4).

For the former application, the quantile random forest provides estimates of quantiles at any order $\tau$, denoted $q^\tau(slc|\mathbf{x}^*)$ for a given instance of the input variables $\mathbf{x}^*$. The quantiles can directly be used to define the prediction intervals at any level $\alpha$: $[q^{(1-\alpha)/2}(slc|\mathbf{x}^*); q^{(\alpha+1)/2}(slc|\mathbf{x}^*)]$. The following figure is a new version of Fig. 4(a), which allows to verify the satisfactory level of predictability for a large range of GSAT values with $Q^2$ ranging from 82%, for the largest GSAT values, to >98%.

[Figure]

**New Figure 4.** Comparison between the true numerically computed *slc* and the emulator's predicted values for the 25 validation tests (described in Sect. 2.4). Each panel corresponds to test samples for a given range of GSAT values (indicated at the top of the panel).

For the latter case, the emulator uncertainty is propagated in addition to the uncertainties to the different input variables based on the following procedure:
(1) Draw a random realization of the input variables $\tilde{\mathbf{x}}$;
(2) Draw a number $\tilde{u}$ between 0 and 1 by assuming a uniform random distribution;
(3) Compute $\widetilde{slc} = q^{\tilde{u}}(slc|\tilde{\mathbf{x}})$ given $\tilde{u}$ using the qRF model.

In addition, we propose to add a new performance indicator to analyse the changes in the emulator's performance in terms of reliability of the predictive probabilistic distribution. This is done using the continuous ranked probability score, denoted *crps*, as used for validating probabilistic weather forecast (Gneiting et al., 2005). To evaluate the *crps* score, the formulation based on quantiles (Berrisch and Ziel (2024): Eq. 2) is used:

$$crps = 2\int_0^1 B(q^\tau(slc|\mathbf{x}^*), slc^{true})\, d\tau \approx \frac{2}{P}\sum_{\tau\in\Gamma} B(q^\tau(slc|\mathbf{x}^*), slc^{true})$$

where the term $B(q^\tau(slc|\mathbf{x}^*), slc^{true})$ is defined as
$\begin{cases}(1-\tau)(q^\tau(slc|\mathbf{x}^*) - slc^{true}) \text{ if } slc^{true} < q^\tau(slc|\mathbf{x}^*) \\ \tau(slc^{true} - q^\tau(slc|\mathbf{x}^*)) \text{ if } slc^{true} \geq q^\tau(slc|\mathbf{x}^*)\end{cases}$, where $slc^{true}$ is the true value of the
sea level contribution, and where the quantiles $q^\tau(slc|\mathbf{x}^*)$ are evaluated using the trained qRF model at given instance of the input variables $\mathbf{x}^*$ for an equidistant dense grid of quantile levels $(\tau_1, ..., \tau_P)$ with $\tau_i < \tau_{i+1}$ and $\tau_{i+1} - \tau_i = 1/P$. In this study, we consider level $\tau_1$=5% and $\tau_P$=95% with 1/P=5%.

This score jointly quantifies the calibration of qRF probability distribution, i.e. the reliability of the estimation, and its sharpness (i.e. the concentration/dispersion of the probability distribution). The lower *crps*, the higher the quality of the qRF probabilistic predictions, with a lower limit of zero.

Our first tests (Figure below) show some similarities with the results for *RAE* and *Q²* performance indicators, but with a clearer effect of ISM experiments (CISM and noCISM). The results will be finalised in the revised version of the manuscript provided in the second phase of the reviewing process. A more thorough discussion will be provided in Sect. 4.

[Figure]

**New Figure. Relative difference (in %) for the estimates of RF predictive capability measured by *CROS*, between the RF reference solution and the RF emulators trained by applying the experiment described in Table 2. The dots indicate the results of the 25 repetitions of random validation tests (described in Sect. 2.4).**

Finally, we will also elaborate more in the Discussion section on the problem of model uncertainty related to the construction of the emulator; in particular the problem of hyperparameters' tuning and its relatively lesser impact for random forest models (Probst et al., 2019; Bischl et al., 2023).

**Added references**
Berrisch, J., Ziel, F., 2024. Multivariate probabilistic crps learning with an application to day-ahead electricity prices. International Journal of Forecasting, 40(4), 1568-1586, doi:10.1016/j.ijforecast.2024.01.005

Bischl, B., Binder, M., Lang, M., Pielok, T., Richter, J., Coors, S., et al., 2023. Hyperparameter optimization: Foundations, algorithms, best practices, and open challenges. Wiley Interdisciplinary Reviews: Data Mining and Knowledge Discovery, 13(2), e1484, doi: 10.1002/widm.1484

Gneiting T, Raftery AE, Westveld III AH, Goldman T. 2005. Calibrated probabilistic forecasting using ensemble model output statistics and minimum crps estimation. Monthly Weather Review 133(5): 1098–1118

Probst, P., Wright, M. N., Boulesteix, A. L., 2019. Hyperparameters and tuning strategies for random forest. Wiley Interdisciplinary Reviews: data mining and knowledge discovery, 9(3), e1301, doi:10.1002/widm.1301

**Minor comment 1: Lack of technical information**

*All the results and conclusions from the study are dependent on the RF emulator. As such, I find that more information on the RF development and evaluation are needed. I highlight some aspects to prioritize here below.*

*(a) The evaluation of the RF (L174-183) is assessed through a random sampling evaluation, but I find the details about the evaluation method somewhat unclear. First, the authors mention the "iteration of the procedure" (L180). However, it is not explained what is iterative in this procedure. Later in the manuscript, the authors often refer to "25 validation tests" (e.g., caption of Figure 4). But this number of 25 is not explained in the description of the evaluation*

*method. Thus, I can only assume that the random validation is iterated 25 times. Second, it is unclear what the validation performance measure shown in Figure 4 represents. In Figure 4a, there are clearly much more than 25 points, but clearly less than 25 × 55 = 1375 points (where 55 is the number of test samples mentioned on L180). Thus, what does each point represent? In addition, why are there much less points shown in Figure 4b than Figure 4a? Finally, the authors explain that there are 55 test samples, but they draw 5 samples for 10% ranges between 0 and 100% (L179-180). As such, there should be 5 × 10 = 50 test samples I believe, not 55.*

*(b) I wonder why it was decided to use this random evaluation procedure. In particular, the commonly-used 10-fold cross validation procedure would have been a more natural choice. This would also avoid the influence of sampling biases related with the random sampling of relatively few experiments (55 from the 1303 experiments per iteration). Since 10-fold cross validation was used for parameter fitting (L197), I suppose that there is no computational issue for this. Also, it would be straightforward to exclude the members from the 9 training folds as required by the specific experiments (e.g., exclude all SSP5-8.5 when training for woSSP585). Thus, is there any reason to prefer the random evaluation over the 10-fold cross validation?*

We reply here to comments (a) and (b). Referee #2 is right to say that 10-fold cross validation would be a more natural choice. The reason for proposing an alternative validation procedure is to make sure to reflect the ability of the emulator to perform well over a wide range of GSAT values instead of randomly selected cases. This ability is important in our case, because we discuss the performance with respect to the probabilistic projections given fixed GSAT values. The new Figure 4 (Major comment 2) illustrates this type of analysis.

Though our GSAT definition does not strictly correspond to the global warming level defined in AR6, they can help end users to interpret the projections associated to temperature constraints as illustrated by recent projections for France by Le Cozannet et al. (2025).

We also thank Referee #2 for noticing the problem with the number of test cases. The presentation in Sect. 2.4 has been clarified. We also propose to increase the number of test cases, now 10 per bin of GSAT values resulting in 100 test samples covering a larger range of GSAT values (new Figure 4, above).

**Added reference**
Goneri Le Cozannet, Remi Thieblemont, Jeremy Rohmer and Cecile Capderrey (2025). Sea-level scenarios aligned with the 3rd adaptation plan in France. (in press) https://doi.org/10.5802/crgeos.290

*(c) More technical details about the RF emulator construction would be beneficial. In particular, mixing categorical and continuous inputs is not straightforward, and may incur performance sensitivity to the RF design. For example, what is the splitting criterion used: mean absolute error, mean squared error, other? And how did the authors alleviate the potential issue of selection bias towards the inputs that have more possible splits? This could partly influence the different sensitivities to, for example, SSP5-8.5 scenario (global annual mean surface air temperature change, GSAT, is a continuous input with many different values), ISM (categorical input), κ (continuous input with few different values). As such, some information on these technical aspects would help the reader understand how modeling challenges may affect the results or not.*

We agree that more technical details should be added.

We use the mean squared error in the loss function of the random forest model. The treatment of categorical variables is based on the recommendation by Hastie et al. (2009): chapter 9.2.4.

We follow the implementation proposed by Wright et al. (2019), who showed that ordering the factor levels a priori, here by their mean response, is at least as good as the standard approach of considering all 2-partitions in all datasets considered, while being computationally faster; It has been shown to be more efficient than dummy coding and simply ignoring the nominal nature of the predictors as well.

However, as Referee #1 highlighted in her/his first comment, "the most readers of this journal are likely geoscientists, who will primarily be interested in the study's results". We propose to move these details in Appendix A on the Random Forest implementation.

**Added references**
Hastie, T., Tibshirani, R., Friedman, J. (2009). The Elements of Statistical Learning. Springer, New York. 2nd edition.
Wright, M. N., & König, I. R. (2019). Splitting on categorical predictors in random forests. PeerJ, 7, e6339.

**Minor comment 2: Use of global mean temperature change**

*The authors aggregate all the combinations of emission scenario (SSP) and global climate model (GCM) as a value of GSAT. I wonder if this does not risk misrepresenting the climate forcing affecting the Greenland ice sheet (GrIS). In particular, a given GSAT could very well lead to different magnitudes of:*
*(1) GrIS surface air temperature change*
*(2) GrIS precipitation*
*(3) GrIS ocean forcing*
*I expect that there may well be some substantial differences in these 3 components between different GCMs. It would be interesting to explore whether separating the single GSAT variable into these 3 separate components refines the emulator predictions.*

This choice is based on the approach followed by previous emulation studies (Edwards et al. (2021). An alternative would use regional climate variables. Although this would improve the signal-to-noise ratio for the emulator, but would restrict us to using computationally expensive general circulation models from CMIP5/6, for which there only a few tens of models. With the GSAT option, as performed by Edwards et al. (2021), the simple climate model like FaIR can be also used to explore uncertainties in each scenario thoroughly, using the latest assessments of equilibrium climate sensitivity.

Referee #2 is also invited to refer to our reply to major comment 1 on the clarification of $\kappa$ which is closely related to the sensitivity of the ocean forcing as a whole.

Finally, in order to be more consistent with IPCC practices, we will slightly change the definition of GSAT by computing the difference between the temperature at the considered year and the mean temperature over the period 1995-2014. To ensure consistency, this is also done for *slc*, which is computed relative to the mean *slc* over the same period.

**Minor comment 3: Interpretation of some results**

I find that the interpretation of results are not always well supported quantitatively. I note that, in some cases, this may simply be due to a lack of clarity in the interpretation. I provide here a few examples.

**2.1 The Dh, DS definition**

*In Figure 6, the authors show the different combinations of decrease in MME size (DS) and deviations from original histograms (Dh) resulting from their model experiments. Firstly, I think that the manuscript would benefit from a clearer definition of Dh. It is defined as "the average difference in the count numbers between the two histograms (normalised by the total number of members)" (L172-173). I believe that the normalization is by the histogram counts, not the total number of members, because otherwise Dh would be proportional to DS. For example, assume that for a given variable, we have a hypothetical 3-category histogram with counts 5, 10, 85 (i.e., n=100). In hypothetical experiment 1, the counts are 0, 10, 85 (i.e., n=95). In this case, DS = 100−5/100 = 0.95 and, following the definition, Dh = 5+0+0 /3 × 1/100 = 1/60 . In hypothetical experiment 2, the counts are 5, 10, 80 (i.e., n=95). In this case, DS = 100−5/100 = 0.95 and Dh = 0+0+5/3 × 1/100 = 1/60 . This shows that taking "the average difference in the count numbers between the two histograms (normalised by the total number of members)" results in an identical pair (DS,Dh) for these two hypothetical experiments. I am probably misunderstanding here, but I think that a more precise definition would help.*

We thank Referee #2 for this insightful comment which made us rethink our procedure. As rightly shown by Referee #2, the proposed indicator $D_h$ might fail to reflect the changes in the histograms.

As a remedy, we propose to rely on a well-established criterion for comparing different probability distributions, namely the Kolomogorov-Smirnov (*KS*) statistic, defined as:

$$KS_X = \max |F_{\text{ref}}(X) - F_{\text{ref}|E_i}(X|E_i)|$$

where $F_{\text{ref}}$ is the cumulative probability distribution CDF for the input $X$ considering the reference solution, without perturbation; $F_{\text{ref}|E_i}(X|E_i)$ is the cumulative probability distribution CDF for $X$ when applying the emulator experiment, i.e. when removing members. For categorical inputs, the CDF is defined by assigning each level an integer index.

The new $D_h$ criterion is then defined as the average value of $KS_X$ over all $X$s. By doing so, we make sure that $D_h$ reflects the average value of the deviations in the members' distributions.

The new Figure shows how each experiment affects the MME. The use of the new $D_h$ criterion is more 'discriminative' by highlighting different groups of points:
-   The group of points at the bottom, left corner corresponding to MAR, Narrow Kappa, woSSP245 and woSSP126. They all have minor-to-moderate impact on both criteria, $D_S$ and $D_h$;
-   The group of points with decrease in size $D_S$ on the same order of magnitude, i.e. between 55 and 65% with different behaviours in terms of histograms' perturbation as shown by the wide range of $D_h$ values. This group corresponds, in increasing order of $D_h$ to 'Med. and Extr. Kappa', woSSP585 and woCISM;
-   The point at an intermediate position with moderate $D_S$ and $D_h$ changes which corresponds to CISM.

[Figure]

**New Figure 6. Position of the emulator's experiment in a ($D_h$, $D_s$) diagram where $D_s$ measures the relative decrease in the MME size after applying the experiment, and $D_h$ measures the deviation of the histograms from the original ones.**

**2.2 The Dh, DS results**

*I do not understand the interpretation of the impact from Dh, DS on the emulator performance (Sect. 3.3). First, the authors write "Excluding the extreme SSP scenario SSP5-8.5 (experiment 'woSSP585') has the largest impact in terms of RAE relative difference with respect to the original RF performance (Sect. 3.1), where RAE is increased of $\sim$ 10% compared to the original RAE value (Fig. 4)" (L245). However, Figure 7 shows a $\sim$ 275% relative difference in RAE, so it is not clear to me where the value "$\sim$ 10%" comes from. Second, I do not follow the logic of the arguments. The authors write that (i) the high DS of woSSP585 causes large errors. But then, (ii) they argue that "this 'size effect' is not the only contributor to the performance impact, as shown by the 'woCISM' experiment, which removes an equivalent number of members to the 'woSSP585' experiment (Fig. 6), and the resulting RAE increase reaches half that of 'woSSP585' experiment" (L253). And (iii) that the woCISM experiment has the largest Dh value. However, when I interpret Figures 6 and 7, I find that (a) woSSP585 and woCISM have similar DS values (i.e., (ii)), (b) woCISM has higher Dh than woSSP585 (i.e., (iii)), but (c) that the errors from woSSP585 ar much higher than those of woCISM (Figure 7). So, it seems that the lower Dh of woSSP585 is accompanied by larger errors. This is the opposite message to that conveyed in the text: "This shows that the second important factor here is the diversity among the members within the MME after applying the experiment. The Dh indicator remains, however, a first-order approximation of this diversity (...)" (L256). The statement of greater diversity leading to lower errors, is not supported by the larger errors of woSSP585 compared to woCISM. To summarize: DS(woSSP585) $\approx$ DS(woCISM), Dh(woSSP585) $<$ Dh(woCISM) where low Dh implies greater "diversity", but RAE(woSSP585) $>>$ RAE(woCISM).*

In the revised version of the manuscript, we will discuss in more details the links between the groups highlighted above and the implications with respect to the emulator's performance as well the influence on the probabilistic projections. Note also that the results will be updated taking also into account new members with HIRHAM RCMs.

**2.3 Figure S3 (in Section S2)**

*The authors write "The analysis of an alternative indicator of emulator's predictive capability in Supplementary materials S2 confirms these results" (L261). However, in my view, Figure 6*

*(RAE results) and Figure S3 (Q2, coefficient of determination results) show contrasting conclusions. For example, RAE of woCISM, CISM, and MAR are comparable (Figure 6). However, Q2 is clearly lower for woCISM than for MAR and CISM (Figure S3). This indicates differences when evaluating relative errors versus explained variance. Thus, these differences are potentially interesting to analyze, instead of being discarded as is done in the main text. In particular, they could relate to the emulator performance sensitivity to high versus low slc (the latter being more influential on relative metrics), or its sensitivity in the ability to predict values away from the mean value, or other aspects that would require investigation. Note that this links back to my general comment about the importance of understanding the RF emulator, because the interpretation of the results depends strongly on this understanding.*

We thank Referee #2 for this valuable suggestion. We totally agree with Referee #2 and will include in the analysis not only *RAE* but also *Q²*. In addition we also propose to analyse the performance indicator crps that measures the quality of the emulator's predictive probability as well. Referee #2 is invited also to refer to our detailed reply to Major comment 2.

**2.4 Figure 8**

*There are many aspects that I find puzzling or questionable in Figure 8. Firstly, the results do not correspond to what is shown in Figure S4, where the Q5% and Q95% are shown with the black error bars. For example, in the column ΔGSAT=+3°, Q95% of woCISM, woSSP245, and woSSP585 are clearly strongly different from the Q95% labeled "original" (Figure S4). But Figure 8 shows that these differences are ≤ 1%. I believe that there is an inconsistency here, or something that I misunderstand about Figure 8.*
*Secondly, I do no understand how it is possible that the changes in median and quantiles at ΔGSAT=+4° are so small for woSSP585. In this design experiment, the RF model has presumably not even seen such levels of warming during training because the SSP 5-8.5 scenario has been excluded. But, by definition, tree models (including RF) predict slc based on decision rules seen during training. Thus, it is not clear how the RF can predict relatively similar slc values under ΔGSAT=+4° when excluding SSP 5-8.5 as when it is not excluded. I am probably misunderstanding something here, but I believe that the authors should explain this counter-intuitive aspect of their results.*

We thank Referee #2 for his careful analysis. We confirm that, for some experiments, there are some discrepancies that have revealed a bug in our scripts for the plotting. We will of course update the results in the new manuscript.

**Minor comment 4: Some conclusions need to be put into perspective**

*For different aspects, I find that better communication and/or more context about the conclusions is needed. I highlight some key examples here.*

*(a) Concerning κ, the authors argue for "the lesser importance of the choice in the range of the Greenland tidewater glacier retreat parameter" (L21). However, they compare it with the influence of the SSP scenario and of the ISM choice. It is expected that a single parameter should have much less influence than a global warming scenario and than a full ice sheet model.*

We thank Referee #2 for this comment. We feel that some clarification about κ should here be given. Here, κ is not thought as a parameter of the ice-flow model, it rather represents the sensitivity of the ocean forcing as a whole. It may be thought of as defining the sensitivity of the downscaling from global model to local ice sheet scale, similar to the combined parameter choices in RCMs for downscaling climate conditions. In the studied MME, we have different RCMs, which have different sensitivities and produce different melt for the same global forcing.

Since we have only one approach to 'downscale' the ocean forcing, $\kappa$ is sampling that uncertainty in a similar way.

In addition, our previous study (Rohmer et al., 2022) highlighted the high importance of $\kappa$ compared to other uncertainties. Referee #2 is also invited to refer to our detailed reply to major comment 1 as well to Referee #1's comment 3.

**Reference**
Rohmer, J., Thieblemont, R., Le Cozannet, G., Goelzer, H., and Durand, G.: Improving interpretation of sea-level projections through a machine-learning-based local explanation approach, Cryosphere, 16, 4637–4657, https://doi.org/10.5194/tc-16-4637-2022, 2022.

*(b) It should be better emphasized that the probabilistic ranges shown by the authors are not probabilistic projections of Greenland slc. Instead, they show a range of emulator predictions (thus conditioned on the emulator architecture) assuming a uniform distribution over the different inputs (L186). Thus, it does not represent calibrated uncertainty accounting for model-observation misfits (e.g., Aschwanden and Brinkerhoff, 2022). And neither does it represent the slc PDF from the MME, because the uniform distribution over the input space is not representative of the MME itself (e.g., the minimum spatial resolution is clearly not uniform between 1 and 40 km, see Fig. 3). As such, I believe that the true meaning of the PDFs shown in Figure 5 should be explained explicitly in order to avoid any reader misinterpreting those PDFs.*
We agree with this comment. To do so, we propose to clarify the caption of Fig. 5 as well as the description in Sect. 2.4.

*(c) The authors make a conclusion on "the utmost importance of including the SSP5-8.5 scenario, due to the large number of simulations available and the range of global warming they cover" (L19- 20). However, I do not think that the authors have proven the co-existence of these two points. For example, could it be that including only a few training simulations with high global warming forcing would be sufficient to drastically decrease the errors of woSSP585 shown in Figure7? In other words, maybe the emulator needs only a few high-warming training examples to correctly interpolate in the existing range of warming scenarios. Or maybe, as the authors write (L19-20), it is also the high number of experiments that is important. However, as far as I understand, the results presented in this study do not allow to evaluate the relative importance of these two aspects.*
We agree with Referee #2 and it also goes in the same line than the comment on the dependence of the results to the MME (Major comment 1). We will nuance our conclusions in this sense. We believe that the influence of the MME size is shown by our results, but disentangling this effect from the range of global warning remains too complicated at least with the procedure proposed here.

**References**

*Andy Aschwanden and DJ Brinkerhoff. Calibrated mass loss predictions for the greenland ice sheet. Geophysical Research Letters, 49(19):e2022GL099058, 2022.*

*Nicolai Meinshausen and Greg Ridgeway. Quantile regression forests. Journal of machine learning research, 7(6), 2006.*

*Lucas Mentch and Giles Hooker. Quantifying uncertainty in random forests via confidence intervals and hypothesis tests. Journal of Machine Learning Research, 17(26):1–41, 2016.*

*Christopher KI Williams and Carl Edward Rasmussen. Gaussian processes for machine learning, volume 2. MIT press Cambridge, MA, 2006.*

We thank Referee #2 for the suggested references.

Orleans,
May 5th, 2025
J. Rohmer[1] on behalf of the co-authors

[1] BRGM, 3 av. C. Guillemin - 45060 Orléans Cedex 2 – France

---

## Author Response (AR1)

**Replies to Referees' comments on "Drawing lessons for multi-model ensemble design from emulator experiments: application to future sea level contribution of the Greenland ice sheet" (egusphere-2025-52)**

We would like to thank both Referees for the constructive comments. We agree with most of the suggestions and, therefore, we have modified the manuscript to take on board the comments and suggestions. We recall the reviews and we reply to each of the comments in turn (outlined in blue). The page and line numbers are those of the document with track-changes.

**Additional changes**

Since the submission of this manuscript, we had the opportunity to include HIRHAM RCM in the MME as well. In addition to the modifications suggested, we have thus included this third RCM model in the analysis. A major advantage is to define an experiment where only the members of HIRHAM and RACMO are used ('woMAR' experiment) in addition to the 'MAR only' experiment. We expect that this modification brings new insights and strengthens our conclusions which are, in the original version of the manuscript, based on two RCMs only. The results discussed in Sect. 4 have been modified accordingly.

**Referee #1:**

Review of "Drawing lessons for multi-model ensemble design from emulator experiments: application to future sea level contribution of the Greenland ice sheet" by Rohmer et al. The paper "Drawing lessons for multi-model ensemble design from emulator experiments: application to future sea level contribution of the Greenland ice sheet" investigates the impact of different choices in constructing an emulator capable of predicting the future contribution of the Greenland Ice Sheet to sea-level rise. Specifically, the authors examine how the selection of models and parameters associated with different physical processes and scales (climate scenarios, regional climate models, ice-flow models, and ice-flow parameters) influences the emulator in terms of its fidelity, the estimated contribution to sea-level rise, and the associated uncertainty.

This study represents a valuable contribution to our understanding of multi-model ensemble approaches and emulator design. The numerical experiments are clearly described, and the results are noteworthy. Overall, the paper provides an interesting addition to the scientific literature. Nonetheless, I have a series of comments and questions that I would like the authors to address. On the one hand, in terms of the way the manuscript is written, some sections could be streamlined for conciseness (see my first general comment below, as well as specific comments). On the other hand, I have a series of more fundamental general comments on key aspects of the manuscript. There are three main comments, described herefater, which are followed by a series of specific comments addressing more detailed points. Once these concerns have been satisfactorily addressed, I will be happy to recommend the paper for publication in the Special Issue: Improving the contribution of the land cryosphere to sea level rise projections.

We thank Referee #1 for the positive analysis. We have taken into account the comments and suggestions. In what follows, we describe in detail the corrections.

**General comment 1**

This paper lies at the intersection of two fields: glaciological modeling and statistical methods. Such interdisciplinary studies are particularly valuable, as the glaciology community may not be fully familiar with statistical techniques, while statisticians may not always be aware of the challenges involved in estimating future sea-level rise. Furthermore, comparative studies have gained increasing importance in glaciology, as they help assess the robustness of different modeling approaches. Given this, it is crucial to also investigate how these comparisons are constructed in the first place. In this context, the present study is highly relevant.

That being said, I believe the paper could be more explicit about its practical conclusions, specifically conclusions (1) and (2) mentioned in the abstract. I would also suggest highlighting the conclusions related to the (different) impact of the RCM/ISM choice in the abstract and in the introduction (see lines 313–315).

Additionally, some technical details could be either removed or moved to an appendix or supplementary materials. The reasoning behind these suggestions is that most readers of this journal are likely geoscientists, who will primarily be interested in the study's results. By streamlining technical details, the paper's key findings —which are noteworthy— could be emphasized more effectively, improving its accessibility.

We thank Referee #1 for these suggestions. We totally agree that the results described in our manuscript should be transferred to a wide readership that is not necessarily specialists of the methods.

To do so, the following modifications have been made:

- The technical details of the emulator implementation are placed in Appendix A. We also added a new Appendix D to detail the methods for the performance analysis suggested by Referee #2. Finally, the specific comments of Referee #2 are also integrated in Appendices to decrease the level of technicality of the core text;
- The conclusions of the RCM/ISM are more highlighted in the abstract and in the conclusions:
- In order to improve the transferability of our message, some choices made for the representation of the results are modified to be more consistent with IPCC standards, i.e. with more largely shared practices:
  - We slightly change the definition of GSAT by computing the difference between the temperature at the considered year and the mean temperature over the period 1995-2014;
  - We analyse in Sect. 3.3 the changes in the likely range, i.e. 17th and 83rd percentile instead of the 5th and 95th percentile. We believe that the use of the IPCC calibrated language can also ease the communication of our results.

I also wonder whether subsections 2.4 and 3.2 could be simplified by focusing on just one of the two indicators (DS and Dh), e.g. DS. While some variability has been observed in the results, these indicators are strongly correlated (see figure 6). Furthermore, from a practical point-of-view, what really matters is the number of runs to be made (i.e., of DS). Similarly, figure captions could be streamlined by omitting details that may not be particularly useful (e.g., the exact method used for kernel density estimation; see specific comments below). The technical lines 196–219 could also be relegated to supplementary materials.

We agree with Referee #1's comment: the number of members is a key criterion, and the evolution of the emulator's predictive capability clearly highlights this aspect. Our primary idea for proposing a second criterion was to be able to reflect how much the distribution of the members is modified in addition to the size. As rightly indicated by Referee #1, our second

criterion  $D_h$  fails, however, to highlight key situations because of the too strong correlation with  $D_S$ . This goes also in the same line with Referee #2's comment (minor comment 2.1).

Therefore, we propose to remove this analysis from the main text. We propose then to support the discussion in Sect 4.1 with a complementary analysis (Supplementary materials S3) based on a well-established, and more widely used, criterion for comparing different probability distributions, namely the Kolomogorov-Smirnov (KS) criterion, instead of a criterion constructed from 'scratch'.

Figure S7: Position of the emulator's experiment in a  $(D_h, D_s)$  diagram where  $D_s$  measures the relative decrease in the MME size after applying the experiment, and  $D_h$  measures the deviation of the histograms from the original ones (see Supplementary Material S3). The blue-coloured marker refers to the reference solution defined as the mean value over the 25 iterations of the random validation exercise, described in Sect. 2.4, applied to the original dataset.

**General comment 2**

Regarding the paper's methodology, I have some questions about how the way ensembles are introduced. Specifically, lines 34–36 state: "Each member of a MME should evenly span a representative and exhaustive set of plausible realizations of the combined sources of uncertainty, (...), equally represented by a single model run". This suggests that each member of the ensemble should have the same weight.

However, I find this somewhat misleading, and I believe the authors could elaborate further on this choice. The assumption that all runs should have equal weight holds only if our current knowledge suggests they are equally probable. From a Bayesian perspective, this would correspond to assuming a uniform prior distribution. However, this assumption may not always be justified. For example:

- (i) Runs from lower-resolution models might be considered less reliable than those from higherresolution models as they might not capture relevant small-scale processes.
- (ii) Some values of the uncertain parameters might be less probable if they lead to results that deviate significantly from current observations.

Formally, these concerns can be addressed by updating the weights of each run based on their likelihood given observational data (e.g., Aschwanden and Brinkerhoff, 2022; Nias et al., 2023).

I understand that the authors did not include such a step in their analysis, as their focus was on assessing the emulator's capabilities. However, it seems to me that this point should be discussed in the methodology or discussion section for two reasons. First, to clarify for the reader that the choice of equal probability stems from an assumption about our current knowledge and that alternative approaches are possible. Second, because weighting model runs based on observational constraints is an emerging direction in the field, and this should be discussed in the context of future ISMIP ensemble designs. This could also be mentioned as a perspective for future work, as it would be interesting to see whether the conclusions remain valid when runs are weighted as a result of a calibration.

We agree with Referee #1 that this point merits further discussion, particularly in view of the forthcoming ISMIP7. The primary goal our study was to study the influence of different factors for a given MME, i.e. discovering the influence of groups of members. Thus, the implicit assumption of this procedure is that we do not assign any weight to the members *a priori*. This is clarified on page 8, in Sect. 2.3 (line 180); this was also Referee #2's suggestion (Minor comment 4(b)).

This clarification is more particularly useful in Sect. 4 regarding the meaning of the emulator-based probabilistic predictions. These projections do neither represent calibrated uncertainty accounting for model-observation misfits nor the *slc* probability distribution from the MME, because the uniform distribution over the input space is not representative of the MME itself (e.g., the minimum spatial resolution is clearly not uniform between 1 and 40 km, see Fig. 3). This is clarified on page 11, in Sect. 2.4.2 (lines 227-234).

In addition, the alternative option based on weighting either through expertise (as illustrated by Referee #1 with the resolution) or based on model-observation misfits is now discussed in Sect; 5. We propose to highlight the benefits of the weighting approach, but also the challenges to do it, namely: (1) the need for good quality data; (2) the need for data over a large period in the past; (3) the need for some types of ISMs to adjust or adapt their implementation. This is clarified on page 20, in Sect. 5 (lines 424-426).

**General comment 3**

My third general remark concerns the parameter  $\kappa$  associated with the calving rate. It is unclear why this particular parameter was chosen over others. From reading the paper, the rationale behind this choice is not obvious—perhaps it is based on modeling considerations or supported by previous studies? If so, it seems to me that the authors should provide a stronger justification for including this specific parameter.

More fundamentally, I wonder whether comparing the effect of  $\kappa$  to that of RCP scenarios or RCMs is entirely meaningful. This comparison contrasts the impact of a single parameter of an ice-flow model with that of an entire climate scenario or a regional climate model, which incorporate numerous physical parameters. Given this, it is perhaps unsurprising that the effect of  $\kappa$  appears quite limited.

We thank Referee #1 for this suggestion. We feel that some clarification about  $\kappa$  should here be given.

In this study, we rely on a standard approach for integrating ocean forcing, i.e. based on an empirically derived retreat parameterization for tidewater glaciers (Slater et al., 2019, 2020) that is forced by a RCM-based run-off and ocean temperature changes in seven drainage basins around Greenland. In this implementation, retreat and advance of marine-terminating outlet glaciers in the ISMs are prescribed as a yearly series of maximum ice front positions (Nowicki et al., 2020).

Here,  $\kappa$  is not thought as a parameter of the ice-flow model, it rather represents the sensitivity of the ocean forcing as a whole. It may be thought of as defining the sensitivity of the downscaling from global model to local ice sheet scale, similar to the combined parameter choices in RCMs for downscaling climate conditions. In the studied MME, we have different RCMs, which have different sensitivities and produce different melt for the same global forcing. Since we have only one approach to 'downscale' the ocean forcing,  $\kappa$  is sampling that uncertainty in a similar way.

We recognize that the  $\kappa$ -based approach remains a strong simplification of the complex interaction between marine-terminating outlet glaciers and the ocean, for which physically based solutions are in development but not available for all models. However, it should be underlined that the advantage of this retreat parameterization is to be applicable in the wide variety of models under consideration. Furthermore, it is currently the most widely used approach for producing large ensemble for sea level projections, as done for instance by Edwards et al. (2021).

**This is clarified on pages 3-4, in Sect. 2.1 (lines 96-100).**

To illustrate this, one could consider a similar comparison in the opposite direction: assessing the impact of choosing a glaciological model of varying complexity (e.g., full Stokes, BP, or SIA) against a single parameter from a RCM. This would likely lead to the conclusion that the specific parameter from the RCM has a minimal influence. Therefore, I wonder whether including  $\kappa$  as an isolated parameter in this study is fully justified. Could the authors maybe clarify its relevance within the broader context of the study's objectives?

In response to the specific comment of Referee #1, our previous study (Rohmer et al., 2022) highlighted the high importance of  $\kappa$  compared to other uncertainties, in particular some of them related to the complexity of the numerical model as suggested by Referee #1, i.e. the choice in the numerical method (Finite Difference, Finite Element), the grid resolution, and the ice flow formulation (approximation, higher order, hybrid). To illustrate, the following figure (adapted from Fig. 7 and Fig. 8 of Rohmer et al., (2022) based on ISMIP6 MME) shows the sensitivity index (denoted  $\mu$ ) that measures the contribution, in terms of sea level equivalent SLE, depending on the value of  $\kappa$  (panel a) or of the choice in the ice flow formulation (panel b). The influence measured by  $\mu$  for  $\kappa$  is on the order of 1-2 cm (at most 5cm) whereas it remains on the order of 0.5-1cm for the ice flow method, hence confirming a large importance of this parameter.

**References**

Edwards, T. L., Nowicki, S., Marzeion, B., Hock, R., Goelzer, H., Seroussi, H., et al. (2021). Projected land ice contributions to twenty-first-century sea level rise. Nature, 593(7857), 74-82.

Nowicki, S., Goelzer, H., Seroussi, H., Payne, A. J., Lipscomb, W. H., Abe-Ouchi, A., Agosta, C., Alexander, P., Asay-Davis, X. S., Barthel, A., Bracegirdle, T. J., Cullather, R., Felikson, D., Fettweis, X., Gregory, J. M., Hattermann, T., Jourdain, N. C., Kuipers Munneke, P., Larour, E., Little, C. M., Morlighem, M., Nias, I., Shepherd, A., Simon, E., Slater, D., Smith, R. S., Straneo, F., Trusel, L. D., van den Broeke, M. R., and van de Wal, R.: Experimental protocol for sea level projections from ISMIP6 stand-alone ice sheet models, The Cryosphere, 14, 2331–2368, https://doi.org/10.5194/tc-14-2331-2020, 2020

Rohmer, J., Thieblemont, R., Le Cozannet, G., Goelzer, H., and Durand, G.: Improving interpretation of sea-level projections through a machine-learning-based local explanation approach, Cryosphere, 16, 4637–4657, https://doi.org/10.5194/tc-16-4637-2022, 2022.

Slater, D. A., Felikson, D., Straneo, F., Goelzer, H., Little, C. M., Morlighem, M., Fettweis, X., and Nowicki, S.: Twentyfirst century ocean forcing of the Greenland ice sheet for modelling of sea level contribution, The Cryosphere, 14, 985–1008, https://doi.org/10.5194/tc-14-985-2020, 2020.

Slater, D. A., Straneo, F., Felikson, D., Little, C. M., Goelzer, H., Fettweis, X., and Holte, J.: Estimating Greenland tidewater glacier retreat driven by submarine melting, The Cryosphere, 13, 2489–2509, https://doi.org/10.5194/tc-13-2489-2019, 2019.

**Specific Comments**

We thank Referee #1 for the specific comments. The revised version of the manuscript now incorporates all of them.

(1) [Line 12] 'projection and the quantification of its uncertainty'  $\rightarrow$  'projections and the quantification of their uncertainties'.

This has been corrected.

(2) [Lines 15, 17, and 65] You use 'experiments' for two distinct concepts: numerical simulations (e.g., line 16) and numerical tests (e.g., line 15). Consider using separate words to avoid any confusion.

To avoid confusion, we now refer to the second type of 'experiments' as 'members'.

(3) [Line 16] '(Regional Climate Model RCM, or Ice Sheet Model ISM)' → '(Regional Climate Model; RCM, or Ice Sheet Model; ISM)'.

This has been corrected.

- (4) [Line 19] Consider removing 'utmost' as it might be overly formal. This has been replaced by 'high'.
- (5) [Line 25] 'projection and the quantification of its uncertainty'  $\rightarrow$  'projections and the quantification of their uncertainties'. This has been corrected.
- (6) [Line 26] 'co-ordinated sets of numerical experiments'→'sets of numerical experiments'. This has been corrected.
- (7) [Line 47] Consider adding references related to (machine-learning-based) emulators. A list of real case applications is now provided on page 2 in Sect. 1 (lines 51-52) as follows: "[...] like linear-regression (Levermann et al., 2020), Gaussian process regression (Edwards et al., 2021), random forest regression (Rohmer et al., 2022), and deep learning-based methods (Van Katwyk et al., 2025)".

**Added references**

Levermann, A., Winkelmann, R., Albrecht, T., Goelzer, H., Golledge, N. R., Greve, R., Huybrechts, P., Jordan, J., Leguy, G., Martin, D., et al.: Projecting Antarctica's contribution to future sea level rise from basal ice shelf melt using linear response functions of 16 ice sheet 600 models (LARMIP-2), Earth System Dynamics, 11, 35–76, https://doi.org/10.1175/JCLI-D-23-0580.1, 2020.

Van Katwyk, P., Fox-Kemper, B., Nowicki, S., Seroussi, H., & Bergen, K. J. (2025). ISEFlow v1. 0: A Flow-Based Neural Network Emulator for Improved Sea Level Projections and Uncertainty Quantification. EGUsphere, 2025, 1-32.

- (8) [Line 47] I might be a bit picky here, but I would argue that the key advantage of statistical emulators is their low computational cost; being able to predict the model response at untried input values is only useful if it can be done at a reasonable cost.

  We totally agree with Referee #1. We have now underlined this aspect.
- (9) [Line 63] Please be consistent with your use of acronyms: either your define what you mean by RCM and GCM, or you use directly the corresponding acronyms. Also, a table of acronyms would be useful in the paper.

A table has been added in Appendix E.

- (10) [Line 63] Avoid using 'validation tests' as this can lead to confusion when it comes to glaciological modeling (for which 'validation' has another meaning).

  We now use the term "numerical experiments".
- (11) [Line 76] '(Goelzer et al. (2020): in particular (...))'  $\rightarrow$  '(Goelzer et al., 2020; in particular (...))'.

This has been corrected.

(12) [Line 79] Consider adding a schematic displaying the modeling chain and indicating where modeling choices (MME inputs) are introduced. This could be very useful to effectively obtain an overview of the context.

A new Figure 1 has been added to clarify the workflow.

New Figure 1: General forcing approach for Greenland ice sheet model projections. The questions relevant for the MME design (detailed in Table 2) are related to the modelling choices made for each of the boxes.

(13) [Line 80] Here you define again what a RCM is. If you have already defined it before that is not necessary.

We have removed this part.

- (14) [Table 1] Ensure consistent formatting of symbols (italics vs. non-italics). This has been corrected.
- (15) [Table 1] Consider renaming 'Symbol' to 'Symbol/Acronym' or simply 'Name' to clarify that most entries are acronyms.

The term 'Name' is now used.

- (16) [Table 1] 'Sliding basal law'→'Sliding law' or 'Basal friction law'. The second term is now used.
- (17) [Line 101] Consider defining 'input setting' explicitly, e.g. as a particular combination of inputs.

This has been specified as suggested.

**Referee #2:**

This study develops a Random Forest (RF) emulator to emulate Greenland 2100 sea level contribution (slc) output from a Multi-model ensemble (MME). In particular, the RF is trained using a set of 7 inputs, associated with the climate scenario, the ice sheet model (ISM) used, the regional climate model (RCM) used, and different settings of the ISM run. The authors investigate how changing the MME design leads to changes in the emulator performance and in its range of emulated slc. Based on these metrics, they provide guidelines for future MME designs that aim at estimating future ranges of slc from the Greenland ice sheet.

This study addresses an important and difficult question: how can we improve the design of MMEs to provide the best information about the probability density function (PDF) of future slc? The concept underlying this study is that the MME itself does not need to characterize this PDF, but that it should be designed optimally such that an emulator can do this characterization a posteriori. This is a valid and efficient approach to uncertainty quantification. It is also a challenging topic, and work on this topic is important. However, at this stage, I believe that several points need to be improved to make this study a valuable contribution in addressing this question. The authors make recommendations and they "expect these recommendations to be informative for the design of next generations of MME" (L22). But I believe that their recommendations are dependent on many assumptions or choices that they made, without always justifying them and making them clear to the reader. Furthermore, more methodological details about the RF emulator are needed because all the results presented depend on the emulation and, therefore, the RF design influences strongly any interpretation, and thorough RF evaluation is critical as well. I detail my concerns in this review, which consists of Major and Minor comments. I do not provide technical comments at this stage of the reviewing process because I believe that the more substantial aspects should be addressed first. Line numbers in this review correspond to the preprint manuscript.

We thank Referee #2 for the in-depth analysis of the manuscript. In what follows, we provide details of the corrections made.

**Major comment 1: Inherent assumptions associated with the MME**

Many of the conclusions are strongly dependent on the particular MME used in this study. I have several reservations about this.

First, it is unclear to me how the MME used in this study was acquired and designed. The only details provided about the MME are (L74): "We focus on the sea level contribution from the Greenland ice sheet (GrIS) in 2100 based on a new MME study performed for the European Union's Horizon 2020 project PROTECT (http://protectslr.eu). Some modelling choices are taken from the protocols of the ISMIP6 initiative (Goelzer et al. (2020): in particular, the two main emissions scenarios, and the main model parameter explored."

Has this MME been peer-reviewed? Why are the authors not using the well-established ISMIP6 MME? The latter MME also has the advantage of providing a larger set of experiments, notably including many more ISMs than the MME of this study. At least, why has the MME not been combined with the ISMIP6 MME? Also, given that no publication describing the MME is referenced, I believe that it is important to give many more details about the MME configuration: Did all ISMs run under high- and low-warming forcing? Are the 15 global climate models used in the MME well-balanced across the runs? Etc.

The MME has been peer reviewed within the H2020 Protect project and has been submitted to Special Issue of the Cryosphere journal. In the revised version of the manuscript, we refer to the egusphere preprint for further details (Goelzer et al., 2025). Since the release on the egusphere platform takes some time, Referee #2 is invited to refer to:

https://drive.google.com/file/d/1S00IHWGa34mLNlLOyrHEN6Sp7q6-

EbLY/view?usp=sharing

\*\* Please to do not distribute / use outside the scope of this review\*\*

**Reference added**

Goelzer, H., Berends, C. J., Boberg, F., van den Broeke, M., Durand, G., Edwards, T., Fettweis, X., Gillet-Chaulet, F., Glaude, Q., Huybrechts, P., Le clec'h, S., Mottram, R., Noel, B., Olesen, M., Rahlves, C., Rohmer, J., van de Wal, R. S. W.: Extending the range and reach of physically-based Greenland ice sheet sea-level projections. Preprint egusphere-2025-3098, 2025.

To specifically reply to Referee #2 comment, we confirm that the experimental design builds on the ISMIP6 protocol with four different ice sheet models and extends it to more fully account for uncertainties in sea-level projections for the GrIS. This is underlined in Sect. 2. The Protect MME is thought as an extension of ISMIP6 MME regarding different aspects:

- we have included a wider range of CMIP6 climate model output, more climate change scenarios (SSP126, SSP245, SSP585);
- we have provided retreat forcing before 2015 that is calculated from reconstructions of past runoff and ocean thermal forcing. This allows for a consistent forcing of the models in past and future and to consider historical retreat of the outlet glaciers, which was an important source of mass loss after 1990;
- we have provided surface mass balance forcing from several RCMs, i.e. MAR and RACMO for the MME used for this study.

Since the submission of this manuscript, we had the opportunity to include HIRHAM as well. We believe that adding this third RCM model brings new insights and makes our conclusions more robust (based on two RCMs in the original version of the study). The results discussed in Sect. 4 have been modified.

To appreciate this extension, the following table gives the repartition of the members with respect to the RCP/SSP scenarios for ISMIP6 MME and for Protect MME (before inclusion of HIRHAM). In addition, it should be noted that the total number of members has increased by a factor of 4 compared to ISMIP6 MME.

|              | ISMIP6 MME | Protect MME |
|--------------|------------|-------------|
| RCP26        | 23         | 40          |
| RCP85        | 156        | 319         |
| SSP126       | 18         | 189         |
| SSP245       | 0          | 189         |
| SSP585       | 59         | 566         |
| Total number | 256        | 1303        |

As a second illustration, the following figure gives an overview of the results of the ensemble of projections at the year 2100 for all available Earth System models (ESMs), RCMs and ISMs under high, med and low retreat sensitivity. This allows to appreciate, graphically, how well the range of sea level is covered. It should however be underlined that, although the collection

of forcing data covers a wide range of variations across different ESMs and scenarios, it ultimately still represents an 'ensemble of opportunity' similarly as for ISMIP6 MME. Our study aims to address the potential implications of this characteristic.

Second, the conclusions of this study are strongly dependent on the initial MME used in the emulation process. For example, the authors argue that there is "a quasi-linear relationship between  $\kappa$  and slc" (L306). But this conclusion is based only on the set of 4 ISMs used in this study: CISM, IMAUICE, GISM, and ElmerIce. Furthermore, given that very little experiments were performed by ElmerIce and GISM, I assume (I need to assume here because no information is provided on the design of the MME) that these two models may well have only been run with a single  $\kappa$  value. In this case, the "quasi-linear relationship" would be derived only from two ISMs. Given that different ISMs can show very different sensitivities to movement of the tidewater glacier front and grounding line positions, this conclusion could well be very different if other ISMs are included. So, if one was to perform a similar study with the ISMIP6 MME, would the "recommendations" for future MME design be different? As another example, I mention above that only 4 ISMs are included in the MME, two of which account for > 90% of the simulations. By excluding CISM from the training experiments, the authors then make "recommendations" about the ability of the emulator to estimate the slc simulated by ISMs not included in the design. Here also, this evaluation depends critically on how similar simulated slc from CISM is to the simulated slc from IMAUICE. This similarity depends on numerous aspects that are specific to these two particular models. I would expect that the "recommendations" would be very different if other ISMs (ISSM, PISM ...) show more or less similarity with CISM.

We thank Referee #2 for the insightful analysis. Here we feel that some clarifications about  $\kappa$  should be given. As described in Sect. 2, on page 3-4 (lines 96-99), we rely on a standard approach for integrating ocean forcing, i.e. based on an empirically derived retreat parameterization for tidewater glaciers (Slater et al., 2019, 2020). In this approach,  $\kappa$  is not a parameter in the ice-flow model; it rather represents the sensitivity of the ocean forcing as a whole. It may be thought of as defining the sensitivity of the downscaling from global model to local ice sheet scale, similar to the combined parameter choices in RCMs for downscaling

climate conditions. In the studied MME, we have different RCMs, which have different sensitivities and produce different melt for the same global forcing.

Regarding the specific comment on the quasi linear behaviour, we expect this relationship not to change much by adding more ISMs because  $\kappa$  is "external" to the ISM as afore-described. To support this result, Referee #2 should refer to the following figure adapted from Fig. 7 of Rohmer et al. (2022) based on ISMIP6 MME: it shows the sensitivity index (denoted  $\mu$ ) that measures the contribution, in terms of sea level equivalent SLE, depending on the value of  $\kappa$ . A quasi-linear trend has here been identified. To complement this analysis, we rely on Supplementary Materials S3, which provides details on the quasi-linear relationship of the partial dependence plot (as originally described in the manuscript).

Finally, we agree with Referee #2 that a more careful attention should be paid not to 'over-interpret' our results by making the recommendations too general. The conclusions are nuanced and reformulated in this sense. In addition, we also propose to modify the title to highlight that our results are linked to the specificities of our ensemble as follows: "Lessons for multi-model ensemble design from emulator experiments: application to a large ensemble for future sea level contributions of the Greenland ice sheet".

**Reference**

Rohmer, J., Thieblemont, R., Le Cozannet, G., Goelzer, H., and Durand, G.: Improving interpretation of sea-level projections through a machine-learning-based local explanation approach, Cryosphere, 16, 4637–4657, https://doi.org/10.5194/tc-16-4637-2022, 2022.

Such assumptions are not made explicit by the authors. This could very well be misleading to the readership targeted by the authors, especially those less familiar with ice sheet modeling (e.g., "stakeholders" (L328) and "coastal adaptation practitioners" (L332)).

We totally agree with Referee #2. The clarification on  $\kappa$  has been added in Sect. 2 on page 3-4 (lines 96-99).

**Major comment 2: Characterization of uncertainty**

The authors use their random forest (RF) emulator such that "changes in the emulator's predictive performance and the emulator-based probabilistic projections provided information on several aspects" (L18). After reviewing the manuscript, I identify remaining limitations about the RF emulator regarding uncertainty characterization.

The authors use changes in the predictive performance of the RF as a proxy for uncertainty remaining about a hypothetical MME (here, a MME excluding some of the experiments). But this metric is sensitive to the particular machine learning model used for the emulation. Here, the emulation output is thus conditioned on the RF architecture, with a single fixed combination of hyperparameters. Is any decrease in predictive performance of this specific RF therefore a meaningful assessment of uncertainty imputable to the MME design? This question is critical, because the conclusions of this study use this as a fundamental assumption.

This issue is further exacerbated by the fact that the RF does not provide probabilistic output. By this, I mean that the RF only provides a point estimate. There is no uncertainty quantification. Ideally, the design of a MME should target the strongest reduction in posterior covariance (i.e., the uncertainty remaining given the current MME). But this particular RF emulator does not provide such metric. This could be addressed by choosing another architecture (e.g., Gaussian processes, Williams and Rasmussen (2006)), by subsampling techniques for RF models (Mentch and Hooker, 2016), or by adapting the RF to output conditional quantiles (Meinshausen and Ridgeway, 2006).

We thank Referee #2 for this suggestion. We agree that complementing the study with uncertainty quantification of the emulators itself bring new insights and allow us to better discuss the results. As suggested, we propose to implement the quantile random forest emulator, denoted qRF, for both the experiments on the emulator's performance (Sect. 3.2) and for the probabilistic projections (Sect. 3.3).

For the former application, the quantile random forest provides estimates of quantiles at any order  $\tau$ , denoted  $q^{\tau}(slc|\mathbf{x}^*)$  for a given instance of the input variables  $\mathbf{x}^*$ . The quantiles can directly be used to define the prediction intervals at any level  $\alpha$ :  $[q^{(1-\alpha)/2}(slc|\mathbf{x}^*);q^{(\alpha+1)/2}(slc|\mathbf{x}^*)]$ .

The new Figure 7 allows to verify the levels of predictability for different intervals of GSAT changes.

New Figure 7: Relative difference (in %) of the performance criteria considering the lowest GSAT values below 2.14°C (top) and the highest GSAT values above 3.83°C (bottom) for RAE (a, d),  $Q^2$  (b, e), and CRPS (c, f).

When performing the probabilistic predictions (Sect. 2.4.2), the emulator uncertainty is propagated in addition to the uncertainty of the different input variables based on the following procedure:

(Step 1) Draw N random realisations of the input variables  $\tilde{\mathbf{x}}$ ;

(Step 2.1) Draw N random number  $\tilde{u}$  between 0 and 1 by assuming a uniform random distribution;

(Step 2.2) Compute the N values  $\widetilde{slc} = q^{\widetilde{u}}(slc|\widetilde{\mathbf{x}})$  given  $\widetilde{u}$  and  $\widetilde{\mathbf{x}}$  using the qRF model;

(Step 2.3) Compute the quantile  $Q_{\widetilde{u}}^{\alpha}$  at the chosen level  $\alpha$  from the set of N values of  $\widetilde{slc}$ ;

(Step 3) Repeat n times Steps 2.1 to 2.3. At Step 2.2,  $\widetilde{slc}$  are calculated for the same set of random input variables  $\tilde{\mathbf{x}}$  defined at Step 1, but for a newly randomly generated set of levels  $\tilde{u}$  defined at Step 2.1. At Step 2.3, the newly calculated quantiles  $Q_{\tilde{u}}^{\alpha}$  vary at each of the repetitions, since each time, new random levels  $\tilde{u}$  of the qRF conditional quantiles are generated at Step 2.1.

The output of the procedure is a set of n quantile values  $(Q_{\widetilde{u}^{(1)}}^{\alpha}, Q_{\widetilde{u}^{(2)}}^{\alpha}, \dots, Q_{\widetilde{u}^{(n)}}^{\alpha})$ . The variability of the set reflects the emulator uncertainty and can be summarized by the  $\tau$ % confidence interval with lower and upper bounds defined by the  $(1-\tau)/2$ , and the  $(1+\tau)/2$  quantile of  $Q_{\widetilde{u}}^{\alpha}$ . In this study, we choose N=10,000, n=100 and  $\tau=90\%$ .

In addition, we propose to add a new performance indicator to analyse the changes in the emulator's performance in terms of reliability of the predictive probabilistic distribution. This is done using the continuous ranked probability score, denoted *CRPS*, as used for validating probabilistic weather forecast (Gneiting et al., 2005). To evaluate the *CRPS* score, the formulation based on quantiles (Berrisch and Ziel (2024): Eq. 2) is used:

$$CRPS = 2 \int_0^1 B(q^\tau(slc|\mathbf{x}^*), slc^{true}) \, \mathrm{d}\, \tau \approx \frac{2}{P} \sum_{\tau \in \Gamma} B(q^\tau(slc|\mathbf{x}^*), slc^{true})$$
 where the term  $B(q^\tau(slc|\mathbf{x}^*), slc^{true})$  is defined as
$$\{(1-\tau)(q^\tau(slc|\mathbf{x}^*) - slc^{true}) \text{ if } slc^{true}

New Figure 5: Boxplot of the RAE (a),  $Q^2$  (b) and CRPS (c) performance indicator for different ranges of GSAT (indicated on the x-axis). The lower RAE and the closer  $Q^2$  to one, the higher the emulator predictive capability. The

lower *CRPS*, the higher quality of the emulator predictive probabilistic distribution. The horizontal red dashed line indicates the median value calculated over all validation tests defined through the repeated validation procedure described in Sect. 2.4.1 considering the whole range of GSAT.

Though our GSAT definition does not strictly correspond to the global warming level defined in AR6, they can help end users to interpret the projections associated to temperature constraints as illustrated by recent projections for France by Le Cozannet et al. (2025).

We also thank Referee #2 for noticing the problem with the number of test cases. The presentation in Sect. 2.4.1 (on page 10, lines 201-211 has been clarified as follows: "In this study, we are more particularly interested in the ability of the emulator to perform well over a wide range of GSAT values. This is important in our case, because constraining the predictions to temperature constraints can help end-users to interpret the projections as illustrated by recent projections for France by Le Cozannet et al. (2025), although it should be noted that our GSAT definition does not strictly correspond to the global warming level (GWL) defined in AR6. Therefore, instead of relying on the widely used cross validation procedure (Hastie et al., 2009), we propose an alternative validation procedure adapted to our objective as follows: (1) the GSATs are classified into a finite number of intervals, the ends of which are defined by the GSAT percentiles, with levels ranging from 0 to 100% with a fixed increase of 25%. This results in the following GSAT intervals,  $[0.705, 2.14^{\circ}C]$ ,  $[2.14, 3.34^{\circ}C]$ ,  $[3.34, 3.83^{\circ}C]$ , and  $[3.83, 5.00^{\circ}C]$ ; (2) for each interval, 50 samples are randomly selected. For one iteration of the procedure, a total of  $n_{\text{test}}$ =200 test samples are randomly selected. The procedure is repeated 25 times."

**Added reference**

Goneri Le Cozannet, Remi Thieblemont, Jeremy Rohmer and Cecile Capderrey (2025). Sealevel scenarios aligned with the 3rd adaptation plan in France. (in press) https://doi.org/10.5802/crgeos.290

(c) More technical details about the RF emulator construction would be beneficial. In particular, mixing categorical and continuous inputs is not straightforward, and may incur performance sensitivity to the RF design. For example, what is the splitting criterion used: mean absolute error, mean squared error, other? And how did the authors alleviate the potential issue of selection bias towards the inputs that have more possible splits? This could partly influence the different sensitivities to, for example, SSP5-8.5 scenario (global annual mean surface air temperature change, GSAT, is a continuous input with many different values), ISM (categorical input),  $\kappa$  (continuous input with few different values). As such, some information on these technical aspects would help the reader understand how modeling challenges may affect the results or not.

We agree that more technical details should be added.

We use the mean squared error in the loss function of the random forest model. The treatment of categorical variables is based on the recommendation by Hastie et al. (2009): chapter 9.2.4. We follow the implementation proposed by Wright et al. (2019), who showed that ordering the factor levels a priori, here by their mean response, is at least as good as the standard approach of considering all 2-partitions in all datasets considered, while being computationally faster; It has been shown to be more efficient than dummy coding and simply ignoring the nominal nature of the predictors as well.

However, as Referee #1 highlighted in her/his first comment, "the most readers of this journal are likely geoscientists, who will primarily be interested in the study's results". We propose to move these details in Appendix A on the Random Forest implementation.

**Added references**

Hastie, T., Tibshirani, R., Friedman, J. (2009). The Elements of Statistical Learning. Springer, New York. 2nd edition.

Wright, M. N., & König, I. R. (2019). Splitting on categorical predictors in random forests. PeerJ, 7, e6339.

**Minor comment 2: Use of global mean temperature change**

The authors aggregate all the combinations of emission scenario (SSP) and global climate model (GCM) as a value of GSAT. I wonder if this does not risk misrepresenting the climate forcing affecting the Greenland ice sheet (GrIS). In particular, a given GSAT could very well lead to different magnitudes of:

- (1) GrIS surface air temperature change
- (2) GrIS precipitation
- (3) GrIS ocean forcing

I expect that there may well be some substantial differences in these 3 components between different GCMs. It would be interesting to explore whether separating the single GSAT variable into these 3 separate components refines the emulator predictions.

This choice is based on the approach followed by previous emulation studies (Edwards et al. (2021). An alternative would use regional climate variables. Although this would improve the signal-to-noise ratio for the emulator, but would restrict us to using computationally expensive general circulation models from CMIP5/6, for which there only a few tens of models. With the GSAT option, as performed by Edwards et al. (2021), the simple climate model like FaIR can be also used to explore uncertainties in each scenario thoroughly, using the latest assessments of equilibrium climate sensitivity.

Referee #2 is also invited to refer to our reply to major comment 1 on the clarification of  $\kappa$  which is closely related to the sensitivity of the ocean forcing as a whole.

Note that we slightly change the definition of GSAT, in order to be more consistent with IPCC practices, by computing the difference between the temperature at the considered year and the mean temperature over the period 1995-2014.

**Minor comment 3: Interpretation of some results**

I find that the interpretation of results are not always well supported quantitatively. I note that, in some cases, this may simply be due to a lack of clarity in the interpretation. I provide here a few examples.

**2.1 The Dh, DS definition**

In Figure 6, the authors show the different combinations of decrease in MME size (DS) and deviations from original histograms (Dh) resulting from their model experiments. Firstly, I think that the manuscript would benefit from a clearer definition of Dh. It is defined as "the average difference in the count numbers between the two histograms (normalised by the total number of members)" (L172-173). I believe that the normalization is by the histogram counts, not the total number of members, because otherwise Dh would be proportional to DS. For example, assume that for a given variable, we have a hypothetical 3-category histogram with

counts 5, 10, 85 (i.e., n=100). In hypothetical experiment 1, the counts are 0, 10, 85 (i.e., n=95). In this case, DS = 100-5/100 = 0.95 and, following the definition,  $Dh = 5+0+0/3 \times 1/100 = 1/60$ . In hypothetical experiment 2, the counts are 5, 10, 80 (i.e., n=95). In this case, DS = 100-5/100 = 0.95 and  $Dh = 0+0+5/3 \times 1/100 = 1/60$ . This shows that taking "the average difference in the count numbers between the two histograms (normalised by the total number of members)" results in an identical pair (DS,Dh) for these two hypothetical experiments. I am probably misunderstanding here, but I think that a more precise definition would help. We thank Referee #2 for this insightful comment which made us rethink our procedure. As rightly shown by Referee #2, the proposed indicator  $D_h$  might fail to reflect the changes in the histograms.

Therefore, we propose to remove this analysis from the main text. In addition, we propose to support the discussion in Sect 4.1, on page 18, lines 352-359 with a complementary analysis (Supplementary materials S3) based on a well-established, and more widely used, criterion for comparing different probability distributions, namely the Kolomogorov-Smirnov (KS) criterion, instead of a criterion constructed from 'scratch'.

Figure S7: Position of the emulator's experiment in a  $(D_h, D_s)$  diagram where  $D_s$  measures the relative decrease in the MME size after applying the experiment, and  $D_h$  measures the deviation of the histograms from the original ones (see Supplementary Material S3). The blue-coloured marker refers to the reference solution defined as the mean value over the 25 iterations of the random validation exercise, described in Sect. 2.4, applied to the original dataset.

**2.2 The Dh, DS results**

I do not understand the interpretation of the impact from Dh, DS on the emulator performance (Sect. 3.3). First, the authors write "Excluding the extreme SSP scenario SSP5-8.5 (experiment 'woSSP585') has the largest impact in terms of RAE relative difference with respect to the original RF performance (Sect. 3.1), where RAE is increased of  $\sim 10\%$  compared to the

original RAE value (Fig. 4)" (L245). However, Figure 7 shows a ~ 275% relative difference in RAE, so it is not clear to me where the value "~ 10%" comes from. Second, I do not follow the logic of the arguments. The authors write that (i) the high DS of woSSP585 causes large errors. But then, (ii) they argue that "this 'size effect' is not the only contributor to the performance impact, as shown by the 'woCISM' experiment, which removes an equivalent number of members to the 'woSSP585' experiment (Fig. 6), and the resulting RAE increase reaches half that of 'woSSP585' experiment" (L253). And (iii) that the woCISM experiment has the largest Dh value. However, when I interpret Figures 6 and 7, I find that (a) woSSP585 and woCISM have similar DS values (i.e., (ii)), (b) woCISM has higher Dh than woSSP585 (i.e., (iii)), but (c) that the errors from woSSP585 ar much higher than those of woCISM (Figure 7). So, it seems that the lower Dh of woSSP585 is accompanied by larger errors. This is the opposite message to that conveyed in the text: "This shows that the second important factor here is the diversity among the members within the MME after applying the experiment. The Dh indicator remains, however, a first-order approximation of this diversity (...)" (L256). The statement of greater diversity leading to lower errors, is not supported by the larger errors of woSSP585 compared to woCISM. To summarize: DS(woSSP585) ≈ DS(woCISM), Dh(woSSP585) < Dh(woCISM) where low Dh implies greater "diversity", but RAE(woSSP585) >> RAE(woCISM).

After reanalysing the results, we believe that the analysis Ds,Dh only supports the discussion in Sect. 4 but further developments may be needed to derive a robust Dh indicator. This is now clearly indicated on page 18, lines 352-359.

**2.3 Figure S3 (in Section S2)**

The authors write "The analysis of an alternative indicator of emulator's predictive capability in Supplementary materials S2 confirms these results" (L261). However, in my view, Figure 6 (RAE results) and Figure S3 (Q2, coefficient of determination results) show contrasting conclusions. For example, RAE of woCISM, CISM, and MAR are comparable (Figure 6). However, Q2 is clearly lower for woCISM than for MAR and CISM (Figure S3). This indicates differences when evaluating relative errors versus explained variance. Thus, these differences are potentially interesting to analyze, instead of being discarded as is done in the main text. In particular, they could relate to the emulator performance sensitivity to high versus low slc (the latter being more influential on relative metrics), or its sensitivity in the ability to predict values away from the mean value, or other aspects that would require investigation. Note that this links back to my general comment about the importance of understanding the RF emulator, because the interpretation of the results depends strongly on this understanding.

We thank Referee #2 for this valuable suggestion. We totally agree with Referee #2. We have included in the analysis not only RAE but also  $Q^2$ . In addition we also propose to analyse the performance indicator  $\underline{CRPS}$  that measures the quality of the emulator's predictive probability as well. Referee #2 is invited also to refer to our detailed reply to Major comment 2.

**2.4 Figure 8**

There are many aspects that I find puzzling or questionable in Figure 8. Firstly, the results do not correspond to what is shown in Figure S4, where the Q5% and Q95% are shown with the black error bars. For example, in the column  $\Delta GSAT=+3^{\circ}$ , Q95% of woCISM, woSSP245, and woSSP585 are clearly strongly different from the Q95% labeled "original" (Figure S4). But Figure 8 shows that these differences are  $\leq 1\%$ . I believe that there is an inconsistency here, or something that I misunderstand about Figure 8.

Secondly, I do no understand how it is possible that the changes in median and quantiles at  $\Delta GSAT = +4^{\circ}$  are so small for woSSP585. In this design experiment, the RF model has

presumably not even seen such levels of warming during training because the SSP 5-8.5 scenario has been excluded. But, by definition, tree models (including RF) predict slc based on decision rules seen during training. Thus, it is not clear how the RF can predict relatively similar slc values under  $\Delta GSAT=+4\circ$  when excluding SSP 5-8.5 as when it is not excluded. I am probably misunderstanding something here, but I believe that the authors should explain this counter-intuitive aspect of their results.

We thank Referee #2 for this careful analysis. We confirm that, for some experiments, there are some discrepancies that have revealed a bug in our scripts for the plotting. We have updated the results in the new manuscript.

**Minor comment 4: Some conclusions need to be put into perspective**

For different aspects, I find that better communication and/or more context about the conclusions is needed. I highlight some key examples here.

(a) Concerning  $\kappa$ , the authors argue for "the lesser importance of the choice in the range of the Greenland tidewater glacier retreat parameter" (L21). However, they compare it with the influence of the SSP scenario and of the ISM choice. It is expected that a single parameter should have much less influence than a global warming scenario and than a full ice sheet model. We thank Referee #2 for this comment. Referee #2 is invited to refer the clarification made above about  $\kappa$ : It is not thought as a parameter of the ice-flow model, it rather represents the sensitivity of the ocean forcing as a whole. It may be thought of as defining the sensitivity of the downscaling from global model to local ice sheet scale, similar to the combined parameter choices in RCMs for downscaling climate conditions. In the studied MME, we have different RCMs, which have different sensitivities and produce different melt for the same global forcing. Since we have only one approach to 'downscale' the ocean forcing,  $\kappa$  is sampling that uncertainty in a similar way.

In addition, our previous study (Rohmer et al., 2022) highlighted the high importance of  $\kappa$  compared to other uncertainties. Referee #2 is also invited to refer to our detailed reply to major comment 1 as well to Referee #1's comment 3.

**Reference**

Rohmer, J., Thieblemont, R., Le Cozannet, G., Goelzer, H., and Durand, G.: Improving interpretation of sea-level projections through a machine-learning-based local explanation approach, Cryosphere, 16, 4637–4657, https://doi.org/10.5194/tc-16-4637-2022, 2022.

(b) It should be better emphasized that the probabilistic ranges shown by the authors are not probabilistic projections of Greenland slc. Instead, they show a range of emulator predictions (thus conditioned on the emulator architecture) assuming a uniform distribution over the different inputs (L186). Thus, it does not represent calibrated uncertainty accounting for model-observation misfits (e.g., Aschwanden and Brinkerhoff, 2022). And neither does it represent the slc PDF from the MME, because the uniform distribution over the input space is not representative of the MME itself (e.g., the minimum spatial resolution is clearly not uniform between 1 and 40 km, see Fig. 3). As such, I believe that the true meaning of the PDFs shown in Figure 5 should be explained explicitly in order to avoid any reader misinterpreting those PDFs.

We agree with this comment. To do so, we propose to clarify the caption of the new Fig. 5 as well as the description in the new Sect. 2.4.2 on page 11, lines 227-230.

(c) The authors make a conclusion on "the utmost importance of including the SSP5-8.5 scenario, due to the large number of simulations available and the range of global warming they cover" (L19-20). However, I do not think that the authors have proven the co-existence of these two points. For example, could it be that including only a few training simulations with high global warming forcing would be sufficient to drastically decrease the errors of woSSP585 shown in Figure 7? In other words, maybe the emulator needs only a few high-warming training examples to correctly interpolate in the existing range of warming scenarios. Or maybe, as the authors write (L19-20), it is also the high number of experiments that is important. However, as far as I understand, the results presented in this study do not allow to evaluate the relative importance of these two aspects.

We agree with Referee #2: we believe that the influence of the MME size is shown by our results, but disentangling this effect from the range of global warning remains too complicated at least with the procedure proposed here.

More broadly, we have nuanced our conclusions in this sense (Sect. 5) by outlining that our results depend on the considered MME. The title has also been modified in this sense, i.e., "Lessons for multi-model ensemble design drawn from emulator experiments: application to a large ensemble for future sea level contributions of the Greenland ice sheet."

**References**

Andy Aschwanden and DJ Brinkerhoff. Calibrated mass loss predictions for the greenland ice sheet. Geophysical Research Letters, 49(19):e2022GL099058, 2022.

Nicolai Meinshausen and Greg Ridgeway. Quantile regression forests. Journal of machine learning research, 7(6), 2006.

Lucas Mentch and Giles Hooker. Quantifying uncertainty in random forests via confidence intervals and hypothesis tests. Journal of Machine Learning Research, 17(26):1–41, 2016.

Christopher KI Williams and Carl Edward Rasmussen. Gaussian processes for machine learning, volume 2. MIT press Cambridge, MA, 2006.

We thank Referee #2 for the suggested references.

Orleans, July 4th, 2025 J. Rohmer1 on behalf of the co-authors

1 BRGM, 3 av. C. Guillemin - 45060 Orléans Cedex 2 – France

---

## Referee Report (RR1)

Review of "Lessons for multi-model ensemble design drawn from emulator experiments: application to a large ensemble for future sea level contributions of the Greenland ice sheet" by Rohmer et al. Reviewer: Vincent Verjans

This is my second review of this manuscript, following the first round of revisions. First, I note the positive aspects of the revisions.

- Using quantile random forest (qRF) regression is a great implementation. This allows to evaluate not only point estimates of emulator predictions, but also their range, which characterizes emulator uncertainty, and increases/decreases of this uncertainty across experiments.

- The authors have clarified that the Multi-Model Ensemble (MME) is taken from the study of Goelzer et al. (2025). Although the study of Goelzer et al. (2025) is still undergoing peer-review, once it has been validated and published, it will serve as a necessary foundation for the present work.

- The wording now includes more nuance about most of the results being specific to this particular MME, and to the particular emulator used.

- Numerous questionable aspects (e.g., previous $D_{\mathrm{h}}$ calculation, errors in figures, etc.) have been corrected.

On the other hand, I believe that the revised manuscript still has major shortcomings, in particular in terms of lack of clarity about the methods used, and interpretation of the results. I hope that my comments will help address these issues. My review is separated into three General comments, followed by Specific comments. Line numbers refer to the revised manuscript without tracked changes.

**General comment 1: Arguments supporting the conclusions should be better grounded in the results**
One of the main conclusions from this study is "the importance of having diverse ISM and RCM models" (L23). In particular, the performance analysis (Figs. 7, 8) shows the decrease in performance when excluding MAR (woMAR) or excluding CISM (woCISM). However, this does not necessarily argue for the importance of diversity in RCMs and ISMs. In my view, it only shows that extrapolation errors of the emulator to unseen RCMs and ISMs are high. For example, $Q^2$ relative differences are higher for experiment woCISM than experiment CISM (Figs. 7, 8), while the former includes 3 ISMs and the latter only includes 1. Thus, while experiment woCISM has a more "diverse" set of ISMs than experiment CISM, its performance is worse: this is in direct contradiction with the conclusion about importance of diversity. Instead, in my view, this worse performance is a consequence of woCISM only including 33.5% of the MME simulations versus 66.5% for CISM (Table 2). Thus, training on a smaller set of simulations is the likely cause of decreased performance.

More generally, one of the conclusions is that having different ISMs is beneficial, while training on just a few $\kappa$ values is inconsequential. And I agree on this point. Let's take a very simple example of a very small MME to predict one unseen case.

○ The study shows that for the task of emulating *slc* from the hypothetical configuration (ISM=Elmer, $\kappa$=0.5), it is more useful for the emulator to be trained on the 4-member MME set {(CISM, 0.1), (CISM, 0.9), (Elmer, 0.1), (Elmer, 0.9)} than to be trained on the 4-member MME set {(CISM, 0.1), (CISM, 0.5), (CISM, 0.9), (Elmer, 0.1)}.

○ However, the study does not show that for predicting *slc* from the hypothetical configuration (GISM, 0.9), training the emulator on the 3-member MME set {(CISM, 0.1), (Elmer, 0.1), (IMAUICE, 0.1)} is more useful than training on the 3-member MME set {(CISM, 0.1), (CISM, 0.5), (CISM, 0.9)}, since GISM is absent from both training sets.

I hope that this example makes the point, despite its simplicity. The key is that the ISM to be emulated is present versus absent in the training set. Note that the exact same argument can be made concerning the "diversity" of RCMs instead of ISMs.

In short, what I try to communicate is that, despite the conclusions from the authors, there is no evidence that higher diversity leads unconditionally to better emulation performance. Instead, performance seems highly sensitive to the training set size, and also depends on the emulation target: it is easier to extrapolate to an unseen $\kappa$ value than to an unseen ISM. This is not a surprise: the qRF cannot predict results from RCMs or ISMs that it has not seen during training (see General comment 3 below). But no evidence is given that this prediction capability increases with the diversity of RCMs/ISMs used during training, as long as the unseen RCM/ISM is not included. Yet, this is one of the key messages from the manuscript, or, at least, this is how it is communicated.

Therefore, the key takeaways should be grounded in metrics that directly support the specific message that is communicated. They should not be expressed as general conclusions if this generality has not been verified. Alternatively, stronger evidence should be provided to justify the general claims, accompanied by clear explanations grounded in quantitative results.

**General comment 2: Lack of clarity regarding the evaluation procedure**
I honestly have a lot of difficulties to understand exactly how the emulator performance has been evaluated (Sect. 2.4.1).

(a) Are the $n_{\text{test}}$ test samples excluded from the emulator training in each of the 25 iterations of the validation procedure (i.e., are they truly unseen)? This needs to be explicitly specified.

(b) Is the validation procedure performed separately for each of the experiments described in Table 2? Fig. 7 suggests that this is the case, but it needs to be explicitly specified.

(c) Similarly, is the validation procedure performed using the full MME? Fig. 5 suggests that this is the case, but it needs to be explicitly specified.

(d) When validating the emulator for the specific experiments (shown in Fig. 7), are simulations excluded from both the training and test samples, or only from the training samples? For example, for evaluation of woMAR, are all MAR simulations excluded from the training and test samples or only from the training samples?

(e) In my previous review, I raised the question about why not using the traditional 10-fold cross-validation procedure rather than the ad-hoc validation procedure prescribed here. I am unconvinced by the response of the authors: "The reason for proposing an alternative validation procedure is to make sure to reflect the ability of the emulator to perform well over a wide range of GSAT values instead of randomly selected cases".
In 10-fold cross validation, each simulation of the MME is left-out exactly once. As such, the evaluation cases are not "randomly selected cases", they are all the cases. After performing cross-validation, it is easy to aggregate results according to their GSAT range, as is the case in Fig. 5. Note that I may be misunderstanding something here, which is why I ask the

authors to add one or two sentences of clear explanation for why their validation procedure is better "adapted to our objective" (L199).

(f) Figs. 7 and 8 show relative differences in performance. Is this relative to the metric for the exact same test sample as evaluated in the full-MME evaluation? In which case, it means that the $25 \times 200$ $n_{\text{test}}$ test samples are shared across all the validation experiments? Or is it "relative" to some other quantity? This needs to be specified.

(g) It is nowhere specified if the evaluation process for the experiments (e.g., woMAR, MAR, woCISM, etc.) is with respect to the full distribution of results from the MME, or only to those specific simulations left out for the particular experiment (e.g., only the MAR simulations for woMAR, only the RACMO and HIRHAM simulations for MAR, only the CISM simulations for woCISM, etc.). As I understand it, the evaluation is with respect to the full MME distribution, but this needs to be explicitly specified.

Also, concerning the results from the evaluation procedure, more clarity or emphasis is required for some key points.

(h) Figs. 7 and 8 show very large relative differences in the metrics. In particular, the $Q^2$ relative difference often exceeds 100%. If I understand correctly, this implies that $Q^2 \leq 0$. This means that the emulator performs worse than simply predicting the mean as a constant output prediction. In other words, the emulator is worthless in such situations.
This is a critical point, which is completely omitted in the manuscript. I recommend that the authors draw a vertical line at the 100% value in Fig. 7b and Fig. 8b,e to emphasize this. They should also mention and discuss this in the main text.

(i) The authors correctly point out that "the RF emulator should be used cautiously over the range of GSAT values around 3°C" (L251). This is a critical point that should be discussed in the Synthesis and Discussion section, and mentioned in the Abstract (e.g., (...) the emulator performance is unsatisfactory at intermediate levels of warming ($\sim 3$°C) (...)).

**General comment 3: Lack of details concerning the emulator**
In my previous review, I asked for more clarification about the emulator. I thank the authors for the additional information included in the manuscript, but I still believe that some critical details are omitted. I raise this as an important concern, because I believe that some of these aspects might have impactful consequences on the emulator results.

Taking the example of the woMAR experiment, the qRF is trained with RCM cases of HIRHAM and RACMO only. The qRF is constructed using decisions at each split, and some splits may separate based on HIRHAM versus RACMO. Then, at prediction time, if the emulator is tested with a MAR case, how can it make a prediction? In other words, how are unseen categories handled at prediction time?

For this reason, it seems strange to me to emulate *slc* from unseen RCMs or unseen ISMs. I take here two examples from recent emulation efforts of ISMIP6 where the emulator was not designed to predict output from unseen ISMs. First, Edwards et al. (2021) take the full ensemble as an emulation target. In contrast to this study, the emulator of Edwards et al. (2021) does not use the ISM as input to the emulator. Instead, their emulator is trained to predict the ensemble response across ranges of GSAT and $\kappa$ values, not the response of a specific ISM (their nugget term accounts for inter-ISM differences). Second, Seroussi et al. (2023) emulate specific missing GCM-ISM experiments. However, they only emulated those missing experiments for which some other experiments using the target ISM were available. As such, inputting the ISM as a predictor

variable for a new prediction was actually meaningful, because it could be associated with samples from the training set. In summary, it is important to explain how unseen categories (e.g., ISMs, RCMs) are handled at prediction time, as this would partly illuminate the interpretation of the prediction performances shown in Figs. 7 and 8.

As a side note, I understand that the authors want to make the article as easy as possible to follow for non-experts, and as such move details to Appendices or omit them entirely. However, in my opinion, the level of technical detail in the main text is insufficient. For example, Sect. 2.2 reads more as an introductory paragraph to qRF regression rather than a description of the emulation process. Sects. 2.4.1 and 2.4.2 also lack the necessary detail to really understand the results shown in Figs. 5,6,7,8. I believe that the editor needs to agree that technical details are quasi absent in some important sections of the main text.

**Specific comments**
Title. Replace "future" by: 2100
L16-18. This sentence should end with a question mark.
L18-19. Specify: (...) to build a random-forest-based emulator of 2100 Greenland sea-level rise contribution (...).
L36-37. Please rephrase: one member cannot span, it is the MME that should span.
L56. In this paragraph, please also refer to the work of Seroussi et al. (2023), which is highly relevant to this study. L71-72. In this sentence, the word "experiments" is used twice to designate two different notions. This can be confusing.
L87-89. This statement needs a citation.
L95. "surface mass balance (SMB) changes" should be: surface mass balance (SMB) anomalies.
L123. Here and in the remainder of the manuscript, why is the wording "credibility interval" used instead of confidence interval? The former suggests that some Bayesian modeling has been performed. Please consider re-wording.
Caption of Fig. 2. Please specify the confidence interval corresponding to the likely range.
L133. Please rephrase because Elmer/Ice is not "more frequent than others".
L134. The minimum resolution of 16 km does not appear in Fig. 4. There is a bar at 20 km, and the most frequent seems to be at 8 km.
L177-179. At the end of this sentence, a brief sentence should be formulated to specify explicitly if the objective is then to evaluate if the emulators constructed from the reduced MME are capable of (i) reproducing the distribution of results from the original MME, or (ii) reproduce the results that have been left-out from the original MME.
L194-197. This last sentence of the 1st paragraph should be moved elsewhere. This Sect. 2.4.1 already includes very little details, so the space dedicated to it should be focused on explaining the performance evaluation procedure.
L201-203. "for each interval, 50 samples are randomly selected. For one iteration of the procedure, a total of $n_{test}$=200 test samples are randomly selected". I find this phrasing somewhat confusing. I recommend rephrasing: for each interval, 50 samples are randomly selected, resulting in a total of $n_{test}$=200 test samples.
L204. Specify if the emulator is trained in each iteration on all the MME simulations, except the $n_{test}$ test samples (see General comment 2).
L205. "mean relative error" should be: mean relative absolute error.
L214. How many samples are drawn for the Monte-Carlo random sampling procedure?
Figure 5. The CRPS is a good metric, but not very intuitive (e.g., what does it mean if CRPS is 0.0025?). I believe that it would also be insightful to evaluate if the emulator is under-dispersed, over-dispersed, or well-calibrated. A common and intuitive metric for this is the spread-error ratio (e.g., Stephenson and Doblas-Reyes, 2000). I think that it would add a lot to the analysis to quantify first the calibration of the emulator (in Fig. 5), and second if the emulator tends to become over- or under- dispersed in the experiments (Figs. 7 and 8). Note that since the qRF does not provide standard deviation of the prediction, the spread-error ratio can be approximated as $\frac{\sigma}{\text{RMSE}} \approx \frac{Q_{75} - Q_{25}}{1.35\,\text{RMSE}}$.

Caption of Figure 5. Please specify explicitly that the performance statistics are computed over test samples unseen during emulator training.

L239-240. Please specify here if this is performed using the full MME (in contrast to the reduced MME used for the experiments of Table 2).

Caption of Figure 6. Typo: constructed using the Monte-Carlo based procedure. Also, please specify the confidence interval corresponding to the likely range.

Caption of Figure 7. Please specify explicitly that the performance statistics are computed over test samples unseen during emulator training.

L273. Please remove "Interestingly", as it is preferable to let readers decide what they find interesting.

L278. "twice that of the third most important contributor, i.e., woCISM". The medians of RAE relative difference are very close. Please be more specific in quantification of the performance.

Caption of Figure 8. Please specify explicitly that the performance statistics are computed over the same test samples as in Figure 7.

Figure 8. These results show that (i) excluding SSP126 and SSP245 has negligible impact for predicting in the GSAT range $\geq 3.83°C$, and (ii) excluding SSP585 has negligible impact for predicting in the GSAT range $\leq 2.14°C$. This should be mentioned and discussed briefly in the text.

L285. Please rephrase: "has the largest impact almost at the same level".

L287. Mention to Table 1 is wrong.

L288. Please remove "it is interesting".

L289. "The analysis of the other GSAT intervals" should be: The analysis of the GSAT interval 3.34 to 3.83°C.

L295. Here, I believe that this analysis applies to the random samples drawn as explained in Sect. 2.4.2, and not to the test samples explained in Sect. 2.4.1. Please specify this explicitly.

L305. "under-estimated by more than 25%": Fig. 9 shows $\sim 22\%$.

L307. Please remove "Interestingly".

L307-311. How can this contrasting result be explained? A perfect performance would mean that all quantiles remain unchanged. As such, why do larger changes in quantiles do not lead to worse performance? The authors should explain this.

Caption of Figure 9. Please specify that these results are computed from the random samples as explained in Sect. 2.4.2 and not from the validation procedure (if I understood correctly).

Table 3 (row SSP-RCP). Please specify that "the strong linearity of the Greenland ice sheet response with global temperature" is valid for the 2100 timescale.

Table 3 (row ISM choice). I believe that the under- versus over-estimation depends on the specific ISM that is excluded. For example, if the CISM model predicts consistently higher $slc$ values than other ISMs, then the experiment CISM would over-estimate the left-out $slc$ values (as it is the case here). However, if the CISM model was predicting consistently lower $slc$ values, then experiment CISM would lead to under-estimation of the left-out $slc$ values. And reciprocally for the experiment woCISM. Therefore, I do not believe that such a general conclusion can be made about over- versus under-estimation (this links to General comment 1). Similarly in the Abstract, the word "under-estimations" may be misleading.

L335. "Here, 'woMAR' is not necessarily the highest contributor to the changes". Please explain this (see comment about L307-311).

L342. Please explain the reasons in the MME design that explain why woCISM leads to a larger perturbation of the member distribution than woMAR (e.g., experiments of a specific SSP scenario have only been done with the CISM model, etc.).

L345. "further work should look into this aspect in more detail". This should be done as part of this study.

L360. "This also relates to the question of initialisation (and initial mass loss estimates) where the RCM choice is a key ingredient (e.g., Otosaka et al., 2023)". It is unclear to me what this sentence implies, and which message the authors try to convey.

L363. "First, our study contributes (...) according to the same report". I do not see how this is a contribution of this study. Here, the authors simply provide the Greenland sea-level rise contribution estimates of the ICCP.

L390-392. "Indeed, scenarios based on global warming levels can be potentially better understood by stakeholders than the SSP or RCP scenarios, and also allow users to better make the link with the climate objectives set out in the Paris agreement to stabilize climate change well below 2°C GWL". This reads as a personal opinion of the authors, so please rephrase or remove.

L399. Please specify: future sea level by 2100.

L400. Please specify: high importance for emulator accuracy.

L417-421. This sentence is too long and confusing.

Equation A1. What does $s$ represent in this equation?

L609. "where I(A) is the indicator operator". The parenthesis notation is not used in Equation A2.

L612 and L615. These two sentences repeat the same information.

L624. Please use consistently $q_\tau$ or $q^\tau$.

Equation B1. Please define inf as the infimum function.

L628. Typo: "weighed".

L637. Please specify that Step 2.2 approximates the CDF of $slc|\tilde{x}$ (if I understand correctly).

L638. If I understand correctly, $Q_{\tilde{u}}^\alpha$ is not specific to a single $\tilde{u}$ sample, but depends on the full set of $\tilde{u}$'s sampled in Step 2.1. This is in contrast to $q^{\tilde{u}}(slc|\tilde{x})$ in Step 2.2. If this is correct, then this notation is confusing, and I recommend writing $\tilde{Q}^\alpha$ instead.

L642. This "variability" corresponds to the emulator uncertainty about a given quantile level $\alpha$. But, if I understand correctly, the range $\left[Q^{\alpha/2} : Q^{1-\alpha/2}\right]$ gives the $1 - \alpha$ confidence interval of the emulator prediction for $slc|\tilde{x}$. If this is correct, please specify it.

L647. Please specify that the p-value quantifies how unlikely the variable importance in the non-permuted data is with respect to the null distribution of variable importance reached from the permutations.

L657. Typo: "should retained".

Figures S2, S3, and S4. Thes figures are identical. Is this an error?

**References**

Tamsin L. Edwards, Sophie Nowicki, Ben Marzeion, et al. Projected land ice contributions to twenty-first-century sea level rise. *Nature*, 593(7857):74–82, May 2021. ISSN 1476-4687. doi: 10.1038/s41586-021-03302-y. URL http://dx.doi.org/10.1038/s41586-021-03302-y.

Hélène Seroussi, Vincent Verjans, Sophie Nowicki, et al. Insights into the vulnerability of antarctic glaciers from the ismip6 ice sheet model ensemble and associated uncertainty. *The Cryosphere*,

17(12):5197–5217, December 2023. ISSN 1994-0424. doi: 10.5194/tc-17-5197-2023. URL `http://dx.doi.org/10.5194/tc-17-5197-2023`.

David B. Stephenson and Francisco J. Doblas-Reyes. Statistical methods for interpreting monte carlo ensemble forecasts. *Tellus A: Dynamic Meteorology and Oceanography*, 52(3):300, January 2000. ISSN 1600-0870. doi: 10.3402/tellusa.v52i3.12267. URL `http://dx.doi.org/10.3402/tellusa.v52i3.12267`.

---

## Referee Report (RR2)

Review of "Lessons for multi-model design drawn from emulator experiments: application to a large ensemble for future sea level contributions of the Greenland ice sheet" by Rohmer et al.

This article is a revised version of a previously submitted paper. While some changes have been made to address my comments –for which I would like to thank the authors– I do not believe that the current version is ready for publication, as it does not yet meet the standards of clarity and readability required for publication. I continue to believe that the content of this article is noteworthy and valuable for the scientific community, but further work is needed to make it publishable.

My review is structured in two parts. First, in the general comments section, I address the major changes made by the authors in response to my original remarks. Then, I provide a list of specific comments. These are mostly minor suggestions and corrections, but they are relatively numerous. In particular, I have not gone through the entire text in detail. To facilitate the next round of revisions, I would therefore ask the authors to carefully proofread their revised manuscript, which would help streamline the revision and correction process. As a side note, the article would be more pleasant to read if the figures were included in vector format or, at least, in higher resolution, so that they do not appear blurry.

**General comments**

General comment 1. I would like to thank the authors for their efforts to reorganize the paper to make it more accessible to a broader audience.

General comment 2. I am pleased to see that a brief discussion on the weight attribution of each ensemble member is now included in the paper. However, unless I am mistaken, this appears primarily in the interpretation of results, and thus appears rather late in the main text (it is first mentioned on page 8). Evaluating the predictive quality of the members is crucial in the context of future ensemble studies; indeed, it is a key step in linking numerical simulations to observations and constraining the former using the latter. It is also a quite natural step from a Bayesian approach, as it allows for relaxing the assumption of a uniform prior. Therefore, it would make sense to introduce this question earlier in the main text, so that the reader clearly understands how each member is to be compared. I suggest adding such a discussion when the inputs are presented in subsection 2.1, at the end of page 5. The authors could also briefly mention it in the introduction when outlining the scope of the paper.

General comment 3. I would like to thank the authors for the additional details concerning the interpretation of the parameter κ. While I am not entirely convinced by the rationale of comparing a single parameter to a full forcing scenario choice, I am fine with the authors retaining this aspect of their analysis. I still wonder whether the meaning of this parameter could be made clearer by explicitly renaming it as 'ocean forcing' parameter instead of 'retreat' parameter, particularly if it is associated with uncertainties in ice–ocean coupling, rather than being a parameter intrinsic to the ice-sheet model itself. This would make it clearer that it represents a forcing. Furthermore, it seems to me that the structural uncertainty of the ice-sheet models has not been addressed in the paper. This omission might bias the results by underestimating the impact of uncertainties related to ice-sheet physics and models.

**Specific Comments**

Note: I am using the author's tracked changes document for the line numbers.

- (1) [Lines 32–33] '(IPCC: e.g. Lee et al., 2021)'  $\rightarrow$  '(IPCC; e.g., Lee et al., 2021).
- (2) [Line 36] '(e.g. Knutti et al., 2010)'  $\rightarrow$  '(e.g., Knutti et al., 2010)'.
- (3) [Line 37] The fact that each member evenly spans a representative set of plausible realizations

- is somewhat misleading, as this is only true if no additional information is not available (e.g., observations). This would benefit from further clarification; see general comment 2.
- (4) [Line 40] '(e.g. Merrifield et al., 2020)'  $\rightarrow$  '(e.g., Merrifield et al., 2020)'.
- (5) [Lines 52–55] This sentence is too long. Consider splitting it in two, maybe after the 'thoroughly'.
- (6) [Line 82] 'the main model parameter' is ambiguous: do you mean  $\kappa$ ?
- (7) [Line 94] Consider adding what you mean by 'as best as possible', as it is vague on its own. Maybe mention that the misfit between computed and observed surface velocities and/or thicknesses is minimized?
- (8) [Line 98] '(Slater et al., 2020, 2019)'  $\rightarrow$  '(Slater et al., 2019, 2020)'.
- (9) [Line 108] 'parameter values' is ambiguous: do you mean the ice-sheet parameter values?
- (10) [Lines 109–112] I would remove entirely the discussion about the merge of the two inputs and present your final setup more directly, namely, the use of a GSAT that corresponds to a combination of SSP-RCP and GCM. This would be easier to follow.
- (11) [Line 114] 'The inputs from the double line' is not clear. Do you mean the inputs below the double line?
- (12) [Table 1] Please be consistent in your system of notations. Some of the names start with capital letters (e.g., 'Sliding'), others do not (e.g., 'thermodin.').
- (13) [Table 1] 'thermodin'  $\rightarrow$  'thermodyn' or even 'thermo'.
- (14) [Table 1] One way to simplify the reading of the table would be to use math symbols to clarify whether the variables are categorical or continuous. For example, the ISM models would become {CISM, Elmer/Ice, GISM, IMAUICE}, while the resolution would become [1, 40] km.
- (15) [Line 118] 'as a particular'  $\rightarrow$  'a particular'.
- (16) [Line 121] 'expressed in meters sea level equivalent SLE' → 'expressed in meters sea level equivalent, SLE'.
- (17) [Figures 2–4] I think it would make more sense to first present the inputs (Figures 3 and 4) before the output (Figure 2).
- (18) [Figure 2] Vertical label: 'density'  $\rightarrow$  'PDF'.
- (19) [Figure 2] Caption: 'Probability density function of the sea level contribution in 2100 (with respect to 2014) from the Greenland ice-sheet (in cm seal level equivalent, SLE) based on the raw MME ensemble data considered in this study' → 'Probability density function of the sea level contribution of the Greenland ice sheet in 2100, with respect to 2014, based on the raw MME ensemble data considered in this study'.
- (20) [Line 132] 'highest importance' is ambiguous. Do you mean 'that contributes the most to the uncertainty'?

- (21) [Line 137] Consider adding one sentence that quickly explains why the design of experiments was indeed unbalanced.
- (22) [Figures 3, 4, C1 and C2] Consider drawing the plots with the count number in the y axis, as that is more common.
- (23) [Line 154] '(named emulator)' is not necessary here as you have already introduced several times the notion of emulator previously.
- (24) [Line 156] Consider putting the reference to the overview as new separate sentence.
- (25) [Line 169] I found the reference to a conditional mean to be not very clear. On what is the mean conditioned here?
- (26) [Line 184] What is meant here by 'tolerance'?
- (27) [Line 184] 'in Sect. 2.4'  $\rightarrow$  'in the next section'.
- (28) [Line 204] You mention AR6 here, but this has not been introduced/defined before.
- (29) [Line 209] Consider adding an adverb at the beginning of the sentence here (e.g., 'consequently'), so that it is clear that the 200 number is directly linked to the 50 number before, and not a new parameter for the validation exercise.
- (30) [Lines 212–218] Given that you introduce three performance criteria, you should use three distinct items, not two.
- (31) [Line 219] Consider removing 'for a GSAT scenario' from the title of the subsection. Both subsections 2.4.1 and 2.4.2 depend on the GSAT scenario.
- (32) [Line 266] [4.6; 7.4cm]  $\rightarrow$  [4.6 cm; 7.4 cm] or [4.6; 7.4] cm.
- (33) [Line 266] [10.4; 17.0cm]  $\rightarrow$  [10.4 cm; 17.0 cm] or [10.4; 17.0] cm.
- (34) [Figure 6] Vertical label: 'density'  $\rightarrow$  'PDF'.
- (35) [Line 269] 'constructed the Monte-Carlo-based procedure': there seems to be a missing word here.
- (36) [Line 281] 'whatever the performance criteria'  $\rightarrow$  'for every performance criterion'.
- (37) [Line 286] 'Table 1'  $\rightarrow$  Table 2'.
- (38) [Line 367] '(based on Goelzer et al. (2020))'  $\rightarrow$  '(based on Goelzer et al., 2020)'.
- (39) [Lines 410–411] '(Merrifield et al., 2023; Evin et al., 2019)' → '(Evin et al., 2019; Merrifield et al., 2023)'.
- (40) [Line 411] 'take'  $\rightarrow$  'took'.

- (41) [Line 609] You use *slc* with a superscript for the index, and then with a subscript later in the text. Please be consistent in your system of notations.
- (42) [Line 612] 'By nature'  $\rightarrow$  'By construction'.
- (43) [Line 615] 'squared errors': errors of what?

---

## Referee Report (RR4)

**Review of "Lessons for multi-model design drawn from emulator experiments: application to a large ensemble for 2100 future sea level contributions of the Greenland ice sheet" by Rohmer et al.**

This is the third round of review for this paper. I would like to thank the authors for their responses to my comments and for the changes made to the manuscript.

Given that the authors have responded positively to my major comments, I recommend that the manuscript be finalized for publication. I still ask the authors to carefully reread their manuscript, as there are still quite a lot of typos in the text (see specific comments below). Provided the authors also respond favorably to the more technical comments addressed by the first reviewer, I would recommend this paper for publication.

**Specific Comments**

Note: I am using the author's tracked-changes document for the line numbers.

(1) There is inconsistency in the use of hyphens for compound nouns used as adjectives; for example, you use both 'sea level contributions' and 'sea-level rise'. Please stick to one writing style throughout the text.

(2) When using mathematical symbols, please use italics. This is particularly relevant for Appendix A ('p' $\rightarrow$ '$p$', 'L' $\rightarrow$ '$L$', ...).

(3) Please avoid using fractions in running text. Either use a '/' symbol or write a full equation. For example, on line 810 you should write $RAE = (1/n_{\text{test}}) \sum_{i=1}^{n_{\text{test}}} |e^{(i)}/slc^{(i)}|$, or

$$ RAE = \frac{1}{n_{\text{test}}} \sum_{i=1}^{n_{\text{test}}} \left| \frac{e^{(i)}}{slc^{(i)}} \right|. \tag{R1} $$

(4) You define $slc$ in several places in the text and use it before it is properly defined. You should define it at the beginning of the manuscript once and for all.

(5) [Line 17] 'specific set' $\rightarrow$ 'a specific set' or 'specific sets'.

(6) [Line 20] 'warning' $\rightarrow$ 'warming'.

(7) [Line 74] '([...] by Edwards et al. (2010))' $\rightarrow$ '([...] by Edwards et al., 2010)'.

(8) [Line 78] 'as well' $\rightarrow$ 'as well as'.

(9) [Line 92] '(SSP126, SSP245, SSP585)' $\rightarrow$ '(SSP1-2.6, SSP2-4.5, SSP5-8.5)'.

(10) [Table 1] Follow-up on my previous comment (14): I am not asking to remove the categorical aspect in the column; rather, my suggestion was to use the symbols $\{a,b\}$ and $[a,b]$ to establish this distinction efficiently. Such notation is used in other multi-ensemble studies.

(11) [Line 148] '([...] by Aschwanden and Brinkerhoff (2022))' $\rightarrow$ '([...] by Aschwanden and Brinkerhoff, 2022)'.

(12) [Figure 4] Why is there an '(a)' at the beginning of the caption?

(13) [Figure 4] There is an inconsistency: you use a median of 7.9 cm SLE here but mention 8.7 cm SLE in the text.

(14) [Line 179] There is no need to mention '(named emulator)', as you have already described the concept of an emulator.

(15) [Line 185] '(sea level science, Tadesse et al. (2020); water resources, Tyralis et al. (2019); flood assessments, Rohmer et al. (2018))' → '(sea level science, Tadesse et al., 2020; water resources, Tyralis et al., 2019; flood assessments, Rohmer et al., 2018)'.

(16) [Line 185] 'Breiman et al. 1984' → 'Breiman et al., 1984'.

(17) [Line 188] 'Breiman et al. 1984' → 'Breiman et al., 1984'.

(18) [Lines 191–192] '(Hastie et al. (2009): chapter 9.2.4)' → '(Hastie et al., 2009; chapter 9.2.4)'.

(19) [Line 243] 'weather forecast' → 'weather forecasts'.

(20) [Line 257] 'GSAT scenario' → 'GSAT scenarios'.

(21) [Line 403] 'error-bars' → 'error bars'.

(22) [Line 404] 'quantile' → 'quantiles'.

(23) [Line 409] 'oppositive' → 'opposite'.

(24) [Line 677] 'Morlinghem' → 'Morlighem'.

(25) [Line 816] '(Bracher et al. (2021): Sect. 2.2)' → '(Bracher et al., 2021: Sect. 2.2)'.

(26) [Appendix E] 'Global Surface Atmosphere Temperature' → 'Global Surface Air Temperature'.

(27) [Appendix E] 'Inter-Sectoral Impact Model' (?) → 'Ice Sheet Model'.

(28) [Appendix E] 'Relative Absolue Error' → 'Relative Absolute Error'.

---

## Author Response (AR2)

**Replies to Referees' comments on "Drawing lessons for multi-model ensemble design from emulator experiments: application to future sea level contribution of the Greenland ice sheet" (egusphere-2025-52)**

We would like to thank the two Referees for taking the time to participate in this new round of reviews. We really appreciate the opportunity the Referees and the Editor have given to continue the revisions based on the various comments/suggestions. This is underlined in the acknowledgement section. We agree with most of the suggestions and, therefore, we have modified the manuscript to take on board the comments and suggestions. We recall the reviews and we reply to each of the comments in turn (outlined in green). The page and line numbers are those of the document with track-changes.

**Referee #1:**

Review of "Lessons for multi-model ensemble design drawn from emulator experiments: application to a large ensemble for future sea level contributions of the Greenland ice sheet" by Rohmer et al.

Reviewer: Vincent Verjans

We would like to thank Dr Verjans for taking the time to participate in this new round of reviews. We really appreciate the opportunity Dr Verjans has given to continue the revisions based on the various comments/suggestions.

In particular, we have taken care to:

- Provide sufficient details on the methods (Sect. 2.3 and 2.4), without using too technical description so that the article can be followed by non-experts, as suggested by Referee #2;
- Explain better how the RF emulator makes predictions with categorical inputs (Sect. 2.3) and add new performance scores;
- Nuance the conclusions regarding the "diversity" of the ISM and RCM model.

This is my second review of this manuscript, following the first round of revisions. First, I note the positive aspects of the revisions.

- Using quantile random forest (qRF) regression is a great implementation. This allows to evaluate not only point estimates of emulator predictions, but also their range, which characterizes emulator uncertainty, and increases/decreases of this uncertainty across experiments.
- The authors have clarified that the Multi-Model Ensemble (MME) is taken from the study of Goelzer et al. (2025). Although the study of Goelzer et al. (2025) is still undergoing peer-review, once it has been validated and published, it will serve as a necessary foundation for the present work.
- The wording now includes more nuance about most of the results being specific to this particular MME, and to the particular emulator used.
- Numerous questionable aspects (e.g., previous Dh calculation, errors in figures, etc.) have been corrected.

On the other hand, I believe that the revised manuscript still has major shortcomings, in particular in terms of lack of clarity about the methods used, and interpretation of the results. I hope that my comments will help address these issues. My review is separated into three General comments, followed by Specific comments. Line numbers refer to the revised manuscript without tracked changes.

**General comment 1: Arguments supporting the conclusions should be better grounded in the results**

One of the main conclusions from this study is "the importance of having diverse ISM and RCM models" (L23). In particular, the performance analysis (Figs. 7, 8) shows the decrease in performance when excluding MAR (woMAR) or excluding CISM (woCISM). However, this does not necessarily argue for the importance of diversity in RCMs and ISMs. In my view, it only shows that extrapolation errors of the emulator to unseen RCMs and ISMs are high. For example, Q² relative differences are higher for experiment woCISM than experiment CISM (Figs. 7, 8), while the former includes 3 ISMs and the latter only includes 1. Thus, while experiment woCISM has a more "diverse" set of ISMs than experiment CISM, its performance is worse: this is in direct contradiction with the conclusion about importance of diversity. Instead, in my view, this worse performance is a consequence of woCISM only including 33.5% of the MME simulations versus 66.5% for CISM (Table 2). Thus, training on a smaller set of simulations is the likely cause of decreased performance.

More generally, one of the conclusions is that having different ISMs is beneficial, while training on just a few  $\kappa$  values is inconsequential. And I agree on this point. Let's take a very simple example of a very small MME to predict one unseen case.

- The study shows that for the task of emulating slc from the hypothetical configuration (ISM=Elmer, κ=0.5), it is more useful for the emulator to be trained on the 4-member MME set {(CISM, 0.1), (CISM, 0.9), (Elmer, 0.1), (Elmer, 0.9)} than to be trained on the 4-member MME set {(CISM, 0.1), (CISM, 0.5), (CISM, 0.9), (Elmer, 0.1)}.
- However, the study does not show that for predicting slc from the hypothetical configuration (GISM, 0.9), training the emulator on the 3-member MME set {(CISM, 0.1), (Elmer, 0.1), (IMAUICE, 0.1)} is more useful than training on the 3-member MME set {(CISM, 0.1), (CISM, 0.5), (CISM, 0.9)}, since GISM is absent from both training sets.

I hope that this example makes the point, despite its simplicity. The key is that the ISM to be emulated is present versus absent in the training set. Note that the exact same argument can be made concerning the "diversity" of RCMs instead of ISMs.

In short, what I try to communicate is that, despite the conclusions from the authors, there is no evidence that higher diversity leads unconditionally to better emulation performance. Instead, performance seems highly sensitive to the training set size, and also depends on the emulation target: it is easier to extrapolate to an unseen  $\kappa$  value than to an unseen ISM. This is not a surprise: the qRF cannot predict results from RCMs or ISMs that it has not seen during training (see General comment 3 below). But no evidence is given that this prediction capability increases with the diversity of RCMs/ISMs used during training, as long as the unseen RCM/ISM is not included. Yet, this is one of the key messages from the manuscript, or, at least, this is how it is communicated.

Therefore, the key takeaways should be grounded in metrics that directly support the specific message that is communicated. They should not be expressed as general conclusions if this

generality has not been verified. Alternatively, stronger evidence should be provided to justify the general claims, accompanied by clear explanations grounded in quantitative results.

We thank Dr Verjans for this useful analysis. This is also helpful for better framing our message.

We have now nuanced the message by specifying that the results are related to the considered MME and to the particular ISM / RCM model used in our study. These corrections have been applied in the abstract, in Sect. 4, and in Table 3.

We agree that the results do not rigorously demonstrate the added value of having diverse RCM or ISM models. Therefore, we propose to remove any reference to "diversity" and to focus the analysis on the results of the experiments and in particular CISM and MAR experiments and their implications.

The description in Sect. 4.1 (on page 21, lines 432-466) has been reworked as follows: "On the other hand, some other conclusions could not necessarily have been anticipated in detail more particularly the implications on the percentile assessment (Sect. 3.3). Our results show that the magnitude of the influence depends on the GSAT scenario considered, the performance criterion and the target percentile level. For the high GSAT scenario, the exclusion of SSP5-8.5 has as much impact as the exclusion of MAR on emulator performance, and is even the biggest contributor to changes in the high percentiles. For the low GSAT scenario, excluding CISM has as much impact as excluding MAR on the emulator performance, and contributes most to changes in the low percentiles. The decrease in MME size induced by 'woCISM' and 'woSSP585' is smaller than that induced by 'woMAR', on the order of 70%, suggesting that it is not only a problem of 'size' but also a problem of the type of information that is removed from the set. Figure 11c shows that, when applying 'woSSP585' experiment, the emulator is learned with slc spanning a restricted range lower than that of the original MME. This means that the emulator is built with little information on large slc values, and to predict cases associated to high GSAT scenarios, the RF model mainly relies on extrapolation. This is a situation where emulator methods such as RF can fail completely; see e.g., Buriticá & Engelke (2024). Analysis of Figs. 11a and b helps to understand why "woMAR" and "woCISM" induce roughly equivalent changes for the 2°C GSAT scenario, as the slc CDF appears to be similarly disrupted by the application of these experiments with a CDF shifted towards low-to-moderate slc values, particularly in the slc range of  $\sim$ 5 to  $\sim$ 15cm. This means that the emulators are built on members whose *slc* values span approximately the same range.

New Figure 11: Comparison between the Cumulative Distribution Function (CDF) of *slc* in 2100 of the original MME (reference) and of the reduced MME after application of the emulator experiments, 'woMAR' (a), 'woCISM' (b), 'woSSP585' (c).

The oppositive experiments that consist in using MME restricted to members to a specific ISM or a particular RCM, here CISM or MAR respectively, are also informative. Although the corresponding emulator experiments imply a reduction of less than 30% of the MME size, the decline in emulator performance or changes in percentiles cannot be considered negligible. Our interpretation is that this effect is related to the importance of the information removed, i.e., the configurations of all input variables associated with the removed members, which is necessary for the RF emulator to make predictions for levels of categorical variables not seen in the training dataset, i.e., ISMs/RCMs, as explained in Section 2.3.

The interaction between the reduction in the size of the MME and the type of information important for the training of the emulator is however complex to analyse due to the multiple joint effects to be taken into account between the inputs. From a methodological viewpoint, this calls for further developments, in particular by relying on the data valuation domain (Sim et al., 2022). These types of tools aim to study the worth of data in machine learning models based on similar methods as the ones used by Rohmer et al. (2022) in the context of sea level projections. Transposed to the MME context, these tools could be used in future studies to assess the impact of each member in the emulator's predictions, i.e. the worth of each member. From a broader perspective on collaborative research, these results on the influence of RCM and ISM models can be seen as an additional justification for intensifying the model intercomparison efforts initiated in the past, in particular ISMIP6 (Nowicki et al., 2016), which included coupled ISMs as well as stand-alone ISMs in CMIP for the first time. They also support, to some extent, a posteriori, the choices that have been made for the construction of the MME considered here (based on Goelzer et al., 2020)".

**General comment 2: Lack of clarity regarding the evaluation procedure**

I honestly have a lot of difficulties to understand exactly how the emulator performance has been evaluated (Sect. 2.4.1).

- (a) Are the ntest test samples excluded from the emulator training in each of the 25 iterations of the validation procedure (i.e., are they truly unseen)? This needs to be explicitly specified.
- (b) Is the validation procedure performed separately for each of the experiments described in Table 2? Fig. 7 suggests that this is the case, but it needs to be explicitly specified.
- (c) Similarly, is the validation procedure performed using the full MME? Fig. 5 suggests that this is the case, but it needs to be explicitly specified.
- (d) When validating the emulator for the specific experiments (shown in Fig. 7), are simulations excluded from both the training and test samples, or only from the training samples? For example, for evaluation of woMAR, are all MAR simulations excluded from the training and test samples or only from the training samples?
- (e) In my previous review, I raised the question about why not using the traditional 10-fold cross-validation procedure rather than the ad-hoc validation procedure prescribed here. I am unconvinced by the response of the authors: "The reason for proposing an alternative validation procedure is to make sure to reflect the ability of the emulator to perform well over a wide range of GSAT values instead of randomly selected cases". In 10-fold cross validation, each simulation of the MME is left-out exactly once. As such, the evaluation cases are not "randomly selected cases", they are all the cases. After performing cross-validation, it is easy to aggregate results according to their GSAT range, as is the case in Fig. 5. Note that I may be misunderstanding something here, which is why I ask the authors to add one or two sentences

of clear explanation for why their validation procedure is better "adapted to our objective" (L199).

- (f) Figs. 7 and 8 show relative differences in performance. Is this relative to the metric for the exact same test sample as evaluated in the full-MME evaluation? In which case, it means that the  $25 \times 200$  ntest test samples are shared across all the validation experiments? Or is it "relative" to some other quantity? This needs to be specified.
- (g) It is nowhere specified if the evaluation process for the experiments (e.g., woMAR, MAR, woCISM, etc.) is with respect to the full distribution of results from the MME, or only to those specific simulations left out for the particular experiment (e.g., only the MAR simulations for woMAR, only the RACMO and HIRHAM simulations for MAR, only the CISM simulations for woCISM, etc.). As I understand it, the evaluation is with respect to the full MME distribution, but this needs to be explicitly specified.

We reply here to comments (a) - (g). To clarify the structure, we have added more details about the implementation of the validation procedure for testing the emulator performance (Sect. 2.4.1, on pages 10-11, lines 216-254) and about the implementation of the Monte-Carlo procedure for estimating the quantiles (Sect. 2.4.2, on pages 11-12, lines 255-268).

Also, concerning the results from the evaluation procedure, more clarity or emphasis is required for some key points.

(h) Figs. 7 and 8 show very large relative differences in the metrics. In particular, the  $Q^2$  relative difference often exceeds 100%. If I understand correctly, this implies that  $Q^2 \le 0$ . This means that the emulator performs worse than simply predicting the mean as a constant output prediction. In other words, the emulator is worthless in such situations. This is a critical point, which is completely omitted in the manuscript. I recommend that the authors draw a vertical line at the 100% value in Fig. 7b and Fig. 8b,e to emphasize this. They should also mention and discuss this in the main text.

We thank Dr. Verjans for this useful comment. As suggested, we have added comments on this aspect in the description of the results in Sect. 3.2 as well as in the caption of Fig. 7 and 8.

(i) The authors correctly point out that "the RF emulator should be used cautiously over the range of GSAT values around  $3 \circ C$ " (L251). This is a critical point that should be discussed in the Synthesis and Discussion section, and mentioned in the Abstract (e.g., (...) the emulator performance is unsatisfactory at intermediate levels of warming ( $\sim 3 \circ C$ ) (...)).

We have specified in the abstract (on page 1, lines 18-21) that "We use these experiments to build a random-forest-based emulator, which shows high predictive capability for assessing 2100 Greenland sea-level rise contributions for low and very high levels of warning".

We have also identified more clearly this aspect as a line for improvement in Sect. 5 (on page 25, lines 537-543) as follows: "Second, our results are based on the use of an emulator, i.e., a statistical approximation of the 'true' chain of numerical models. The RF emulator trained in our study showed satisfactory predictive capabilities for low and high levels of warning (GSAT values of respectively 2 and 4°C). The emulator performance remained however unsatisfactory at intermediate levels of warming (3°C). Despite, the efforts made in our study to nuance the results by including indicators of the emulator uncertainty, the emulator training should be

improved in the future by considering alternative emulator models (e.g., Yoo et al., 2025) but also more robust approaches for hyperparameter tuning (Bischl et al., 2023), and more particularly more advanced categorical variables' encoding (Au, 2018; Smith et al., 2024), which is key to apply the proposed emulator experiments".

**Added references**

Au, T. C.: Random forests, decision trees, and categorical predictors: the" absent levels" problem. Journal of Machine Learning Research, 19(45), 1-30, 2018.

Smith, H. L., Biggs, P. J., French, N. P., Smith, A. N., and Marshall, J. C.: Lost in the Forest: Encoding categorical variables and the absent levels problem. Data Mining and Knowledge Discovery, 38(4), 1889-1908, 2024.

**General comment 3: Lack of details concerning the emulator**

In my previous review, I asked for more clarification about the emulator. I thank the authors for the additional information included in the manuscript, but I still believe that some critical details are omitted. I raise this as an important concern, because I believe that some of these aspects might have impactful consequences on the emulator results.

Taking the example of the woMAR experiment, the qRF is trained with RCM cases of HIRHAM and RACMO only. The qRF is constructed using decisions at each split, and some splits may separate based on HIRHAM versus RACMO. Then, at prediction time, if the emulator is tested with a MAR case, how can it make a prediction? In other words, how are unseen categories handled at prediction time?

For this reason, it seems strange to me to emulate slc from unseen RCMs or unseen ISMs. I take here two examples from recent emulation efforts of ISMIP6 where the emulator was not designed to predict output from unseen ISMs. First, Edwards et al. (2021) take the full ensemble as an emulation target. In contrast to this study, the emulator of Edwards et al. (2021) does not use the ISM as input to the emulator. Instead, their emulator is trained to predict the ensemble response across ranges of GSAT and  $\kappa$  values, not the response of a specific ISM (their nugget term accounts for inter-ISM differences). Second, Seroussi et al. (2023) emulate specific missing GCM-ISM experiments. However, they only emulated those missing experiments for which some other experiments using the target ISM were available. As such, inputting the ISM as a predictor variable for a new prediction was actually meaningful, because it could be associated with samples from the training set. In summary, it is important to explain how unseen categories (e.g., ISMs, RCMs) are handled at prediction time, as this would partly illuminate the interpretation of the prediction performances shown in Figs. 7 and 8.

As a side note, I understand that the authors want to make the article as easy as possible to follow for non-experts, and as such move details to Appendices or omit them entirely. However, in my opinion, the level of technical detail in the main text is insufficient. For example, Sect. 2.2 reads more as an introductory paragraph to qRF regression rather than a description of the emulation process. Sects. 2.4.1 and 2.4.2 also lack the necessary detail to really understand the results shown in Figs. 5,6,7,8. I believe that the editor needs to agree that technical details are quasi absent in some important sections of the main text.

As rightly pointed out, we are striving here to satisfy two contradictory (but justified!) demands, namely (1) to provide sufficient technical detail so that readers have all the information they need to analyse and understand the results; (2) not to overload the text with overly technical aspects that could hinder readability for non-experts and obscure the practical implications of the work. Therefore we proposed having technical appendices in the previous version.

On the one hand, we partly agree with Dr. Verjans regarding the description of performance scores. We believe that indicating the interpretation of scores rather than equations in the main text should serve our twofold objective.

On the other hand, we totally agree with Dr. Verjans regarding the problem of prediction for unseen categories. This is problem intrinsically connected to the use of random forest models, known as the "absent level" problem (Au, 2018). We propose to clarify this problem in the main text in a relatively non-technical way. To do so, we have re-organized Sect. 2, by first defining the 'emulator experiments' related to the design decisions, then by describing in more details the functioning of the random forest model, and the problem of prediction for unseen categories in relation to the emulator experiments.

Sect. 2.3 (page 9-10, lines 190-202) has been updated as follows: "A key aspect of our study is to be able to handle many categorical variables with large number of levels (unordered values). However, the partitioning algorithm described above tends to favour categorical predictors with many levels (Hastie et al. (2009): chapter 9.2.4). To alleviate this problem, we rely on the computationally efficient algorithm proposed by Wright and König (2019) based on ordering the levels a priori, here by their slc mean response. A second key aspect is to be able to predict for new levels of the categorical variables, since the emulator experiments defined in Sect. 2.1 involve leaving out specific members from the original MME assigned to a given model, RCM / ISM, or a given SSP-RCP scenario, i.e., some specific levels. This problem is related to the more general 'absent levels' problem for RF models (Au, 2018), which arises when a level of a categorical variable is absent when a tree is grown, but is present in a new observation for prediction. Here, the chosen ordering algorithm of Wright and König (2019) alleviates this problem: by treating the categorical variables as ordinal, levels not present at a given partition during the splitting procedure can still be assigned to a next partition in the next iteration by directing all observations with absent levels down the same branch of the tree (in our implementation, chosen as the "left" branch). In this manner, the observations with absent levels are kept together and can be split down the tree by another input variable".

Dr. Verjans is right to underline that our experiments test the extrapolation capability of the RF model by using it for predicting new categories. However, it is important to note that this type is not so "strange" because this is not a "pure" extrapolation problem: the members that are removed from the training by applying the emulator experiments share information, with those used for training via the other variables. The RF model relies on them to make the prediction as explained above. In our study, this means that the emulator experiments test whether the information left in the MME after removing specific members is sufficient to predict *slc* at a reasonable accuracy (Sect. 2.3, page 10, line 203).

In conclusion, we do not claim to overcome the difficult problem of "absent levels", which is inherent to the RF method. Therefore, we use a standard procedure implemented in several packages for RF modelling. Other options exist (Au, 2018; Smith et al., 2024), and we have clearly pointed out the need for investigating them as an avenue of this work in the concluding remarks (Sect. 5, page 25, lines 543); see also reply to comment (i) described above.

**Added references**

Au, T. C. (2018). Random forests, decision trees, and categorical predictors: the" absent levels" problem. *Journal of Machine Learning Research*, 19(45), 1-30.

Smith, H. L., Biggs, P. J., French, N. P., Smith, A. N., & Marshall, J. C. (2024). Lost in the Forest: Encoding categorical variables and the absent levels problem. *Data Mining and Knowledge Discovery*, 38(4), 1889-1908.

**Specific comments**

Title. Replace "future" by: 2100

This has been corrected.

*L16-18. This sentence should end with a question mark.*

This has been corrected.

L18-19. Specify: (...) to build a random-forest-based emulator of 2100 Greenland sea-level rise contribution (...).

This has been corrected.

L36-37. Please rephrase: one member cannot span, it is the MME that should span. This has been corrected as suggested.

L56. In this paragraph, please also refer to the work of Seroussi et al. (2023), which is highly relevant to this study.

We agree with Dr. Verjans. We now refer to this study in the introduction (page 3, lines 66-68).

L71-72. In this sentence, the word "experiments" is used twice to designate two different notions. This can be confusing.

This has been reformulated on page 3 (lines 74-78) as follows: "To address these questions, we take advantage of a large MME of Greenland ice sheet contributions to sea level this century, based on which we define a series of numerical experiments (referred to as emulator's experiments) that are closely related to practical MME design decisions. These experiments consist in leaving out specific results from the original MME assuming that all members have the same weight in the ensemble."

**L87-89. This statement needs a citation.**

We have added references to (Slater et al., 2019, 2020) and Rahlves et al. (2025).

**Added reference**

Rahlves, C., Goelzer, H., Born, A., and Langebroek, P. M.: Historically consistent mass loss projections of the Greenland ice sheet, The Cryosphere, 19, 1205-1220, https://doi.org/10.5194/tc-19-1205-2025, 2025.

L95. "surface mass balance (SMB) changes" should be: surface mass balance (SMB) anomalies.

This has been corrected.

L123. Here and in the remainder of the manuscript, why is the wording "credibility interval" used instead of confidence interval? The former suggests that some Bayesian modeling has been performed. Please consider re-wording.

We agree with Dr. Verjans. There is no notion of Bayesian modeling and we now refer to "confidence intervals".

Caption of Fig. 2. Please specify the confidence interval corresponding to the likely range. This has been corrected.

L133. Please rephrase because Elmer/Ice is not "more frequent than others". This has been corrected.

L134. The minimum resolution of 16 km does not appear in Fig. 4. There is a bar at 20 km, and the most frequent seems to be at 8 km.

This is a problem with the endpoint of the y-axis. This has been corrected. We thank Dr. Verjans for noticing this problem.

L177-179. At the end of this sentence, a brief sentence should be formulated to specify explicitly if the objective is then to evaluate if the emulators constructed from the reduced MME are capable of (i) reproducing the distribution of results from the original MME, or (ii) reproduce the results that have been left-out from the original MME.

This part (Sect. 2.2, page 9, lines 170-175) has been reformulated as follows: "To measure the influence of removing specific members from the original MME, we assess if the emulators constructed from the reduced MME are capable of reproducing the results of an emulator trained with the complete original MME, named the 'reference solution' in the following. We analyse the changes in two types of criteria: (1) emulator performance to predict *slc* in 2100 for input configurations unseen during the training; (2) probabilistic predictions for *slc* in 2100 given future GSAT change scenarios, here chosen at 2°C (+/- 0.5°C) or 4 °C (+/- 0.5°C). The details of this assessment are explained in Sect. 2.4".

L194-197. This last sentence of the 1st paragraph should be moved elsewhere. This Sect. 2.4.1 already includes very little details, so the space dedicated to it should be focused on explaining the performance evaluation procedure.

Section 2.4.1 has been fully re-written by providing more details on the performance evaluation. We chose to keep the comment on GSAT, because it is the motivation for the development of an evaluation procedure adapted to our objective. We hope that the description is now clearer.

L201-203. "for each interval, 50 samples are randomly selected. For one iteration of the procedure, a total of ntest=200 test samples are randomly selected". I find this phrasing somewhat confusing. I recommend rephrasing: for each interval, 50 samples are randomly selected, resulting in a total of ntest=200 test samples.

This part has been reformulated as suggested.

L204. Specify if the emulator is trained in each iteration on all the MME simulations, except the ntest test samples (see General comment 2).

The description of the procedure at the beginning of Sect. 2.4.1 clarifies this aspect.

L205. "mean relative error" should be: mean relative absolute error. This has been corrected.

L214. How many samples are drawn for the Monte-Carlo random sampling procedure? A total of 10,000 random samples are considered. This is now more clearly indicated in Sect. 2.4.2 (page 12, line 262).

Figure 5. The CRPS is a good metric, but not very intuitive (e.g., what does it mean if CRPS is 0.0025?). I believe that it would also be insightful to evaluate if the emulator is under-dispersed,

over-dispersed, or well-calibrated. A common and intuitive metric for this is the spread-error ratio (e.g., Stephenson and Doblas-Reyes, 2000). I think that it would add a lot to the analysis to quantify first the calibration of the emulator (in Fig. 5), and second if the emulator tends to become over- or under- dispersed in the experiments (Figs. 7 and 8). Note that since the qRF does not provide standard deviation of the prediction, the spread-error ratio can be approximated as  $\sigma/RMSE \approx Q75-Q25/1.35~RMSE$ .

We thank Dr Verjans for his valuable suggestion. Building on this idea of calibration, we propose two additional performance scores:

- The first criterion analyses the coverage of the prediction intervals at the level  $\alpha$ . If the coverage reaches the expected value of  $\alpha$ , the prediction interval can be considered reliable;
- The spread to error ratio with a formulation using the interquartile distance as proposed by Dr. Verjans.

We have however a concern with the equation proposed. The correction by 1.35 is valid assuming a normal distribution of the predictive distribution provided by the RF emulator. This is not necessarily the case here. In addition, we can find different formulations in the literature. For instance, Bellon-Maurel et al., (2010) proposes a formulation without correction.

We propose to use the formulation without correction and to clearly describe the interpretation used in our study (Sect. 2.4.1, page 11, lines 250-254): "the ratio of performance to the interquartile distance IQR (Bellon-Maurel et al., 2010) compares the emulator prediction uncertainty, measured by the difference between the 75th and the 25th quantiles - named interquartile distance, with the prediction error measured by the root mean square error. If  $IQR \approx 1$ , the interquartile distance provides valuable information about the prediction error. If  $IQR \approx 1$  (>1), this means that the emulator prediction uncertainty under-(over-)estimates the prediction error, i.e., the emulator provides over-(under-)confident predictions".

In addition, we have included a new reference in Appendix D for the formal description of the *CRPS* score, i.e., Bracher et al. (2021).

**Added reference**

Bracher, J., Ray, E. L., Gneiting, T., & Reich, N. G., 2021. Evaluating epidemic forecasts in an interval format. PLoS computational biology, 17(2), e1008618.

Bellon-Maurel, V., Fernandez-Ahumada, E., Palagos, B., Roger, J.M. and McBratney, A.: Critical review of chemometric indicators commonly used for assessing the quality of the prediction of soil attributes by NIR spectroscopy. TrAC Trends in Analytical Chemistry, 29(9), 1073-1081, 2010.

Caption of Figure 5. Please specify explicitly that the performance statistics are computed over test samples unseen during emulator training. This has been specified.

L239-240. Please specify here if this is performed using the full MME (in contrast to the reduced MME used for the experiments of Table 2).

This has been specified. The new description of the procedure in Sect. 2.4.1 and 2.4.2 should help to clarify as well.

Caption of Figure 6. Typo: constructed using the Monte-Carlo based procedure. Also, please specify the confidence interval corresponding to the likely range.

This has been corrected. The specification has also been added.

Caption of Figure 7. Please specify explicitly that the performance statistics are computed over test samples unseen during emulator training.

This has been specified.

L273. Please remove "Interestingly", as it is preferable to let readers decide what they find interesting.

This has been corrected.

L278. "twice that of the third most important contributor, i.e., woCISM". The medians of RAE relative difference are very close. Please be more specific in quantification of the performance. The description of the results has fully been revised.

Caption of Figure 8. Please specify explicitly that the performance statistics are computed over the same test samples as in Figure 7.

This has been specified as well as in the new Fig. 9.

Figure 8. These results show that (i) excluding SSP126 and SSP245 has negligible impact for predicting in the GSAT range  $\geq 3.83$ °C, and (ii) excluding SSP585 has negligible impact for predicting in the GSAT range  $\leq 2.14$ °C. This should be mentioned and discussed briefly in the text.

This has been added on page 17, line 367.

L285. Please rephrase: "has the largest impact almost at the same level". This sentence has been modified.

L287. Mention to Table 1 is wrong.

This has been corrected.

L288. Please remove "it is interesting".

This has been removed.

L289. "The analysis of the other GSAT intervals" should be: The analysis of the GSAT interval 3.34 to 3.83°C.

This has been reformulated.

L295. Here, I believe that this analysis applies to the random samples drawn as explained in Sect. 2.4.2, and not to the test samples explained in Sect. 2.4.1. Please specify this explicitly. This has been clarified in Sect. 3.3, page 18, line 388.

L305. "under-estimated by more than 25%": Fig. 9 shows  $\sim$  22%. We thank Dr Verjans for noticing this inconsistency. This has been corrected.

L307. Please remove "Interestingly".

This has been removed.

L307-311. How can this contrasting result be explained? A perfect performance would mean that all quantiles remain unchanged. As such, why do larger changes in quantiles do not lead to worse performance? The authors should explain this.

We recognize that this statement is too simplistic, and we have now improved our analysis. Our results in Sect. 3.2 (page 17, Fig. 9 and 10) show that there are two main drivers of the performance depending on the GSAT scenario considered, i.e., 'woCISM' and 'woMAR' at low GSAT and 'woSSP585' and 'woMAR' at high GSAT. This means that there is no reason to opposite the two analyses, performance (Sect. 3.2) and percentile changes (Sect. 3.3), and there are some consistencies in the results.

We agree with Dr. Verjans that a perfect performance would mean that all quantiles remain unchanged. However, the opposite situation is not necessarily always valid, i.e., having poor performance scores do not necessarily mean that all levels of percentile are affected in the same way. The analysis of the percentile changes highlights this aspect. This is now clarified in Sect. 3.3 and in Sect. 4.1. Please refer also to our reply to comment 1.

Caption of Figure 9. Please specify that these results are computed from the random samples as explained in Sect. 2.4.2 and not from the validation procedure (if I understood correctly). This has been specified.

Table 3 (row SSP-RCP). Please specify that "the strong linearity of the Greenland ice sheet response with global temperature" is valid for the 2100 timescale. This has been specified.

Table 3 (row ISM choice). I believe that the under- versus over-estimation depends on the specific ISM that is excluded. For example, if the CISM model predicts consistently higher slc values than other ISMs, then the experiment CISM would over-estimate the left-out slc values (as it is the case here). However, if the CISM model was predicting consistently lower slc values, then experiment CISM would lead to under-estimation of the left-out slc values. And reciprocally for the experiment woCISM. Therefore, I do not believe that such a general conclusion can be made about over- versus under-estimation (this links to General comment 1). Similarly in the Abstract, the word "under-estimations" may be misleading.

We agree that the relation to the considered MME and the particular RCM or ISM model used should be made clearer. We propose a new formulation in Table 3 as follows: "Excluding the most frequently selected ISM in the considered MME, i.e., CISM, has a significant impact on emulator performance and percentile values with a more pronounced effect for low GSAT values. The opposite situation, i.e., limiting to CISM, leads to changes of lower magnitude." The term "under-estimation" has been removed from the abstract.

L335. "Here, 'woMAR' is not necessarily the highest contributor to the changes". Please explain this (see comment about L307-311).

We have improved the presentation of the results. Please refer also to the reply to comment 1 above. As underlined in Sect. 4.1, page 21, from line 433: "Our results show that the magnitude of the influence depends on the GSAT scenario considered, the performance criterion and the target percentile level. For the high GSAT scenario, the exclusion of SSP5-8.5 has as much impact as the exclusion of MAR on emulator performance, and is even the biggest contributor to changes in the high percentiles. For the low GSAT scenario, excluding CISM has as much impact as excluding MAR on the emulator performance, and contributes most to changes in the low percentiles".

L342. Please explain the reasons in the MME design that explain why woCISM leads to a larger perturbation of the member distribution than woMAR (e.g., experiments of a specific SSP scenario have only been done with the CISM model, etc.).

We have provided an additional analysis of the *slc* CDFs in Sect. 4.1, on page 21, from line 438 as follows: "The decrease in MME size induced by 'woCISM' and 'woSSP585' is smaller than that induced by 'woMAR', on the order of 70%, suggesting that it is not only a problem of 'size' but also a problem of the type of information that is removed from the MME. Figure 11c shows that, when applying 'woSSP585' experiment, the emulator is learned with *slc* spanning a restricted range lower than that of the original MME. This means that the emulator is built with little information on large *slc* values, and to predict cases associated to high GSAT scenarios, the RF model mainly relies on extrapolation. This is a situation where emulator methods such as RF can fail completely; see e.g., Buriticá & Engelke (2024). Analysis of Figs. 11a and b helps to understand why "woMAR" and "woCISM" induce roughly equivalent changes for the 2°C GSAT scenario, as the *slc* CDF appears to be similarly disrupted by the application of these experiments with a CDF shift towards low-to-moderate *slc* values, particularly in the *slc* range of ~5 to ~15cm. This means that the emulators are built on members whose *slc* values span approximately the same range.

New Figure 11: Comparison between the Cumulative Distribution Function (CDF) of *slc* in 2100 of the original MME (reference) and of the reduced MME after application of the emulator experiments, 'woMAR' (a), 'woCISM' (b), 'woSSP585' (c).

**L345. "further work should look into this aspect in more detail". This should be done as part of this study.**

Although informative, the CDF analysis described above is not sufficient to fully answer the question. We believe that quantifying the importance of group of members is related to the information removed when applying the experiment, i.e., to which extent the configurations of the input variables associated with the corresponding members are valuable for the RF emulator to make predictions for ISM/RCM unseen in the training dataset as explained in more details in Sect. 2.3. This is underlined in Sect. 4.1, page 22, lines 454-456.

We have underlined in Sect. 4.1 (page 22, lines 457-462), a line for further work as follows: "The interaction between the reduction in the size of the MME and the type of information important for the training of the emulator is however complex due to the multiple joint effects to be taken into account between the inputs. From a methodological viewpoint, this calls for further developments, in particular by relying on the data valuation domain (Sim et al., 2022). These types of tools aim to study the worth of data in machine learning models based on similar methods as the ones used by Rohmer et al. (2022) in the context of sea level projections. Transposed to the MME context, these tools could be used in future studies to assess the impact of each member in the emulator's predictions, i.e. the worth of each member".

L360. "This also relates to the question of initialisation (and initial mass loss estimates) where the RCM choice is a key ingredient (e.g., Otosaka et al., 2023)". It is unclear to me what this sentence implies, and which message the authors try to convey.

We agree that this sentence is outside the scope and have decided to delete it.

L363. "First, our study contributes (...) according to the same report". I do not see how this is a contribution of this study. Here, the authors simply provide the Greenland sea-level rise contribution estimates of the ICCP.

We agree that this point was made not sufficiently explicit. We have added in Sect. 4.2 (page 23, lines 480-484) the two following sentences to explain how our study can contribute to a better understanding on the Greenland ice sheet melting to sea-level rise: "Here, we showed that some choices made by modelers, such as the tidewater glacier retreat parameter, have a minor impact on the spread of the Greenland sea-level rise contribution, whereas others, such as using only MAR as a regional climate model, have a large impact. These findings can be useful to inform future modelling experiments, and could help identifying where modelling efforts could focus to better characterize the spread of the projected contribution of the Greenland ice-sheet and to increase our understanding of that spread."

L390-392. "Indeed, scenarios based on global warming levels can be potentially better understood by stakeholders than the SSP or RCP scenarios, and also allow users to better make the link with the climate objectives set out in the Paris agreement to stabilize climate change well below 2°C GWL". This reads as a personal opinion of the authors, so please rephrase or remove.

We thank Dr Verjans for this comment. We have clarified that this statement is not a personal opinion but a choice made by adaptation decision makers in at least one country, i.e., France. We have added a clarification in Sect. 4.2 (page 24, lines 506-510) as follows: "For example, the latest adaptation plan in France requires adaptation practitioners to test their adaptation measures against a climate change scenario reaching 2°C in 2050 and 3°C in 2100 globally (Le Cozannet et al., 2025). Motivations for considering these GWLs rather than SSP or RCP scenarios include their perceived clarity for a wide range of adaptation practitioners, as well as the direct links that can be made with the climate objectives set out in the Paris agreement to stabilize climate change well below 2°C GWL."

L399. Please specify: future sea level by 2100. This has been added.

L400. Please specify: high importance for emulator accuracy.

Results show that this importance is for both situations, emulator performance and percentile estimates. This has now been specified.

**L417-421. This sentence is too long and confusing.**

The sentence in Sect. 5 (on page 25, lines 545-547) has been simplified as follows: "This could be done iteratively. The procedure could alternate between simulation phases, i.e. either test simulations to assess sensitivity to different inputs, or small exploratory sets that do not use all the available computing time/human/project resources, and training and retraining of the emulator."

Equation A1. What does s represent in this equation?

This is mistake. This has been removed.

L609. "where I(A) is the indicator operator". The parenthesis notation is not used in Equation A2.

This has been corrected.

L612 and L615. These two sentences repeat the same information.

We have removed Line 612.

*L624. Please use consistently*  $q_{\tau}$  *or*  $q^{\tau}$  .

This has been corrected.

*Equation B1. Please define inf as the infimum function.*

This has been specified.

L628. Typo: "weighed".

This has been corrected.

L637. Please specify that Step 2.2 approximates the CDF of slc|~x (if I understand correctly). Dr Verjans is correct about that. We have specified it.

L638. If I understand correctly,  $Q_{\tilde{u}}^{\alpha}$  is not specific to a single  $\tilde{u}$  sample, but depends on the full set of  $\tilde{u}$ 's sampled in Step 2.1. This is in contrast to  $q^{\tilde{u}}(slc|\tilde{x})$  in Step 2.2. If this is correct, then this notation is confusing, and I recommend writing  $\tilde{Q}^{\alpha}$  instead.

The original intent was to indicate the dependence to the full set of  $\tilde{u}$ 's sampled in Step 2.1. From Dr Verjans's comment, we have feeling that its adds more confusion, and we have removed this notation and used the one proposed.

L642. This "variability" corresponds to the emulator uncertainty about a given quantile level  $\alpha$ . But, if I understand correctly, the range  $[Q^{\alpha/2}; Q^{1-\alpha/2}]$  gives the  $1-\alpha$  confidence interval of the emulator prediction for  $(slc|\tilde{x})$ . If this is correct, please specify it. This has been specified when describing step 2.3.

L647. Please specify that the p-value quantifies how unlikely the variable importance in the nonpermuted data is with respect to the null distribution of variable importance reached from the permutations.

This has been specified.

L657. Typo: "should retained".

This has been corrected.

Figures S2, S3, and S4. Thes figures are identical. Is this an error?

We confirm that this is not an error. We agree that with this type of representation, little differences can visually be seen. We now propose a new representation with CDFs and a zoom on one case to better highlight the impact of the emulator error.

**References**

Tamsin L. Edwards, Sophie Nowicki, Ben Marzeion, et al. Projected land ice contributions to twenty-first-century sea level rise. Nature, 593(7857):74–82, May 2021. ISSN 1476-4687. doi: 10.1038/s41586-021-03302-y. URL http://dx.doi.org/10.1038/s41586-021-03302-y.

Hélene Seroussi, Vincent Verjans, Sophie Nowicki, et al. Insights into the vulnerability of Antarctic glaciers from the ismip6 ice sheet model ensemble and associated uncertainty. The Cryosphere, 17(12):5197–5217, December 2023. ISSN 1994-0424. doi: 10.5194/tc-17-5197-2023. URL http://dx.doi.org/10.5194/tc-17-5197-2023.

David B. Stephenson and Francisco J. Doblas-Reyes. Statistical methods for interpreting monte carlo ensemble forecasts. Tellus A: Dynamic Meteorology and Oceanography, 52(3):300, January 2000. ISSN 1600-0870. doi: 10.3402/tellusa.v52i3.12267. URL <a href="http://dx.doi.org/10.3402/tellusa.v52i3.12267">http://dx.doi.org/10.3402/tellusa.v52i3.12267</a>.

**Referee #2:**

Review of "Lessons for multi-model design drawn from emulator experiments: application to a large ensemble for future sea level contributions of the Greenland ice sheet" by Rohmer et al.

This article is a revised version of a previously submitted paper. While some changes have been made to address my comments—for which I would like to thank the authors—I do not believe that the current version is ready for publication, as it does not yet meet the standards of clarity and readability required for publication. I continue to believe that the content of this article is noteworthy and valuable for the scientific community, but further work is needed to make it publishable.

We would like to thank Referee #2 for taking the time to participate in this new round of reviews. We really appreciate the opportunity Referee #2 has given to continue the revisions based on the various comments/suggestions.

In particular, we have taken care to:

- Provide sufficient details on the methods (Sect. 2.3 and 2.4), without using too technical description so that the article can be followed by non-experts;
- Underline the presence of other sources of uncertainty;
- Improve the quality of the figures.

My review is structured in two parts. First, in the general comments section, I address the major changes made by the authors in response to my original remarks. Then, I provide a list of specific comments.

These are mostly minor suggestions and corrections, but they are relatively numerous. In particular, I have not gone through the entire text in detail. To facilitate the next round of revisions, I would therefore ask the authors to carefully proofread their revised manuscript, which would help streamline the revision and correction process. As a side note, the article would be more pleasant to read if the figures were included in vector format or, at least, in higher resolution, so that they do not appear blurry.

We apologize for this problem. This is due to the word-to-PDF conversion of the preprint. We have now converted with a higher quality of the figures. In addition, we have uploaded a zip file with all figures on the github <a href="https://github.com/rohmerj/MMEdesign/blob/main/Figures.zip">https://github.com/rohmerj/MMEdesign/blob/main/Figures.zip</a>. Note that the Copernicus platform does not allow to upload the figures as separate files.

**General comments**

General comment 1. I would like to thank the authors for their efforts to reorganize the paper to make it more accessible to a broader audience.

We appreciate this positive feedback. Please note however that Referee #1 asks for detailed technical description, and we are striving here to satisfy two contradictory (but justified!) demands. Most of the technical details are kept in Appendices but additional technical descriptions have been added in Sect. 2.3 to clarify how the RF model makes predictions as well in Sect. 4.1 to provide more details on the validation procedure.

General comment 2. I am pleased to see that a brief discussion on the weight attribution of each ensemble member is now included in the paper. However, unless I am mistaken, this appears primarily in the interpretation of results, and thus appears rather late in the main text (it is first mentioned on page 8). Evaluating the predictive quality of the members is crucial in the context

of future ensemble studies; indeed, it is a key step in linking numerical simulations to observations and constraining the former using the latter. It is also a quite natural step from a Bayesian approach, as it allows for relaxing the assumption of a uniform prior. Therefore, it would make sense to introduce this question earlier in the main text, so that the reader clearly understands how each member is to be compared. I suggest adding such a discussion when the inputs are presented in subsection 2.1, at the end of page 5. The authors could also briefly mention it in the introduction when outlining the scope of the paper.

We thank Referee #2 for this comment. We agree that our assumption of the weight attribution should be better underlined. Therefore, we have specified it in Sect. 2.1 on page 7, lines 146-149 as follows "In this study, we assume that each member has the same weight, in particular, without differentiating members based on their reliability (e.g., low-resolution models compared with high-resolution models) or any observational constraints (as done for instance by Aschwanden and Brinkerhoff (2022)). Under this assumption of uniform weighting, [...]".

As suggested, we have also briefly introduced this aspect in the introduction on page 3 in line 76-77: "[...] we define a series of numerical experiments (referred to as emulator's experiments) that are closely related to practical MME design decisions consisting in leaving out specific results from the original MME assuming that all members have the same weight in the ensemble".

We recall also this assumption in the description of the emulator experiments in Sect. 2.2 (page 9, line 168).

Finally, we mention a weighting approach as an avenue of this work in Sect. 5 on page 24 in lines 529-533.

General comment 3. I would like to thank the authors for the additional details concerning the interpretation of the parameter  $\kappa$ . While I am not entirely convinced by the rationale of comparing a single parameter to a full forcing scenario choice, I am fine with the authors retaining this aspect of their analysis.

I still wonder whether the meaning of this parameter could be made clearer by explicitly renaming it as 'ocean forcing' parameter instead of 'retreat' parameter, particularly if it is associated with uncertainties in ice—ocean coupling, rather than being a parameter intrinsic to the ice-sheet model itself.

This would make it clearer that it represents a forcing.

We thank Referee #2 for this comment. We appreciate the point of Referee #2, and understand the reasons for the possible confusion. However, we would like to underline that it is called "retreat" parameter and "retreat" parameterization in Edwards et al. (2021) and Rahlves et al., (2025) and above all it is also how in the community the modelers refer to it since ISMIP6. For sake of consistency, we think it is clearer to keep it as is.

Furthermore, it seems to me that the structural uncertainty of the ice-sheet models has not been addressed in the paper. This omission might bias the results by underestimating the impact of uncertainties related to ice-sheet physics and models.

We agree that this aspect deserves further investigation; We have indicated it in the conclusions on pages 25, lines 533-536, as follows: "To address this question, a wider range of uncertainties should be considered, more specifically model and structural uncertainties (i.e. uncertainty in the formulation of the model and its ability to represent the physics of the system), in addition to uncertainties in model parameters (related to ice dynamics and atmospheric/oceanic forcing)".

**Specific Comments**

*Note: I am using the author's tracked changes document for the line numbers.*

- (1) [Lines 32–33] '(IPCC: e.g. Lee et al., 2021)'  $\rightarrow$  '(IPCC; e.g., Lee et al., 2021). This has been corrected.
- (2) [Line 36] '(e.g. Knutti et al., 2010)' $\rightarrow$ '(e.g., Knutti et al., 2010)'. This has been corrected.
- (3) [Line 37] The fact that each member evenly spans a representative set of plausible realizations is somewhat misleading, as this is only true if no additional information is not available (e.g., observations). This would benefit from further clarification; see general comment 2.

Please see our reply to comment 2.

- (4) [Line 40] '(e.g. Merrifield et al., 2020)'→ '(e.g., Merrifield et al., 2020)'. This has been corrected.
- (5) [Lines 52–55] This sentence is too long. Consider splitting it in two, maybe after the 'thoroughly'.

This has been split into two sentences as suggested.

- (6) [Line 82] 'the main model parameter' is ambiguous: do you mean  $\kappa$ ? This has been reformulated as "[...] the retreat parametrisation described below"
- (7) [Line 94] Consider adding what you mean by 'as best as possible', as it is vague on its own. Maybe mention that the misfit between computed and observed surface velocities and/or thicknesses is minimized?

We agree that this may be read like a vague statement, but it is vague on purpose. It goes back to the ISMIP6 protocol, which was intentionally very open to accommodate a range of different modelling approaches. It was really up to the individual modellers to interpret that as they saw fit. We would like to keep the original formulation.

- (8) [Line 98] '(Slater et al., 2020, 2019)'→'(Slater et al., 2019, 2020)'. This has been corrected.
- (9) [Line 108] 'parameter values' is ambiguous: do you mean the ice-sheet parameter values? This has been corrected.
- (10) [Lines 109–112] I would remove entirely the discussion about the merge of the two inputs and present your final setup more directly, namely, the use of a GSAT that corresponds to a combination of SSP-RCP and GCM. This would be easier to follow.

We have simplified the presentation by specifying in the main text that: "The inputs below the double line in Table 1 are those used for the building of the RF emulator, in particular with the use of global annual mean surface air temperature change relative to 1995-2014, denoted GSAT, that corresponds to a combination of SSP-RCP and GCM by following a similar approach as Edwards et al. (2021)".

(11) [Line 114] 'The inputs from the double line' is not clear. Do you mean the inputs below the double line?

This has been corrected.

(12) [Table 1] Please be consistent in your system of notations. Some of the names start with capital letters (e.g., 'Sliding'), others do not (e.g., 'thermodin.').

This has been corrected.

(13) [Table 1] 'thermodin'→'thermodyn' or even 'thermo'. This has been corrected.

(14) [Table 1] One way to simplify the reading of the table would be to use math symbols to clarify whether the variables are categorical or continuous. For example, the ISM models would become {CISM, Elmer/Ice, GISM, IMAUICE}, while the resolution would become [1, 40] km.

We thank Referee #2 for this suggestion. As pointed by Referee #1, the treatment of categorical variables is key in our study, and we believe that Table 1 should specify this aspect explicitly. Therefore, we choose to keep this column.

(15) [Line 118] 'as a particular'  $\rightarrow$  'a particular'. This has been corrected.

(16) [Line 121] 'expressed in meters sea level equivalent  $SLE' \rightarrow$  'expressed in meters sea level equivalent, SLE'.

This has been corrected.

(17) [Figures 2–4] I think it would make more sense to first present the inputs (Figures 3 and 4) before the output (Figure 2).

We thank Referee #2 for this suggestion. This improves readability.

(18) [Figure 2] Vertical label: 'density'→ 'PDF'. This has been modified.

(19) [Figure 2] Caption: 'Probability density function of the sea level contribution in 2100 (with respect to 2014) from the Greenland ice-sheet (in cm seal level equivalent, SLE) based on the raw MME ensemble data considered in this study' → 'Probability density function of the sea level contribution of the Greenland ice sheet in 2100, with respect to 2014, based on the raw MME ensemble data considered in this study'.

We thank Referee #2 for the suggestion. This has been corrected.

(20) [Line 132] 'highest importance' is ambiguous. Do you mean 'that contributes the most to the uncertainty'?

This has been reformulated as suggested.

(21) [Line 137] Consider adding one sentence that quickly explains why the design of experiments was indeed unbalanced.

We thank Referee #2 for this comment. A sentence has been added (Sect. 2.1, page 6, lines 132-134), namely that "this parameter was sampled for only 3 different values by most models (the median, the 25% and the 75% percentile), and the additional 2 values were only sampled by one ISM at a later stage to broaden the parameter range".

(22) [Figures 3, 4, C1 and C2] Consider drawing the plots with the count number in the y axis, as that is more common.

This has been modified for Figures 3, 4 (now Figures 2 and 3) as well for Figure C2. We chose not to change Fig. C1 because of the long list of variables to be specified on the axis.

(23) [Line 154] '(named emulator)' is not necessary here as you have already introduced several times the notion of emulator previously.

This section has been reformulated, and the definition of emulator is in the introduction.

- (24) [Line 156] Consider putting the reference to the overview as new separate sentence. This has been corrected.
- (25) [Line 169] I found the reference to a conditional mean to be not very clear. On what is the mean conditioned here?

This is more clearly defined in Appendix A: equation A1. To avoid any confusion with non-expert readers, we have decided to refer to the mean provided by the RF model.

(26) [Line 184] What is meant here by 'tolerance'?

This term is confusing and has been removed here. The "tolerance" is now clarified in the description of Monte-Carlo procedure in Sect. 2.4.2 – step (2).

- (27) [Line 184] 'in Sect. 2.4'  $\rightarrow$  'in the next section'. This has been corrected.
- (28) [Line 204] You mention AR6 here, but this has not been introduced/defined before. We have more clearly defined it.
- (29) [Line 209] Consider adding an adverb at the beginning of the sentence here (e.g., 'consequently'), so that it is clear that the 200 number is directly linked to the 50 number before, and not a new parameter for the validation exercise.

  This has been added.
- (30) [Lines 212–218] Given that you introduce three performance criteria, you should use three distinct items, not two.

This has been corrected.

(31) [Line 219] Consider removing 'for a GSAT scenario' from the title of the subsection. Both subsections 2.4.1 and 2.4.2 depend on the GSAT scenario.

This has been corrected.

(32) [Line 266] [4.6; 7.4cm]  $\rightarrow$  [4.6 cm; 7.4 cm] or [4.6; 7.4] cm. This has been corrected.

(33) [Line 266] [10.4; 17.0cm]  $\rightarrow$  [10.4 cm; 17.0 cm] or [10.4; 17.0] cm. This has been corrected.

(34) [Figure 6] Vertical label: 'density'→ 'PDF'. This has been corrected.

(35) [Line 269] 'constructed the Monte-Carlo-based procedure': there seems to be a missing word here.

This has been corrected.

- (36) [Line 281] 'whatever the performance criteria'→'for every performance criterion'. This has been corrected.
- (37) [Line 286] 'Table 1'→Table 2'.

This has been corrected.

- (38) [Line 367] '(based on Goelzer et al. (2020))'→'(based on Goelzer et al., 2020)'. This has been corrected.
- (39) [Lines 410–411] '(Merrifield et al., 2023; Evin et al., 2019)'  $\rightarrow$  '(Evin et al., 2019; Merrifield et al., 2023)'.

This has been corrected.

- (40) [Line 411] 'take'→ 'took'. This has been corrected.
- (41) [Line 609] You use slc with a superscript for the index, and then with a subscript later in the text. Please be consistent in your system of notations.

  This has been corrected.
- (42) [Line 612] 'By nature'  $\rightarrow$  'By construction'. This has been corrected.
- (43) [Line 615] 'squared errors': errors of what?

This has been replaced by the more specific term used for training RF emulators, i.e., the variance.

Orleans,
October 1st, 2025
J. Rohmer1 on behalf of the co-authors

1 BRGM, 3 av. C. Guillemin - 45060 Orléans Cedex 2 – France

---

## Author Response (AR3)

**Replies to Referees' comments on "Drawing lessons for multi-model ensemble design from emulator experiments: application to future sea level contribution of the Greenland ice sheet" (egusphere-2025-52)**

We would like to thank the two Referees for taking the time to participate in this third round of reviews. We agree with most of the suggestions and, therefore, we have modified the manuscript to take on board the comments and suggestions. We recall the reviews and we reply to each of the comments in turn (outlined in green). The page and line numbers are those of the document with track-changes.

In addition to the corrections suggested by the Referees, we have made some changes:

- corrections of typos and reformulations;
- correction of Fig. 4 where the upper bound of the likely range was corrected (it corresponded to the 95th percentile);
- clarification of the GSAT definition by specifying that we analyse GSAT change relative to 1995-2014.

**Referee #1:**

This is my third review of this manuscript, following the second round of revisions. I am very pleased with the revisions made since the last round.

We would like to thank Dr Verjans for taking the time to participate in this third round of reviews. We really appreciate the opportunity Dr Verjans has given to continue the revisions based on the valuable comments/suggestions.

The most notable improvements include: - The analyses and conclusions that were previously not well supported by quantitative results have been thoroughly re-worded and/or removed. The arguments about the findings are now presented in a more methodological and substantiated manner.

- The evaluation procedure has been clarified. The revisions of Sects. 2.4.1 and 2.4.2 allow the reader to fully understand the predictive performance of the emulator.

More generally, I commend the authors for having very substantially improved the quality and scientific robustness of the manuscript compared to the very first submission. I also thank the authors for their efforts in addressing the concerns that I previously raised. In this review, I only raise one Minor General Comment, and some Specific Comments. My comments focus on some remaining minor issues, and on improving the clarity and readability of the manuscript. I believe that once these final points, as well as potential comments from other reviewers, are addressed, this study will be a valuable contribution to The Cryosphere. Line numbers refer to the main manuscript without tracked changes.

**Minor General Comment: Adding an explicit statement about ensemble size importance**

One of the most important driver of emulator performance is the size of the training set. I believe that the reduction in the MME size drives a large part of the strong performance decrease in the woMAR, woCISM, and woSSP585 experiments. It is not fortuitous that these 3 experiments

lead to the largest performance decreases (Fig. 7), and are the ones with most restricted number of available members (Table 2). This is a strong argument for a key conclusion: the availability of large ensembles of ISM simulation outputs is the most important factor for developing accurate and reliable emulators.

In the current manuscript, this is is alluded to (e.g., L429-430). But I believe that a clear and explicit statement about the critical importance of the MME size should be added in both the Abstract and in Sect. 5 "Concluding remarks and further work". This would emphasize to the glaciological community that large participation to projects such as ISMIP6 are needed for designing useful emulators, with as many simulations from as many groups as possible. We thank Dr. Verjans for this suggestion.

We propose the following formulation in the abstract: "These results point to the size of the training set as the key driver of the changes, which supports the need for large ensembles to develop accurate and reliable emulators, hence encouraging large participation to projects such as the Ice Sheet Model Intercomparison Project ISMIP".

In the concluding remarks (on page 24, lines 538-544), we have also emphasised the same message as follows: "They also show that an ensemble designed only with a unique ISM and RCM model, i.e., here with the one that is most frequently selected in the considered MME, has non-negligible implications. These results point to the size of the training set as the key driver of the changes in the emulator performance and percentile estimates, hence underlying the need for building large ensembles to develop accurate and reliable emulators. Broad participation in projects such as ISMIP, with as many simulations as possible contributed by numerous groups, appears to be an effective option to this end."

**Specific comments**

L19-20. "for low and high levels of warning": this sentence somewhat hides that the predictive performance is not satisfactory for intermediate levels of warming. Please specify this explicitly in the abstract.

We have reformulated as follows: "We use these experiments to build a random-forest-based emulator, whose predictive capability to assess Greenland sea level rise contributions in 2100 proves very satisfactory for low and high levels of warming but less effective for intermediate levels."

L20. Typo: "warning". This has been corrected.

L175. "(...) are used to rank the different emulator experiments in terms of influence." I see what the authors mean here. However, I think that the wording could be misinterpreted. It is not clear what the "influence" of an experiment refers to. Maybe rephrase this sentence, focusing more explicitly on the different impacts on emulator performance of different MME restriction experiments.

This is now specified as follows: "Quantified criterion changes are then used to rank the different emulator experiments in terms of the magnitude of their impact on emulator performance and emulator-based probabilistic predictions".

L247. Throughout the manuscript, I find the notations  $CA^{\alpha}$  and  $PI^{\alpha}$  potentially confusing. In statistical terminology,  $\alpha$  typically denotes the significance level, so  $\alpha=0.1$  corresponds to the 90% confidence level, for example. However, the authors use expressions such as "CA at level 90%" (caption of Figure 5) and  $CA^{90}$  (e.g., L300). This is inconsistent with standard statistical conventions. Please revise the notation and associated wording to clarify whether superscripts refer to the  $\alpha$  level or to the confidence range. For example, I would recommend writing  $CA^{1-\alpha}$ , which would be consistent with, for example, writing  $CA^{90}$ .

We thank Dr. Verjans for this clarification. We have corrected the notation as suggested.

L248. Typo: "fall" should be falls. This has been corrected.

*L257. Typo: "scenario" should be scenarios.* This has been corrected.

L290. The word "predictability" is misused here, since this refers to an intrinsic characteristic of a system. Please change this to predictive capacity or something similar. This has been replaced by predictive capability.

Caption of Figure 6. "Note these probability density functions are derived using the conditional mean of the RF emulator (Appendix A) and do not include uncertainty arising from the emulator itself". This seems to contradict the explanations provided in Sect. 2.4 (L269-270). Please verify if the emulator uncertainty is included or not.

We confirm that Fig. 6 has been built by using the RF mean, i.e., without including the emulator uncertainty. We have added elements on this aspect in Sect. 3.1 as follows: "The results are computed using the mean of the RF emulator (Appendix A), and do not include uncertainty arising from the emulator itself. The procedure described in Appendix B is further applied to assess the impact of the emulator uncertainty, and shows that the width of the 90% confidence interval for the percentiles considered remains in the order of 0.1 cm, hence indicating minor influence of the emulator uncertainty in this case".

L316. Please specify here relative to what the "relative differences" are computed. I believe that it is relative to the performance metrics computed from the validation test applied without leaving experiments out, but this should be 100% clear.

The introduction of Sect. 3.2, on page 14, lines 326-332 has been reformulated as follows: "We analyse in Figure 7 the impact of design decisions on the RF predictive capability and on the reliability of the RF prediction intervals. The decrease of RF predictive capability is measured by the decrease of the relative differences of RAE and CRPS (Fig. 7a,c) and the increase of the relative differences of  $Q^2$  (Fig. 7b). The reliability of the RF prediction intervals is measured by  $CA^{90}$  and  $CA^{50}$  (Fig. 7d,e), which are respectively related to the prediction intervals at the 10% and 50% significance level, and by IQR (Fig. 7f). This assessment is conducted relative to the performance metrics of the reference solution computed from the validation test applied without excluding the experiments as explained in Sect. 2.4.1".

L330. Change "goes along" to: goes with. This has been corrected.

L359. Change "turns to be worse" to: is worse. This has been corrected.

L364. This should be: (...) than that of 'woSSP585' (...). This has been corrected.

Caption of Figure 10. The word "quantile" in the last sentence should be plural. This has been corrected.

L412. The word "significantly" should be replaced by substantially or a similar word. That is because, according to the error bars shown in Fig. 10, Q50 and Q83 are also significantly influenced, although the magnitudes are small.

This has been replaced by "substantially".

L418. "(...) regardless of the GSAT change and the considered percentile". I believe this is not true. See for example GSAT 2°, Q17, Narrow Kappa. Please consider revising this sentence. We have nuanced the remark by pointing out this exception. Note that most results for Kappa experiments have values on the same order of magnitude than the emulator uncertainty. This is also underlined.

L443-447. I agree with this analysis. However, it does not explain why woCISM has stronger impacts on performance at  $GSAT = 2^{\circ}$  than at  $GSAT = 4^{\circ}$ . In fact, I find this difference in woCISM influence somewhat surprising, given that the CDF seems more affected at high rather than low slc values (Fig. 11b). I would appreciate if the authors could attempt to explain this, or at least mention this aspect in the manuscript.

We recognize that Fig. 11 does not explain all aspects of the problem. We now clearly underline this in Sect. 4.1, page 22, lines 475-477 as follows: "Analysis of Fig. 11 reveals certain similarities in the effect of the different emulation experiments, but is not sufficient to explain all aspects of the problem; for example, this type of analysis does not fully explain why 'woCISM' has a stronger impact on performance at GSAT change of 2°C than 4°C."

L454-456. This sentence is very unclear to me. I read it multiple times, but I cannot understand the message that the authors try to convey. Please rephrase.

We have reformulated (now lines 470-472) as follows: "This suggests that removing members associated with other ISMs / RCMs from the training set has an impact, because these members contain information relevant to the RF emulator capability to make predictions, especially in the situations explained in Sect. 2.3, for levels of categorical variables not seen in the training dataset".

*L476. Please add the word estimated: the estimated contribution.* This has been corrected.

L533-536. I appreciate this more extensive discussion on the various types of uncertainties. To make this discussion more complete, I recommend mentioning briefly the influence of irreducible uncertainties on Greenland sea-level contribution projections (e.g., Verjans et al., 2025). It would be valuable to include a short statement on how such irreducible uncertainties can be addressed through the use of emulators.

We thank Dr. Verjans for this suggestion. We have added this aspect as follows: "To address this question, a wider range of uncertainties should be considered, more specifically model and structural uncertainties (i.e. uncertainty in the formulation of the model and its ability to represent the physics of the system), in addition to uncertainties in model parameters (related to ice dynamics and atmospheric/oceanic forcing), but also irreducible uncertainties such as

internal climate variability as investigated by Verjans et al. (2025) on Greenland sea level contribution projections".

Regarding the question of climate variability, we believe that the emulators could play a useful role to explore the space of climate forcings; more particularly because climate forcing is kept constant in ISMIP6. This should however be done in addition to the exploration of other uncertain factors / parameters. We have added a short statement in this sense as follows: "Here, emulators are expected to play a key role to explore this wide uncertain space thoroughly".

However, we have chosen to keep the statement relatively generic, because addressing this problem may potentially require additional developments, which are out of the scope of our study. Depending on the number of uncertainty sources, combination with adaptive sampling (e.g., Rohmer and Idier, 2012) may be required to improve the uncertain space exploration. The influence of aleatoric uncertainties may be viewed through the lens of stochastic simulation codes (e.g., Marrel et al., 2012), which may also require adapted emulators. These possible lines of further developments are only suggestions, and we believe that further analysis is here required.

**References (for the replies)**

Rohmer, J., & Idier, D. (2012). A meta-modelling strategy to identify the critical offshore conditions for coastal flooding. *Natural Hazards and Earth System Sciences*, 12(9), 2943-2955. Marrel, A., Iooss, B., Da Veiga, S., & Ribatet, M. (2012). Global sensitivity analysis of stochastic computer models with joint metamodels. *Statistics and Computing*, 22(3), 833-847.

L775. Please revise the notation by respecting the convention that  $\alpha$  represents the level, not the % of coverage (see comment about L247). Note also that the current notation is in disagreement with the notation of  $Q^{\alpha/2}$ ;  $Q^{1-\alpha/2}$  on L783

We have corrected the notation as suggested in the comment for L247.

L816. The word "score" in "CRPS score" is redundant, please remove it. This has been corrected.

L823-828. Same comment about  $\alpha$  notation as for L247.

We have corrected the notation as suggested in the comment for L247.

From the Supplementary Information.

p2, L7. This sentence does not make sense: "Overall the emulator is of moderate magnitude". This is replaced by "Overall the emulator uncertainty is of moderate magnitude."

Figure S3. I recommend using the same x-axis for all sub-figures. This has been corrected.

**References**

V. Verjans, A. A. Robel, L. Ultee, H. Seroussi, A. F. Thompson, L. Ackermann, Y. Choi, and U. Krebs-Kanzow. The greenland ice sheet large ensemble (grislens): simulating the future of greenland under climate variability. The Cryosphere, 19(9):3749–3783, Sept. 2025. ISSN 1994-0424. doi: 10.5194/tc-19-3749-2025. URL <a href="http://dx.doi.org/10.5194/tc-19-3749-2025">http://dx.doi.org/10.5194/tc-19-3749-2025</a>. We thank Dr. Verjans for the suggested reference which has been added to the reference list.

**Referee #2:**

This is the third round of review for this paper. I would like to thank the authors for their responses to my comments and for the changes made to the manuscript.

We would like to thank Referee #2 for taking the time to participate in this new round of reviews. We really appreciate the opportunity Referee #2 has given to continue the revisions based on the valuable comments/suggestions.

Given that the authors have responded positively to my major comments, I recommend that the manuscript be finalized for publication. I still ask the authors to carefully reread their manuscript, as there are still quite a lot of typos in the text (see specific comments below). Provided the authors also respond favorably to the more technical comments addressed by the first reviewer, I would recommend this paper for publication.

Specific Comments

*Note: I am using the author's tracked-changes document for the line numbers.*

(1) There is inconsistency in the use of hyphens for compound nouns used as adjectives; for example, you use both 'sea level contributions' and 'sea-level rise'. Please stick to one writing style throughout the text.

We thank Referee #2 for noticing this problem. We have corrected by removing the hyphens.

(2) When using mathematical symbols, please use italics. This is particularly relevant for Appendix A ('p'  $\rightarrow$  'p', 'L'  $\rightarrow$  'L', …).

We apologise for this problem. We now use italics for all symbols.

- (3) Please avoid using fractions in running text. Either use a '/' symbol or write a full equation. The notations have been corrected by writing full equations.
- (4) You define slc in several places in the text and use it before it is properly defined. You should define it at the beginning of the manuscript once and for all.

The definition is now provided at the beginning of Sect. 2.1. We have also removed redundant definitions from the main text.

- (5) [Line 17] 'specific set' → 'a specific set' or 'specific sets'. This has been corrected.
- (6) [Line 20] 'warning'→ 'warming'. This has been corrected.
- (7) [Line 74] '([...] by Edwards et al. (2010))'  $\rightarrow$  '([...] by Edwards et al., 2010)'. This has been corrected.
- (8) [Line 78] 'as well' → 'as well as'. This has been corrected.
- (9) [Line 92] '(SSP126, SSP245, SSP585)'  $\rightarrow$  '(SSP1-2.6, SSP2-4.5, SSP5-8.5)'. This has been corrected.

- (10) [Table 1] Follow-up on my previous comment (14): I am not asking to remove the categorical aspect in the column; rather, my suggestion was to use the symbols {a,b} and [a,b] to establish this distinction efficiently. Such notation is used in other multi-ensemble studies. We thank Referee #2 for this clarification. We have corrected Table 1 using the suggested notations.
- (11) [Line 148] '([...] by Aschwanden and Brinkerhoff (2022))'  $\rightarrow$  '([...] by Aschwanden and Brinkerhoff, 2022)'.

This has been corrected.

(12) [Figure 4] Why is there an '(a)' at the beginning of the caption? This is a mistake. This has been corrected.

Orleans, November 7th, 2025 J. Rohmer1 on behalf of the co-authors

1 BRGM, 3 av. C. Guillemin - 45060 Orléans Cedex 2 – France

---

## Author Response (AR4)

**Replies to Referees' comments on "Drawing lessons for multi-model ensemble design from emulator experiments: application to future sea level contribution of the Greenland ice sheet" (egusphere-2025-52)**

Horst Machguth

We thank Prof. Machguth for this additional check.

Throughout the entire document: Please explain abbreviations, or refer to Appendix E (the text has no reference to Appendix E) as early as possible, thereby making clear that all abbreviations/acronyms are explained therein. Currently some abbreviations are explained in the text (e.g. MME on line 5), others not (GSAT on line 122 - note that GSAT is then explained a bit later in the caption of Fig. 3).

We have specified in line 120 that "All abbreviations used in the text are explained in Appendix E".

There should always a be a space between numerical value and unit symbol (except for  $^{\circ}$ , ' and "). Currently this is handled inconsistently in the manuscript.

We have corrected the notations.

line 267: this should be an active citation, "... as in Edwards et al. (2021) ..." This has been corrected.

Figure 5: Consider labeling the x-axis on the plot and also indicate the unit. One label would be enough for all subplots as all x-axes are identical. I might not be familiar with certain terminologies and use of symbols: is there a reason behind combining round and square brackets in all GSAT ranges on the x-axis? Elsewhere in the text ranges of values are indicated using square brackets.

We have modified the figure as suggested. We have also removed the use of parenthesis "(". For your information, this is a mathematical convention to specify the exclusion of the value.

Line 303: I might simply not understand the details, but aren't the small GSAT changes below 2.14 °C shown in black in Figure 5?

We have checked, and we confirm that it is in dark blue; please refer to <a href="https://rpubs.com/mjvoss/psc viridis">https://rpubs.com/mjvoss/psc viridis</a> for any information on the colours used.

Line 348: tiny detail, but RCM already abbreviates the word "model" This has been corrected.

Orleans, November 14th, 2025 J. Rohmer1 on behalf of the co-authors

1 BRGM, 3 av. C. Guillemin - 45060 Orléans Cedex 2 – France